# Rainforest transformation reallocates energy from green to brown food webs

Anton M. Potapov[1,2,3 ✉], Jochen Drescher[1], Kevin Darras[4], Arne Wenzel[5], Noah Janotta[1], Rizky Nazarreta[6], Kasmiatun[6], Valentine Laurent[1], Amanda Mawan[1], Endah H. Utari[6], Melanie M. Pollierer[1], Katja Rembold[7,8], Rahayu Widyastuti[9], Damayanti Buchori[6,10], Purnama Hidayat[6], Edgar Turner[11], Ingo Grass[12], Catrin Westphal[5], Teja Tscharntke[4] & Stefan Scheu[1,13]

Terrestrial animal biodiversity is increasingly being lost because of land-use change[1,2]. However, functional and energetic consequences aboveground and belowground and across trophic levels in megadiverse tropical ecosystems remain largely unknown. To fill this gap, we assessed changes in energy fluxes across 'green' aboveground (canopy arthropods and birds) and 'brown' belowground (soil arthropods and earthworms) animal food webs in tropical rainforests and plantations in Sumatra, Indonesia. Our results showed that most of the energy in rainforests is channelled to the belowground animal food web. Oil palm and rubber plantations had similar or, in the case of rubber agroforest, higher total animal energy fluxes compared to rainforest but the key energetic nodes were distinctly different: in rainforest more than 90% of the total animal energy flux was channelled by arthropods in soil and canopy, whereas in plantations more than 50% of the energy was allocated to annelids (earthworms). Land-use change led to a consistent decline in multitrophic energy flux aboveground, whereas belowground food webs responded with reduced energy flux to higher trophic levels, down to −90%, and with shifts from slow (fungal) to fast (bacterial) energy channels and from faeces production towards consumption of soil organic matter. This coincides with previously reported soil carbon stock depletion[3]. Here we show that well-documented animal biodiversity declines with tropical land-use change[4–6] are associated with vast energetic and functional restructuring in food webs across aboveground and belowground ecosystem compartments.

Losses of biodiversity in terrestrial ecosystems have been documented across continents, biomes, clades and ecosystem compartments[1]. Tropical ecosystems are among the most threatened globally, with losses driven primarily by land-use change, such as the conversion towards commodity crops[2]. However, understanding of these transformations is hampered by the complexity and enormous biodiversity of tropical ecosystems. On first approximation, the spread of agricultural monocultures causes drastic declines in plant diversity in comparison to rainforests[4]. These effects cascade beyond basal trophic levels through food webs and also affect higher trophic-level invertebrate and vertebrate consumers[2,5,6]. Thus, to mechanistically understand the consequences of land-use changes for animal biodiversity and related functions, we need to know the resulting complex changes in food webs across multiple trophic levels and along different food chains.

Losses of animal diversity may be explained by reduced primary ecosystem productivity[7] and by changes in the structure of, and interactions in, consumer communities, as has been shown in studies on the impacts of invasive species, climate or other environmental changes[8,9]. Energy, as a common currency which sustains life[10], can impose limits on the total number of species in an ecosystem[7], whereas shifts in community structure can change energy pathways through ecological networks (energy flux), which is closely associated with the distribution of biodiversity across different trophic levels and ecosystem compartments[11]. For instance, under tropical land-use change, large declines in the number of species were correlated with a simultaneous reduction in total energy flux in litter invertebrate communities[12], demonstrating that biodiversity loss is associated with a loss in available energy. In soil, however, a similar decline in biodiversity was not associated with reduced total energy flux but with a redistribution of energy across the food web[8,12]. This indicates that biodiversity loss is associated with exclusion of specific functional groups, rebalancing the system energetically. Disentangling total available energy changes from shifts in its distribution may help us to determine appropriate measures for restoration of ecosystem functioning.

[1]Animal Ecology, University of Göttingen, Göttingen, Germany. [2]German Centre for Integrative Biodiversity Research (iDiv) Halle-Jena-Leipzig, Leipzig, Germany. [3]Insitute of Biology, University of Leipzig, Leipzig, Germany. [4]Agroecology, University of Göttingen, Göttingen, Germany. [5]Functional Agrobiodiversity, University of Göttingen, Göttingen, Germany. [6]Department of Plant Protection, IPB University, Bogor, Indonesia. [7]Botanical Garden of University of Bern, University of Bern, Bern, Switzerland. [8]Biodiversity, Macroecology & Biogeography, University of Göttingen, Göttingen, Germany. [9]Department of Soil Science, IPB University, Bogor, Indonesia. [10]Centre for Transdisciplinary and Sustainability Sciences, IPB University, Bogor, Indonesia. [11]Department of Zoology, University of Cambridge, Cambridge, UK. [12]Ecology of Tropical Agricultural Systems, University of Hohenheim, Stuttgart, Germany. [13]Centre of Biodiversity and Sustainable Land Use, University of Göttingen, Göttingen, Germany. ✉e-mail: potapov.msu@gmail.com

The distribution of biomass and energy fluxes in terrestrial ecosystems is largely structured in 'green' (aboveground) and 'brown' (belowground) food-web compartments, which jointly shape ecosystem functioning and stability[13]. Redirection of energy across aboveground and belowground compartments is of interest to agricultural management, including, for example, nutrient availability[14], yield[3,15], soil carbon storage[3] and pest control[16]. However, despite close linkages of these two compartments by means of common primary producers (plant shoots and roots) and mobile animals, including generalist predators[17], belowground and aboveground tropical food webs have been studied independently of each other and the distribution of energy across aboveground–belowground and invertebrate–vertebrate food webs has never been quantified. This non-integrated perspective hampers understanding of the consequences of conversion of rainforest into agricultural production systems on total animal energy flux and, accordingly, on animal biodiversity and ecosystem functioning.

Here, we quantified energy fluxes across earthworms, birds and arthropods in soil and canopies of tropical rainforests in Sumatra, Indonesia to describe the energetic structure of tropical animal food webs across aboveground and belowground ecosystem compartments. Our group selection represents most animal biomass in these systems (arthropods and earthworms)[18,19], including ecosystem engineers (earthworms and ants) and animals at different trophic levels—from detritivores, microbivores and herbivores (various arthropod groups) to top predators (for example, spiders and birds)—thus reliably reflecting the composition of the food web as a whole. We further assessed changes in the energy flux distribution after rainforest transformation into plantation systems, including jungle rubber (selectively logged rainforest with planted rubber trees), as well as rubber and oil palm monoculture plantations, to show how altered land use changes the trophic functioning of aboveground versus belowground food webs. Our main hypothesis was that there are different keystone animal groups which channel most of the energy in rainforest and plantations and that energy distribution changes with land use: (1) across strata more energy is allocated to aboveground food webs in plantations because plantation management commonly aims to maximize aboveground production; (2) across trophic levels less energy is channelled to higher trophic levels in plantations because monocultures cannot sustain abundant and diverse predator communities; and (3) across resources at the base of the food web living plants are more important, whereas leaf litter is less important in plantations because of lower predation pressure, monodominant plant species and a reduction in litterfall. Such energy re-allocation is associated with changes in animal trophic functions across aboveground and belowground ecosystem compartments, with functional consequences at the ecosystem level.

To test our hypotheses, we estimated abundance and biomass of canopy arthropods using insecticide fogging, of birds using audio recorders and point counts and of soil arthropods and earthworms using high-gradient heat extraction from soil cores across 32 sites representing rainforests and plantations[20]. We linked collected body mass and biomass data to literature data on traits and feeding preferences of taxa to define 62 trophic guilds across all animal groups and to reconstruct food-web topologies at each site. We further used steady-state food-web modelling, which assumes that energetic demands of each trophic guild (including metabolic rate, losses during food assimilation and consumption by higher trophic levels) are compensated by energy uptake from lower trophic levels. Metabolic rates of each guild per biomass unit were estimated from body masses using metabolic regressions and multiplied by the observed biomasses. Resulting energy fluxes were used as quantitative measures of the distribution of energy and consumption of different resources (living plants, litter, bacteria, fungi, soil organic matter and other animals) in aboveground and belowground food webs[11,12]. We validate our results with another independent survey at the same sites (except jungle rubber) 4 years after the main survey, to prove the generality of our findings.

## Aboveground and belowground rainforest food webs

We found that most of the energy in rainforests was channelled in belowground, rather than in aboveground, animal food webs. The total aboveground energy flux (sum of all energy fluxes to canopy arthropods and birds) was 21.6 ± 9.7 (1 s.d.) mW m$^{-2}$ with a total fresh animal biomass of 0.8 ± 0.6 g m$^{-2}$, whereas the total belowground energy flux (sum of all energy fluxes to litter and soil arthropods and earthworms) was 295.8 ± 125.5 mW m$^{-2}$ and the biomass was 9.5 ± 7.1 g m$^{-2}$ (Figs. 1 and 2). These figures question the existing research focus on aboveground tropical food webs and animal biomass[21]. This energetic dominance of soil over canopy animals in rainforest is unexpected because about 95% of the energy channelled belowground is assumed to be processed by microorganisms[22]. The soil biomass numbers generally resembled those reported previously for animals in rainforests[8,22] but for canopy arthropods they were slightly lower[6,21,23]. Because canopy fogging may result in potential undersampling (suggested numbers span from twofold[21] to sixfold[23]), we also ran a sensitivity analysis, assuming that canopy height affected the effectiveness of this method (Methods; Extended Data Fig. 2). This analysis suggested that the real energy flux aboveground (assuming uniform distribution of arthropods in canopies) could be 62.0 ± 24.5 mW m$^{-2}$ in the most-severe undersampling scenario but could still not explain the 14-fold aboveground–belowground difference in energy flux we recorded. The belowground energetic dominance could be related to plant production, animal metabolism and resource quality: (1) tropical trees allocate twice as much produced organic matter belowground, in the form of litter and root biomass, as they store aboveground[3,24]; (2) soil is inhabited by numerous small animals which have high metabolic rates per unit biomass[10] and together make up the biggest share of energy channelling across aboveground and belowground compartments; and (3) basal food resources belowground (litter and soil organic matter) are of poor palatability which results in a low assimilation efficiency. Thus, more resource consumption belowground than aboveground is needed to gain the same amount of energy[25]. This finding also indicates a perceived 'biomass/energy flux–diversity discrepancy' between aboveground and belowground tropical communities, with tropical canopies being extremely species-rich but having relatively low animal biomass and energy flux in comparison to soil and litter communities. However, very little is known about species diversity of arthropods in tropical soils[3,12,26], so it is possible that biodiversity levels are much higher in rainforest soils than is estimated at present.

## Rainforest canopy arthropods and birds

We found that arthropods dominated energetically over birds in rainforest canopies. Energy flux to canopy arthropods was 18.0 ± 9.7 mW m$^{-2}$, whereas birds contributed only 1.6 ± 1.9 mW m$^{-2}$ (Figs. 1 and 2). The bird biomass estimate (0.3 g m$^{-2}$) matches a previous detailed inventory in the neotropics[27], suggesting that our estimates are realistic. As we did not measure contributions by other vertebrate groups (for example, bats and amphibians), we cannot be certain about the relative contributions of vertebrates versus invertebrates based on our data. However, including more vertebrate groups would also increase invertebrate energy flux, as many of them feed on invertebrates, making it unlikely that this would compensate for the 12-fold difference in energy flux we detected. Overall, it is evident that rainforest food webs are energetically dominated by invertebrates and are largely 'brown'.

## Keystone groups across land uses

We found strong community shifts in plantations in comparison to rainforest, which supports our main hypothesis that different taxa play key energetic roles in different systems (Extended Data Fig. 6). These shifts were not associated with total animal energy flux decline

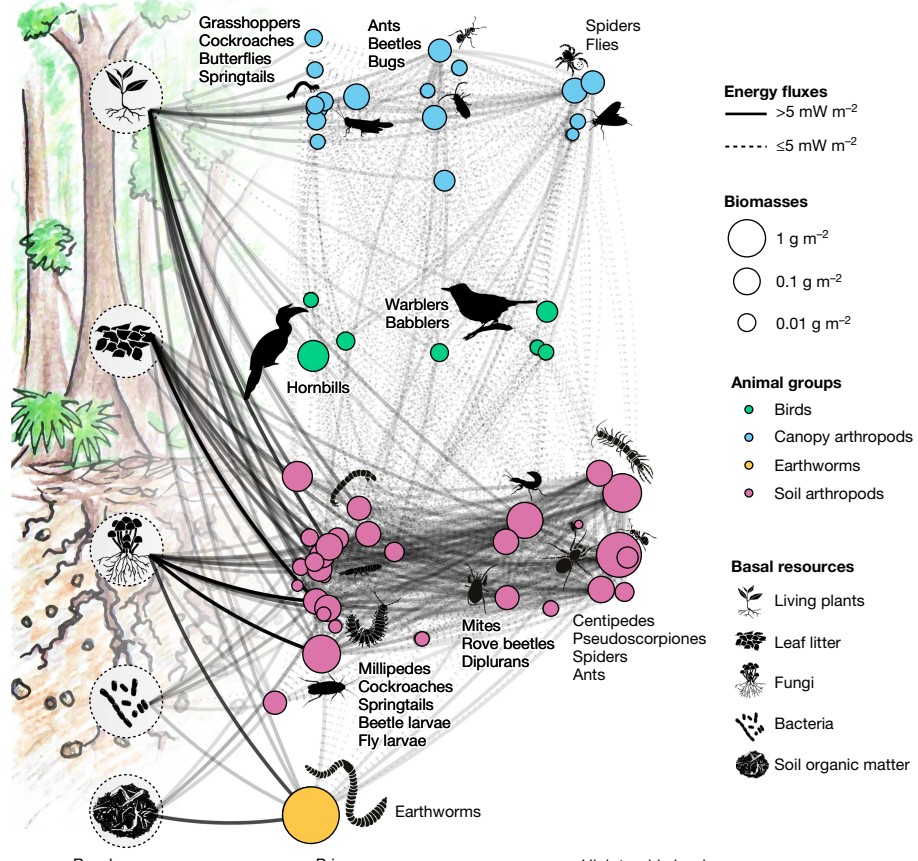

**Fig. 1 | Distribution of energy across birds, earthworms and arthropods in rainforest food webs across aboveground and belowground compartments.** Connecting lines on the food-web diagram represent average energy fluxes. Fluxes are classified into strong (solid lines) and weak (dotted lines), on the basis of an arbitrary threshold of 5 mW m$^{-2}$. The opacity of the lines scales with flux values. Food-web nodes include basal resources (displayed with black drawings/diagrams on the left) and consumer trophic guilds (coloured points), grouped into canopy arthropods (blue), birds (green), soil arthropods (pink) and earthworms (yellow). Sizes of consumer nodes are proportional to node fresh biomasses (square root scale). Nodes are ordered horizontally according to the trophic position (continuous variable; nodes were slightly jittered to avoid overlaps but the general order remains) and vertically according to the ecosystem stratification (positions within the four major animal groups/colours are random). Exemplary dominant taxonomic groups in the major trophic levels (primary consumers, omnivores and primary predators, top predators) are shown with text. The scheme summarizes data across all rainforest sites ($n = 8$). Illustrations of a plant seedling, litter, fungi, bacteria, soil organic matter, ant, spider, springtail, mite, diptera larvae, millipede, earthworm, centipede and bird were drawn by S. Meyer.

but mainly with its re-allocation. The total animal energy flux was similar in rainforest and monoculture plantations (310–317 mW m$^{-2}$) and was about 50% higher in jungle rubber, although the variation was very high (the total system effect was not significant; Fig. 2 and Extended Data Table 1). Differences were strongest in earthworms, which were responsible for an average of 13% of the energy flux per site in rainforest (29.4 ± 37.1 mW m$^{-2}$) but for 60–79% of the energy flux across plantations (group × system interaction $\chi^2_9 = 50.1$, $P < 0.0001$; Extended Data Table 1). The high energy flux in jungle rubber may be explained by intermediate disturbance of the ecosystem combined with favourable conditions for earthworms (for example, higher pH due to liming and ashes after burning[8]), which are able to exploit earlier accumulated soil organic matter as an extra resource and incorporate it into the food web (Fig. 1). The increase in the earthworm-associated energy flux was mirrored by a decline in the soil arthropod-associated energy flux (Fig. 1). It is known that earthworms may negatively affect soil and litter arthropods through direct (consumption of small fauna) and indirect trophic interactions and environmental modifications (litter removal and microbial feeding)[8,28], but the arthropod decline may also have been a result of reduced leaf litter input and reduced soil organic carbon and nitrogen in plantations[29]. Energetically important arthropod groups in rainforest included springtails (12%), beetles

(9%) and ants (7%; belowground food webs; Fig. 1), whereas in plantations they included springtails (3–5%), beetles (1–5%) and termites, symphylans, butterfly larvae, millipedes and dipterans, depending on specific ecosystem type (belowground food webs; Supplementary Table 1). These shifts illustrate different susceptibility of animal taxa to ecosystem transformation[30,31]. Tropical land-use change has been found to result in an 18–70% decline in species richness in arthropods, birds and other taxa[30–33]. Our findings show that this species decline is associated with fundamental changes in the energy distribution across food webs, rather than overall energy flux decline in converted tropical ecosystems.

## Aboveground-to-belowground shift with land use

Plantation management commonly aims to maximize yield and associated aboveground production. Therefore, it is likely that energy flux will be higher in aboveground compared to belowground food webs in plantation systems. In support of this, a previous study found that biomass of canopy arthropods declined less than that of soil arthropods after rainforest transformation to oil palm monoculture plantations[6]. Thus, we initially proposed that belowground energy flux (sum of all energy fluxes belowground) would be stronger in rainforests, whereas

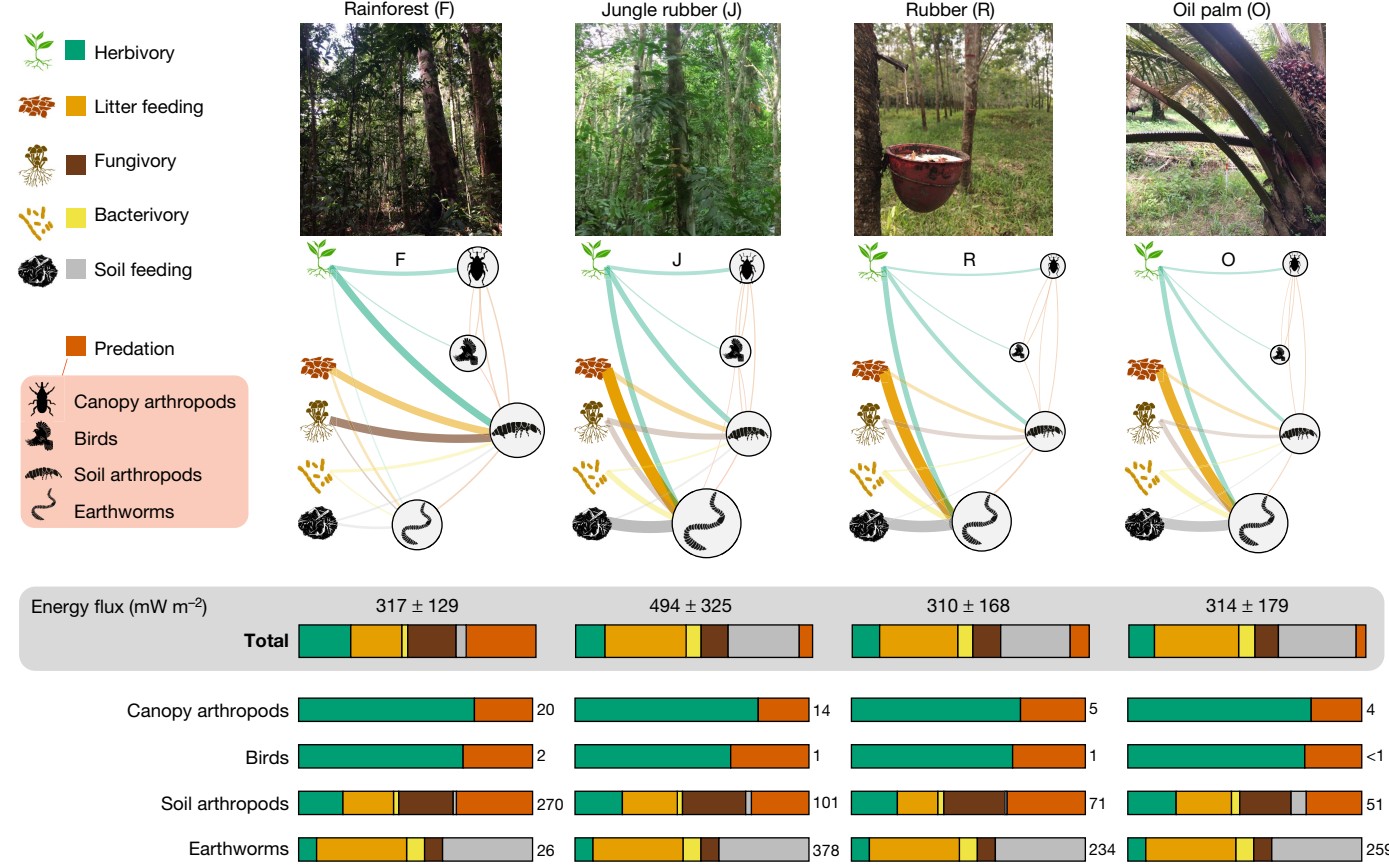

**Fig. 2 | Distribution of energy across aboveground and belowground compartments in rainforest and plantation food webs.** Food-web nodes include basal resources shown in different colours (living plants, green; plant litter, orange; fungi, brown; bacteria, yellow; soil organic matter, grey) and consumers merged into four major groups according to their ecological niches (canopy arthropods, birds, soil arthropods and earthworms). Sizes of consumer nodes are proportional to node biomasses. Connecting lines on the food-web diagram represent average energy fluxes, quantified in mW m⁻² (represented by line thickness). Colours of energy fluxes reflect colours of the donor nodes and represent associated 'trophic functions': herbivory (on leaves or roots), litter transformation, bacterivory, fungivory, soil transformation and predation. Average trophic functions for each major group of consumers and for the food web in total are summarized as stacked proportional bar charts (*n* = 8 sites per system). Estimated mean energy fluxes are shown with numbers to the right of the bars; total energy flux (sum of all fluxes) is given as mean ± 1 s.d. Illustrations of a plant seedling, litter, fungi, bacteria, soil organic matter, springtail, earthworm and bird were drawn by S. Meyer.

aboveground energy flux (sum of all energy fluxes aboveground) would be stronger in plantations. However, contrary to our hypothesis, rainforest transformation resulted in a relative increase in belowground compared to aboveground fluxes. The belowground energy flux was higher than the aboveground in rainforest (about 14-fold) and this difference increased in jungle rubber (about 30-fold), rubber (55-fold) and oil palm monocultures (68-fold), with an even higher difference in biomass (Fig. 3a,b; significant system:compartment interactions). This change in the ratios resulted from reduction of the total aboveground energy flux by −75% to −79% in both monoculture plantation types in comparison to rainforest (up to −92% considering potential undersampling of canopy arthropods; Extended Data Fig. 2), whereas belowground energy flux changed little. This change may be because of a delayed impact of land-use change on belowground compared to aboveground biodiversity, which could be explained by legacy effects due to the high inertia of soils[34], for example, exploitation of earlier accumulated soil organic matter. The differing energetic responses of aboveground and belowground systems to land-use change in tropical landscapes echo the recently demonstrated differences in aboveground and belowground biodiversity responses observed in temperate grasslands[35]. This implies that such diverging responses might be universal, fitting the 'green–brown imbalance' hypothesis, which suggests a higher resistance of belowground than aboveground food webs

owing to a lower number of specialized links in the former[13] (because of restricted mobility of organisms and thus a more opportunistic food selection). At present, belowground processes in plantations seem to be stabilized by earthworms which energetically compensate for losses in arthropod communities[36]. However, earthworms in plantations are mainly represented by invasive species[37] and their dominance reduces the entire food web to a detritus–microbe–animal or detritus–animal scheme. The number of trophic interactions in both aboveground and belowground webs in plantation systems decreased by 13% to 37%, reflecting reduced biodiversity aboveground and belowground (Fig. 3c). Therefore, soil animal communities in plantations rely on fewer interactions (on average −21%), reflecting documented losses of biodiversity and multifunctionality[8,12,30,38] but nevertheless process a similar amount of energy as soil animal communities in rainforests. This demonstrates a remarkable adaptability of belowground food-web functioning to perturbations[35].

## Predation decline in plantations

It has been suggested that diverse plant communities avoid resource concentrations and promote nutrient heterogeneity, which prevent (specialized) herbivores from being very abundant; at the same time, diverse plant communities provide greater refuge and resources for

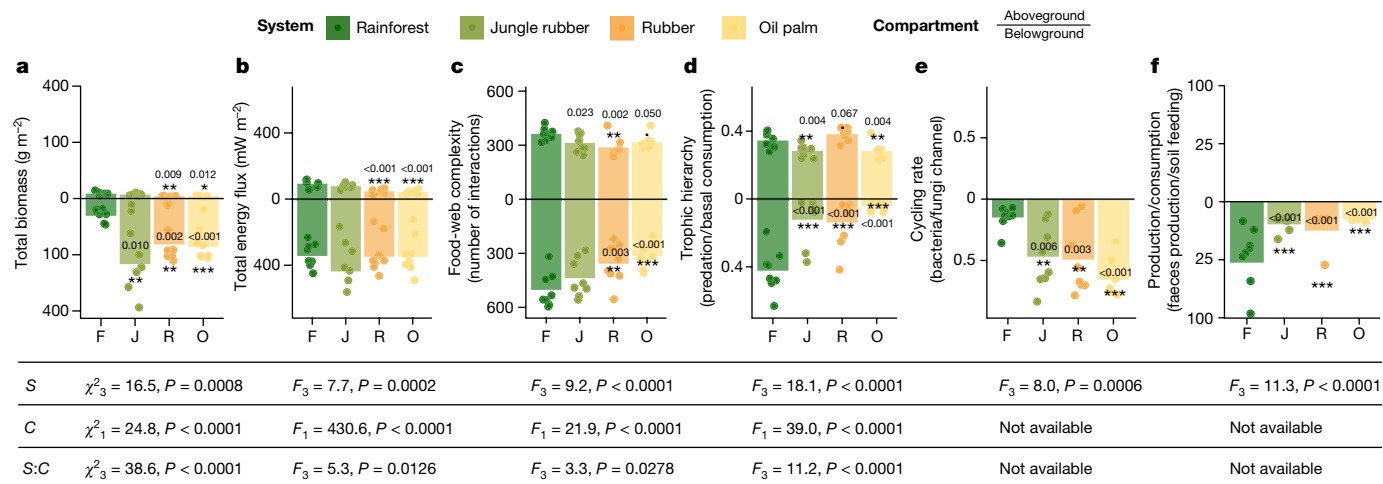

**Fig. 3 | Aboveground and belowground food-web indicators in rainforest and plantations. a–d,** Bulk indicators were calculated separately for aboveground (canopy arthropods and birds, above the zero line) and belowground food webs (soil arthropods and earthworms, below the zero line) for total biomass (**a**), total energy flux (**b**), food-web complexity (**c**) and trophic hierarchy (**d**). Trophic hierarchy was calculated as the ratio of all 'predatory' energy fluxes to all 'basal resource consumption' energy fluxes. **e,** Carbon cycling rate was calculated as the ratio of all outgoing fluxes from bacteria to all outgoing fluxes from fungi. **f,** Carbon balance was calculated as the ratio of all produced faeces (unassimilated food) to all outgoing fluxes from soil organic matter. Each point is a site, bars represent means ($n = 8$ sites per system). Colours denote land-use systems (dark green, rainforest; light green, jungle rubber; orange, rubber; yellow, oil palm). Units for each parameter are given in brackets; note square root scale in **a**, **b** and **f**. Asterisks mark significant differences of mean values for the given parameter aboveground or belowground from that in rainforest (generalized linear mixed-effects models; two-tailed ***$P < 0.001$, **$P < 0.01$, *$P < 0.05$). Effects of land-use system (*S*) and aboveground/belowground ecosystem compartment (*C*) and their interaction (*S:C*) on the tested parameters are given below the corresponding bar charts. F, rainforest; J, jungle rubber; R, rubber; O, oil palm.

(generalist) predators than do monocultures, which jointly sustain higher predation-to-herbivory rates[16,39]. Indeed, previous studies have shown that proportionally less energy flows to predators in soil and litter food webs in plantations than in rainforests[8,12]. Thus, we also suggested that predation to primary consumption rates would be lower across aboveground and belowground food webs in plantations than in rainforest. In agreement with this, the predation/consumption ratio declined by 18% aboveground and by up to 90% belowground with rainforest transformation to jungle rubber and oil palm. However, in monoculture rubber plantations the proportion of predation in canopy and soil arthropods (but not in birds) was similar or even slightly higher than that in rainforest (increase of 11% aboveground; Fig. 3d). High predation in rubber canopies might be associated with a simple canopy structure[40] but this does not explain low predation in oil palm. Because the high predation in rubber canopies was mainly associated with a large biomass of blood-sucking gnats and mosquitoes, it may be explained by the presence of small water bodies (rubber sap collection buckets) in rubber plantations which can host aquatic dipteran larvae. The different effects of oil palm and rubber cultivation on relative predation suggest that tropical land-use choices can have a predictable impact on specific food-web functions. Our results illustrate that decline in predation is a common trend across aboveground and belowground compartments and taxa with agricultural transformation[8,12]. Agroecosystems often have a weaker natural control of pests in comparison to more natural ecosystems[41], which may partly explain pest outbreaks in plantation systems such as oil palm[42]. Reduced natural pest control in oil palm is also supported by a lower predation-to-herbivory ratio ($0.37 \pm 0.16$ in birds, $0.28 \pm 0.05$ in canopy arthropods and $1.14 \pm 0.63$ in soil arthropods) in comparison to rainforest ($0.64 \pm 0.29$ in birds, $0.34 \pm 0.05$ in canopy arthropods and $1.95 \pm 0.74$ in soil arthropods).

## Changes in belowground carbon cycling

We classified non-predatory energy fluxes according to five major basal resource classes, corresponding to the 'trophic functions' of herbivory, litter feeding, fungivory, bacterivory and soil feeding (Fig. 2)[43]. We

proposed that the dominant trophic functions would change with land use, indicating different carbon pathways at the ecosystem scale—specifically, we expected proportionally higher use of primary basal food resources, especially living plants, in plantations, resulting from a decrease in alternative resources, such as microbial biomass and leaf litter[44]. We found that land-use change to plantations consistently altered energy distribution at the base of food webs by reducing total herbivory and fungivory, while increasing bacterivory and soil feeding (function × system interaction $\chi^2_{15} = 111.1$, $P < 0.0001$; Fig. 2 and Extended Data Table 1). We recorded a 3.2- to 4.4-fold increase in bacteria/fungi energy flux ratio across plantation systems (Fig. 3e). This increase was explained mostly by the high abundance of earthworms in plantations, which can effectively assimilate bacterial carbon from old soil organic matter[45]. However, an almost twofold increase in bacteria/fungi energy flux ratio was also observed in soil arthropods in oil palm monocultures (Fig. 2). These results are in line with previous studies showing that disturbance associated with agriculture and high fertilization rates may change the balance from slow (for example, fungal) to fast (for example, bacterial) energy fluxes in soil food webs[15,46]. At the same time, these results are in contrast to the existing evidence of higher bacteria consumption by soil animal communities in rainforests, as indicated by bacteria-specific fatty acid biomarkers[47]. However, the same study reported an increase in non-specific bacterial biomarkers[47]. The likely increase in bacterivory therefore indicates that there is accelerated energy processing (faster turnover rates) in these systems. A shift from the naturally observed balance to food webs dominated by fast energy channelling may make the system more susceptible to perturbations (resulting from an increase in strong interactions[48]) and may accelerate depletion of carbon stocks[15]; the latter has been observed in rubber and oil palm plantations[3]. This depletion is associated with high soil feeding by earthworms, which can effectively use old soil carbon resources[49]. However, the net effect of earthworm feeding activity on carbon sequestration and emission remains a controversial topic in soil ecology[50,51]. To quantify animal effects on soil carbon stocks, we here calculated the ratio between the production of faeces (unassimilated food) and the consumption of soil organic matter by all soil

invertebrates. It has been shown that conversion of plant materials into faeces by soil invertebrates increases microbial biomass production[52], which is the key process contributing to soil organic matter formation and stabilization[53]. In turn, invertebrates are able to mobilize and recycle this stored carbon while feeding on bulk soil. Supporting the link between the belowground food-web structure and net carbon loss in plantations, we found that the production-to-consumption ratio decreased by more than 75% from 27.6 ± 29.6 in rainforest to 3.8 ± 2.9 in jungle rubber, 6.2 ± 10.4 in rubber and 2.3 ± 0.3 in oil palm plantations (Fig. 3f). Overall, our analysis suggests that changes in energy flux distribution due to habitat transformation have large functional consequences for carbon cycling. However, the exact mechanisms involved and quantification of these animal effects over time requires dynamic ecosystem-level modelling and targeted experiments.

## Methodological caveats

There are few empirical studies on tropical invertebrate food webs and food-web analysis can be sensitive to assignment of trophic guilds and interactions[54]. Here, we based our reconstruction on a recent review[55] and empirical data collected from our study sites[36], which make our food webs as close to reality as possible at the current state of knowledge. Sensitivity tests of our food-web reconstruction model revealed feeding specialization/omnivory as the main characteristic affecting absolute estimates of belowground-to-aboveground energy balance but none of the possible coefficients affected our conclusions (Extended Data Fig. 1). Our aboveground energy flux estimates could also be biased because we did not sample all vertebrate animal groups. Amphibians, reptiles, bats and other mammals are important invertebrate predators in tropical rainforests. However, as discussed above, this is unlikely to change our conclusions which are based on more than tenfold differences in energy fluxes, with the same applying to the potential undersampling of canopy invertebrates (Extended Data Fig. 2). Finally, our plantation systems were 14–18 years old and were unlikely to be at a stable state, especially considering higher rates of change in the aboveground than in the belowground ecosystem compartments. We therefore call for studies evaluating tropical land-use systems in the longer term. To prove the generality of our findings, we performed another survey at the same sites (except jungle rubber) in 2016–2017. This validation survey showed lower estimates of the absolute biomass and energy flux but validated energetic dominance of the belowground over the aboveground energy flux, canopy arthropods over birds, energetic decline in canopies, re-allocation of energy to belowground food webs in plantations and shifts in trophic functions, such as an increase in bacteria-to-fungi and a decrease in faeces production-to-soil consumption ratios. However, it did not validate the general loss of trophic links across aboveground and belowground compartments (Extended Data Figs. 5, 6 and 7; Supplementary Notes). Potentially, some trophic links were restored as plantation aged (from about 15 years old in the main survey to about 19 years old in the validation survey) but future plantation replanting (normally done at 25 years) will probably result in a second wave of biodiversity decline[56], which may lead to further food-web disassembly. Overall, it is clear that our assumptions and approaches do not affect our main conclusions.

## Conclusions

Our study provides an energetic description of tropical rainforest and plantation food webs across aboveground and belowground compartments, demonstrating generalities of land-use effects previously observed only in temperate ecosystems. In addition, we report new and nuanced patterns of food-web responses depending on specific land uses and ecosystem compartments. Overall, we conclude that (1) rainforest animal communities are energetically dominated by arthropods in belowground food webs; (2) animal communities in tropical canopies suffer higher total energetic losses due to rainforest transformation than those in belowground food webs but the energy in belowground food webs in plantations is reallocated from functionally diverse arthropod communities to invasive earthworms[8]; (3) land-use change is associated with a decline in predation and an increase in relative herbivory both aboveground and belowground in jungle rubber and oil palm, however, the high predation in rubber suggests that crop choices can have predictable outcomes for trophic functions in food webs; and (4) belowground food webs in plantations rely on different basal resources than those in rainforest, promoting faster energy channelling and shifting carbon balance from production of faeces to consumption of soil organic matter. These changes are associated with previously observed depletion of carbon stocks[3] but the mechanisms driving animal effects in this context remain to be tested experimentally.

It is well documented that tropical land-use change results in animal biodiversity losses both aboveground and belowground[30,31]. We show here that biodiversity losses are associated with changes in food-web structure, consumption of different pools of organic matter and energy fluxes and these changes are distinctly different between the aboveground and belowground realm. We suggest that restoration and management practices in the tropics which alter the energetic balance across ecosystem compartments, taxa, size classes and trophic levels, need to be more closely considered and trialled. Plantations, especially oil palm, are very productive[3] but the available energy for maintaining multitrophic biodiversity is disproportionately low, which is associated with re-allocation of energy fluxes to basal trophic levels in belowground food webs. The high total energy flux indicates that energy is not a limiting factor for animal biodiversity in plantations and restoration measures should focus on other ecosystem aspects. Improving belowground habitat structure through mulching[38,57] and reducing herbicide use[58] could be sufficient to partly restore soil biodiversity and energetic balance in belowground food webs. However, it may take time for the effects of these measures to become visible as a result of high historical inertia of the soil system. Aboveground, measures directly affecting vegetation are needed. For example, increasing canopy complexity by planting trees in monoculture plantations[59,60] and designing diverse landscapes[30] could provide more ecological niches, probably resulting in re-allocation of more energy to aboveground food webs. In the absence of restoration measures, intensive tropical land use may foster earthworm invasion belowground, further depletion of soil organic stocks and increase risks of aboveground pest outbreaks. This is likely to result in intensification of fertilizer, herbicide and pesticide use. Experimental studies exploring the effect of restoration measures on the energy distribution and trophic functions of food webs across aboveground and belowground compartments of tropical ecosystems will be crucial for better management of the energy of tropical ecosystems, to sustain tropical biodiversity and ecosystem services.

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

## Methods

### Study region and design

The study was carried out in Jambi Province, Sumatra, Indonesia in the framework of the Collaborative Research I 990 'EFForTS'[20]. Over the last few decades, lowlands in this region have experienced drastic land-use change from rainforests to smallholder-dominated cash crop agriculture of mainly rubber (*Hevea brasiliensis*) and oil palm plantations (*Elaeis guineensis*)[30]. We studied four common land-use systems: primary but slightly degraded lowland rainforest[61], jungle rubber, rubber monocultures and oil palm monocultures. Forest plots were located in the Bukit Duabelas National Park and the Harapan Rainforest Restoration concession (PT REKI) and had a 90th percentile tree height of 29.5 m and tree density of 556 trees ha⁻¹ (diameter at breast height ≥ 10 cm). Jungle rubber plots represented an extensively managed agroforest system, which is established by planting rubber trees into secondary or disturbed forest and had a 90th percentile tree height of 21.0 m and tree density of 580 trees ha⁻¹. Rubber and oil palm monocultures represented smallholder plantations, often with intensive management (fertilizers and herbicides) and a 90th percentile tree height of 17.0 and 12.7 m and tree density of 467 and 138 trees ha⁻¹, respectively. At the time of the main survey (May–November 2013), the age of all monoculture plantations was between 8 and 17 years. In total, 32 sampling sites were surveyed in an area of about 80 × 80 km spanning two regions (with loamy Acrisol and clayey Acrisol soils)[62]; each land-use system was replicated eight times, four in each of the two regions[20,30]. Statistical methods were not used to predetermine the sample size, no blinding and randomization were used. Each plot measured 50 × 50 m and had five permanent 5 × 5 m subplots. More information is provided in the introductory EFForTS paper[20]. In each of the 32 plots, we applied a combination of collection methods to assess bird, canopy arthropod, soil arthropod and earthworm communities. Our assessment is a snapshot which cannot represent all animal species at the study sites. However, the functional composition of communities is typically more stable than the species composition[38]; that is, despite species turnover, different species will perform similar roles in the food web. This turnover, however, is expected to be moderate because of a limited seasonality at the study region, with a rainier period during December–March and a dryer period during July–August[20]. Although we were not able to fully cover the spatial heterogeneity in each plot, our sampling design compensates for this with true replication of $n = 8$ plots per system. To account for the temporal variation, validate results of the main survey and prove the generality of our findings, we did another independent survey with the same approach at the same sites (except jungle rubber; that is, 24 plots) in 2016–2017. Data from both surveys were processed in the same way to reconstruct food webs across aboveground and belowground compartments[43]. To do this, we estimated densities and biomasses of taxa, classified trophic guilds and assigned body masses, habitat preferences and feeding preferences to each guild[43]. Feeding preferences at the base of the food web were assigned to the five major basal resource classes[43]: living plants (leaves and roots), leaf litter, fungi, bacteria and soil organic matter (dead organic matter mainly associated with the mineral soil fraction). The biomass of basal resources was not used in the food-web modelling because we focussed on consumption/energy flux[63].

### Birds

Birds were sampled with point counts as well as automated sound recordings from May to July 2013. All plots were visited three times for 20 min point counts. The observer stood in the plot middle and all birds detected in the plot were recorded. Point counts took place between 6:00 and 10:00 and the timing for individual plots alternated between early and late morning[32]. We excluded detections from fly-overs and bird vocalizations that could not be identified immediately were recorded using a directional microphone (Sennheiser ME-66)

to compare with recordings from the xeno-canto online bird call database (http://xeno-canto.org/). In addition to point counts, we recorded stereo sound at 44,100 Hz sampling frequency (SMX-II microphones, SM2+ recorder, Wildlife acoustics); the recorders were attached to the central tree of the plot at 2.0–2.5 m height. We recorded sound in eight plots simultaneously; sampling all 32 plots took 4 days (10 and 13 May and the 3 and 7 June 2013). We uploaded the first 20 min of recording after sunrise to the online eco-acoustics platform BioSounds[64] so that two independent ornithologists could identify all audible bird calls and calls visible on the spectrogram (within an estimated 35 m radius) to species. For each plot, bird species identified by both ornithologists in the recordings were subsequently merged with the species obtained from the point counts to generate the dataset used in the analysis. In total, 418 bird occurrences were detected in 2013 and 542 in 2016 (validation survey). Guilds were defined on the basis of feeding preferences of species (five levels: fruits and nectar, plants and seeds, invertebrates, vertebrates and scavenging, omnivores), spatial distribution (canopy, ground foraging or both) and body masses; following information obtained from a public database[65]. In total, 11 guilds were distinguished (raw data are available from figshare; Data availability).

### Canopy arthropods

Canopy arthropods were collected by fogging (the application of a knockdown insecticide) in three locations per plot between May and October 2013 (main survey) and 2017 (validation survey). Target locations were randomly positioned in the plot; fallen trees and canopy gaps were avoided. Fogging was conducted immediately after sunrise, in dry conditions to avoid small arthropods sticking to precipitation. A mixture of 50 ml of DECIS 25 EC (Bayer Crop Science, deltamethrine 25 g l⁻¹) and 4 l of petroleum white oil was applied to each target canopy, about 20 min per fogging event. Underneath each target canopy, square 1 × 1 m funnel traps were placed at about 1.5 m above ground level using ropes and each funnel was fitted with a 250 ml plastic bottle containing 100 ml of 96% ethanol. Sixteen funnels were used during the main survey in 2013, whereas eight funnels were used for the validation survey in 2017. Two hours after the application of the insecticide, stunned or dead arthropods were collected and cleaned from debris, the ethanol was exchanged and the samples were stored at −20 °C until further analysis. The data used in this study were based on combined abundances of canopy arthropods across the three subsamples per plot, resulting in one abundance value per plot. More details on the sampling are provided elsewhere[66]. Overall, 366,975 individual canopy arthropods were collected during the main survey and 179,334 during the validation survey. Arthropods were then sorted to 12 major arthropod orders (Acarina, Araneae, Blattodea, Coleoptera, Collembola, Diptera, Hemiptera, Hymenoptera, Lepidoptera, Orthoptera, Psocoptera and Thysanoptera). As large flying taxa such as Apoidea and Vespoidea in part actively evaded the insecticide fog at the time of application (J.D., personal observation), the order Hymenoptera in this study is represented by Formicidae (ants) and Braconidae (a family of parasitoid wasps), both of which were highly abundant in the samples[66,67]. Also, four abundant beetle families with contrasting feeding strategies were analysed separately from the rest of the order Coleoptera (henceforth termed 'other Coleoptera')—Chrysomelidae, Curculionidae, Elateridae and Staphylinidae. Arthropod taxa listed above were used as trophic guilds (17 in total), each assigned with feeding preferences to living plants or other invertebrates and vertebrates according to existing literature[55] and unpublished data on stable isotope composition measured in the collected animals. We extrapolated general knowledge on the trophic ecology of high-rank taxa (for example, Chrysomelidae are herbivores whereas Staphylinidae are predators) to all collected individuals in these taxa assuming phylogenetic signal in trophic niches and because information on the feeding preferences of most tropical invertebrate species is lacking. Average body mass of each guild at each plot was estimated using group-specific

length–mass regressions[63]; body lengths were measured for all animal groups in each sample (up to ten random individuals per sample per group to estimate the mean). Density of canopy arthropods per square metre was calculated by dividing the total abundance of collected arthropods by the number of traps used. Detailed biodiversity declines over the investigated land-use change gradient are published for arboreal communities of ants[67], beetles[68], springtails[69], spiders[33] and parasitoid wasps[66].

## Soil arthropods and earthworms

Soil invertebrates were collected using a high-gradient extraction method. In each plot, three soil samples were taken (one in each of three subplots) during October and November 2013. Samples measured 16 × 16 cm and comprised the litter layer and the underlying mineral soil layer to a depth of 5 cm. Litter and soil were extracted separately but merged in the food-web analysis. Animals were extracted from litter and soil for 6–8 days under a heat gradient from 40 to 50 °C above the sample to 15 °C below the sample and collected in dimethyleneglycol:water solution (1:1) and thereafter transferred to 70% ethanol. More details on the sampling and extraction procedure are given elsewhere[70]. In total, 29,956 soil invertebrate individuals were collected in 2013 and 50,401 individuals in 2016 (validation survey). The lower total number of collected individuals in 2013 is because mites and springtails were counted in only two out of three samples per plot. Collected animals, including earthworms were sorted to high-rank taxa (orders and families) under a dissecting microscope, allowing allocation to trophic guilds[55]. Soil invertebrate taxa are generally consistent in their trophic niches[71]. However, to reflect widespread omnivory, most of them were assigned to feed on multiple basal resources (living plants, litter, bacteria, fungi, soil organic matter and other invertebrates) on the basis of existing knowledge[55] and stable isotope composition previously measured in the collected animals[36]. Average body mass of each guild at each plot was estimated using group-specific length–mass regressions[63,72–74], with body lengths measured from all individuals in each sample (for ants and symphylans we measured only the first ten individuals per sample). Vertical distribution across soil, litter and ground for each trophic guild of soil arthropods was estimated using the relative abundance of this guild in litter (litter and ground layers) or soil[10]. In total, 33 guilds of soil arthropods and one guild of earthworms were distinguished (Extended Data Fig. 6). Density of soil invertebrates per square metre was calculated by recalculating the abundance from the sample to the metre scale.

## Food-web reconstruction

All data manipulations and statistical analyses were done in R v.4.2.0 with R studio interface v.1.4.1103 (RStudio, PBC). We used a 'multichannel' food-web reconstruction approach[43]. We combined all trophic guilds across birds, canopy and soil arthropods and earthworms into a single table which included the following traits of each guild: feeding preferences to plants (including phototrophic microorganisms and endophytic/epiphytic microorganisms aboveground), litter, fungi, bacteria, soil organic matter or animal food (predation on invertebrates or vertebrates), mean body mass, body mass variation (standard deviation), biomass per square metre and spatial niche (soil, litter, ground and canopy). The table was complemented with published information on protection traits and C and N content for each guild[55] (full data with all traits are available from figshare; Data availability). Because species-level biology of tropical invertebrates is poorly known and we did not have species-level information for about 50% of the studied arthropods, traits were assigned to supraspecific taxa assuming their general trophic and functional consistency[71]. Generic rules of food-web reconstruction based on food-web theory were used to infer weighted trophic interactions among all nodes with the following assumptions[43]: (1) there are phylogenetically inherited differences in feeding preferences for various basal resources and predation capability among

soil animal taxa which define their feeding interactions (reflected as resource preferences in the raw data table)[55]; (2) predator–prey interactions are primarily defined by the optimum predator–prey mass ratio (PPMR)[75,76]—typically, a predator is larger than its prey but certain predator traits (hunting traits and behaviour, parasitic lifestyle) can considerably modify the optimum PPMR[43]. We measured body mass distribution overlap for each potential pair of predator and prey in each food web to determine the most plausible trophic interactions; (3) strength of the trophic interaction between predator and prey is defined by the overlap in their spatial niches related to vertical differentiation, with greater overlap leading to stronger interactions (no overlap among specialized canopy and soil arthropods and full overlap between 'canopy' birds and arthropods collected using canopy fogging); (4) predation is biomass-dependent[77]—because of higher encounter rate, predators will preferentially feed on prey that are locally abundant; and (5) strength of the trophic interaction between predator and prey can be considerably reduced by prey protective traits—prey with physical, chemical or behavioural protection are consumed less[78]. All these assumptions are applied together to infer the most plausible trophic interaction matrix. For example, feeding preferences of omnivorous nodes to basal resources or other invertebrates were assigned on the basis of literature (assumption 1), whereas prey selection among other invertebrates was based on size, spatial niche, total biomass and protection of prey (assumptions 2–5). The reconstruction R script is available from figshare (Code availability). Food-web reconstruction was carried out separately for each plot; collected data were averaged across subplots. Plots were assumed to represent local food webs and were used as biological replicates in statistical analyses (Extended Data Figs. 3 and 4).

## Energy flux estimation

To calculate energy fluxes among food-web nodes we used reconstructed interaction networks, biomasses, body mass-dependent metabolic losses and environmental temperature and applied the fluxweb package[77]. In brief, per-biomass metabolic rates were calculated from average fresh body masses using the equation and coefficients for corresponding phylogenetic groups of invertebrates[13,79] and endothermic vertebrates[12] (the used metabolic regressions typically have $R^2 > 95\%$ if calculated for a wide range of body masses; Extended Data Table 2). The mean annual soil temperature was taken from meteorological measurements at our study sites (forest 25.0 °C, jungle rubber 25.6 °C, rubber and oil palm 26.1 °C)[80]. The energy flux to each node was calculated from per-biomass metabolism, accounting for assimilation efficiencies (proportion of energy from food that is metabolized by the consumer) and losses to predation assuming a steady-state energetic system (energetic losses from each node are compensated by the lower trophic levels; for example, if herbivores are present in the system there is enough plant biomass to sustain them)[13,81]. Although the steady-state assumption is unlikely to be fully supported in most real-world ecosystems, this assumption allows for comparison of dominant energy processes across different ecosystems that are stable at the time of consideration (years) and thus was appropriate for our aims. We used diet-specific assimilation efficiencies which we calculated from nitrogen content of each prey/basal resource node using a published equation[25]. Assimilation efficiencies for basal resources were calculated as 21% from plant material, 18% from leaf litter, 13% from soil organic matter, 96% from bacteria and 36% from fungi and from 61% in millipedes to 97% in centipedes, earthworms and several other animal groups[43]. Then, we applied the fluxing function to the reconstructed interaction networks, which delivered energy flux estimations among all food-web nodes. Data were expressed in mW m$^{-2}$. Because the absolute estimates of the energy flux can be biased as a result of the abovementioned assumptions and regression-based conversions, we focus mainly on comparisons in our main conclusions. Detailed information on the approach can be found in the energy flux methodology paper ref. 63.

## Food-web parameters

To analyse food-web structure and energetics and test our hypotheses, we calculated bulk parameters for each food web and classified energy fluxes according to 'trophic functions'. Trophic functions were primarily linked to the consumed food: herbivory represented a sum of outgoing fluxes from living plants (leaves/shoots and roots); litter and soil feeding represented a sum of outgoing fluxes from plant litter and soil organic matter, respectively; bacterivory and fungivory represented a sum of outgoing fluxes from bacteria and fungi, respectively; predation represented a sum of outgoing fluxes from all animal nodes. Six bulk parameters were calculated: (1) total biomass of all studied animal groups per square metre; (2) total energy flux (in mW) across all studied animal groups per square metre; (3) number of trophic links among trophic guilds in the reconstructed food web (a proxy for food-web complexity); (4) ratio of all energy fluxes from prey to predators to all energy fluxes from basal resources to primary consumers (a proxy for trophic hierarchy and predation control[18]); (5) ratio of all energy fluxes from bacteria versus all energy fluxes from fungi (a proxy for carbon cycling rate[48]); and (6) ratio between the production of faeces and the consumption of soil organic matter (a proxy for soil organic matter/carbon balance). The last indicator is new and is based on three main lines of evidence: (i) conversion of plant material into faeces by soil invertebrates increases microbial biomass production[52]; (ii) microbial biomass production is the key process contributing to soil organic matter formation and stabilization[50]; (iii) consumption of soil organic matter by invertebrates (a sum of outgoing fluxes from soil organic matter) leads to consumption of associated microbial biomass[50] and thus has opposite effects to the first two lines of evidence. To calculate the production of faeces, we multiplied all energy fluxes by inverted assimilation efficiency and summed them up, thus quantifying all unassimilated food in the food web. We highlight that this parameter is new and should be validated through controlled experiments, as the effect of soil feeders on soil organic matter sequestration is context-dependent (although often negative as predicted)[50]. All parameters, except (5) and (6), and all trophic functions were calculated for the entire food web, separately for aboveground and belowground food-web compartments and for individual animal groups (birds, canopy arthropods, soil arthropods and earthworms).

## Statistical analyses

To analyse the overall distribution of energy flux across animal groups and trophic functions, we first ran two mixed-effect models testing the effect of land-use system (rainforest, jungle rubber, rubber and oil palm), region (two regions included in the design) and either major animal group or trophic function on energy fluxes in food webs (the lme4 package)[82]. Two models were run separately for groups and functions because not all functions are performed by all groups. Chi-square, significance and degrees of freedom were approximated using Wald Chi-square tests (the car package)[83]. We allowed for random intercepts depending on the plot to account for interdependence of groups and functions in the same site. The model code was lmer(Flux ~ Group (or Function) * Landuse + Region + (1 | Plot), data). To test specific hypotheses related to changes in trophic functions (first, more energy allocated to aboveground food webs in plantations; second, lower predation in plantations; third, a shift in basal resource feeding and carbon cycling across land-use systems), generalized linear models were run for each of the four bulk food-web parameters calculated separately for aboveground and belowground food-web compartments (response variables: total biomass, total energy flux, number of trophic links and trophic hierarchy) and two indicators of carbon cycling in belowground food webs (response variables: bacteria-to-fungi ratio and faeces production-to-soil consumption ratio). Data distribution selection followed visual inspection of the frequency distributions of raw data and

homogeneity in the residuals of the model. Gaussian distribution was used for the number of trophic links, bacteria-to-fungi ratio, trophic hierarchy and production-to-soil consumption ratio and log-normal distribution was used for the total energy flux. The model code was lm(Flux ~ Landuse * Above/belowground + Region, data). Owing to a strong heteroscedasticity of variance across aboveground and belowground compartments, we used generalized least-squares models to analyse the total biomass[84] (the nlme package)[85]. The model code was gls(Flux ~ Landuse * Above/belowground + Region, weights = vf, data), where vf <- varIdent(form = ~1 | Above/belowground * Landuse). To test for significant differences between rainforest and other land-use systems, we applied the same types of models for aboveground and belowground food-web compartments separately, testing the effect of the land-use system and reported P values of the linear model coefficients.

## Sensitivity analyses

We ran two further analyses to evaluate sensitivity of our conclusions to food-web reconstruction assumptions and undersampling of canopy arthropods. To test if the revealed patterns are robust to our food-web reconstruction assumptions (see section on Food-web reconstruction), we re-ran food-web reconstructions and energy flux calculations 50 times, varying the following parameters from 0 to 1 with the step of 0.1 (Extended Data Fig. 1): (1) omnivory, where 0 is full resource specialization and 1 is full trophic generalism; (2) self-predation, where 0 is no self-predation and 1 is no limits on self-predation; (3) size-structured predation, where 0 is strictly size-structured predation and 1 is trophic interactions independent of body masses; (4) spatial-structured predation, where 0 is predation strictly defined by spatial niche overlaps and 1 is trophic interactions independent of spatial niche overlaps; and (5) protection, where 0 is all protection considered and 1 is no protection considered. Results are presented in Extended Data Fig. 1. Overall, we found that our absolute energy flux estimations in belowground food webs were most sensitive to the degree of omnivory. This effect was driven by a low assimilation efficiency of specific food resources (for example, soil organic matter). Degree of omnivory used in the main analysis (auxiliary resources were assumed to be five times less important than the main ones) seems realistic considering that multichannel feeding has repeatedly been reported in soil invertebrates[55,86,87]. None of the tested settings undermined our main conclusions.

To test if our results were biased because canopy fogging underestimated the canopy arthropod biomasses, we used data on canopy heights. For that, during February 2013 to August 2014 we measured all trees in all plots with a minimum diameter at breast height of 10 cm, allowing us to calculate the average tree height and 90th quantile per plot. We assumed that canopy fogging was efficient until a certain tree height but failed to assess arthropods above this height. Ten different heights were tested, starting from 14 m maximum fogging efficiency (high undersampling) to 22 m maximum fogging efficiency (low undersampling). In each iteration, we used the range from 5 m (lower canopy) to the height of maximum efficiency as the 'assessed community' and everything above that height as 'unassessed community'. Assuming the same density and community composition in the unassessed community, we multiplied canopy arthropod biomass by the ratio of unassessed to assessed community. Final multiplication coefficients varied from 1.0 in most of the plantation plots (no undersampling) to 3.5–3.8 in several rainforest plots (only about 30% of arthropods were sampled). Food-web reconstructions and energy flux calculations were re-run using new canopy arthropod biomasses. Results are presented in Extended Data Fig. 2. Overall, we found that under 'high undersampling' scenario energy fluxes in aboveground food webs in rainforest increased almost threefold in comparison to our initial model. This increase was less evident in jungle rubber and almost not present in plantations in which the canopy height was low. Thus, the 'high undersampling' scenario exacerbated land-use effects on the total energy fluxes aboveground (−87% decline in rubber and

−92% decline in oil palm in comparison to −75% and −79% decline in our initial analysis, correspondingly). Despite these pronounced differences, none of the tested settings undermined our main conclusions.

## Reporting summary

Further information on research design is available in the Nature Portfolio Reporting Summary linked to this article.

## Data availability

Raw data used in the analysis are available from figshare: https://doi.org/10.6084/m9.figshare.24648438. The following datasets were used for bird identification and assignment of traits to invertebrates: xeno-canto online bird call database (http://xeno-canto.org/); Elton Traits[65]; feeding habits of invertebrates[60]; and stable isotope data[14]. Source data are provided with this paper.

## Code availability

Food-web reconstruction code is available from figshare: https://doi.org/10.6084/m9.figshare.24648438. Statistical models are specified in Methods.

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

**Acknowledgements** This study was funded by the Deutsche Forschungsgemeinschaft (DFG), project number 192626868–SFB 990 in the framework of the collaborative German-Indonesian research project CRC990. We thank PT REKI for granting us access and use of their properties. A.M.P. acknowledges support of the DFG Emmy Noether programme (project no. 493345801) and of iDiv (DFG–FZT 118, 202548816). C.W. is grateful for support by the DFG Heisenberg programme (project no. 493487387). We thank B. Gauzens for advice on energy flux calculations and discussion about the index for carbon balance. We thank S. Meyer and contributors of PhyloPic.org for the animal and resource silhouettes used in Figs. 1 and 2.

**Author contributions** A.M.P. developed the idea and led the writing. A.M.P., J.D., K.D., A.W., R.N., K.R., K., E.H.U., N.J., V.L. and A.M. provided the data. S.S., T.T., C.W., I.G., E.T., A.M.P., J.D., K.D., R.W., D.B. and P.H. contributed to coordination of the data collection. S.S., T.T., C.W., I.G., E.T., M.M.P., A.M.P., J.D., K.D. and A.W. contributed to conceptualization of the idea. All authors revised the manuscript.

**Competing interests** The authors declare no competing interests.

**Additional information**
**Correspondence and requests for materials** should be addressed to Anton M. Potapov.

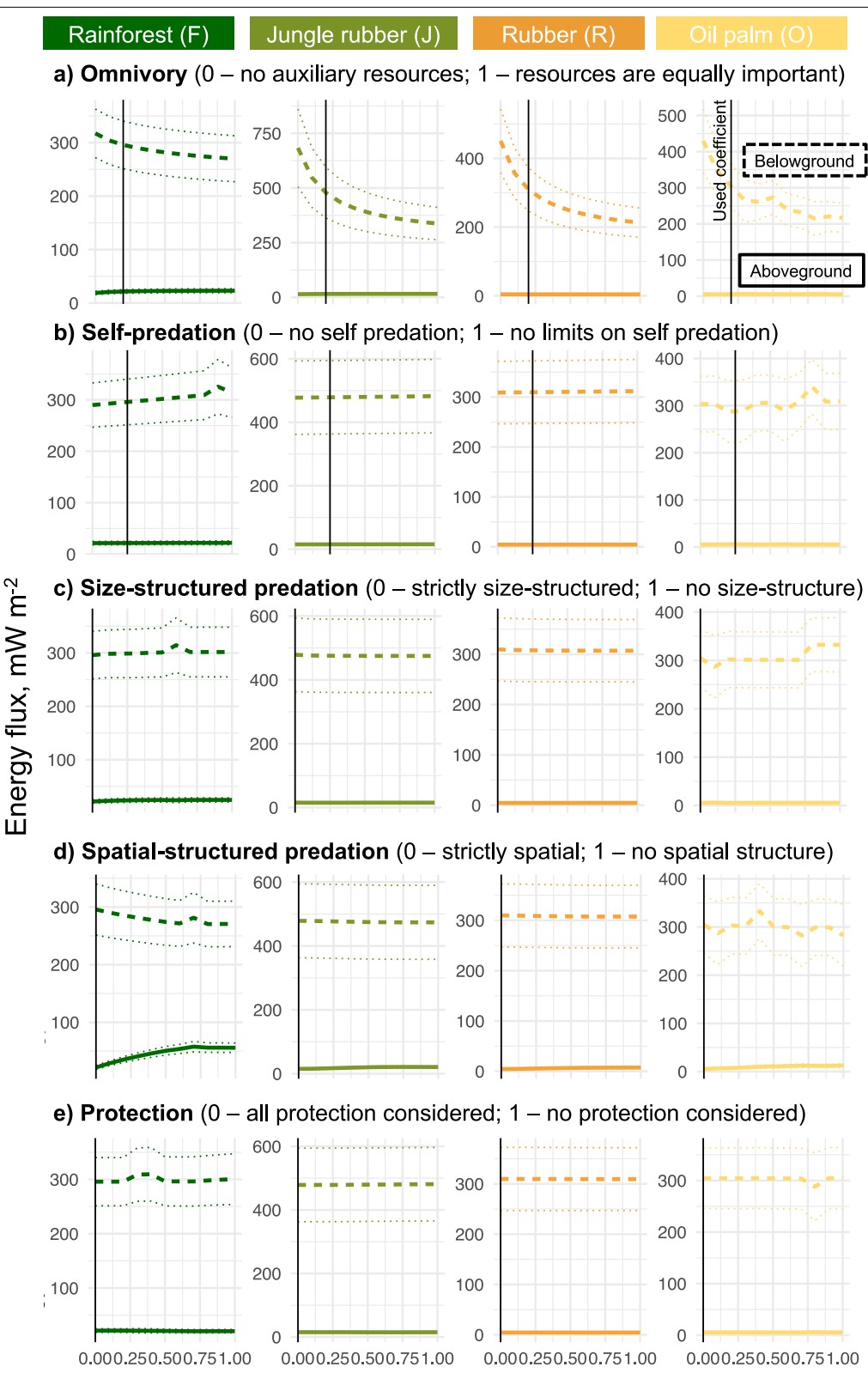

**Extended Data Fig. 1 | Sensitivity analysis of the food-web reconstruction.** Effect of the food-web reconstruction coefficients (i.e. food-web topology) on aboveground (solid line) and belowground energy fluxes (dashed line) in four land-use systems (dark green – forest, light green – jungle rubber, orange – rubber, yellow – oil palm; n = 8 sites per system, standard error is shown as the variation measure). Coefficients used in the main analysis are shown with black vertical lines. Omnivory controls how important auxiliary resources are for the consumers (a). Self-predation controls for the extent to which each node feeds on itself (cannibalism), where 1 means that individuals of their own guild are considered in the same way as those in all other nodes (b). Size-structured predation controls for deviations from the rules of predator–prey mass ratios (c). Spatial predation controls for deviations from the rules of spatial niche overlap (d). Protection controls for the importance of prey protection (e).

## a) Tree canopy heights

| Rainforest (F) | | | | Jungle rubber (J) | | | | Rubber (R) | | | | Oil palm (O) | | |
|---|---|---|---|---|---|---|---|---|---|---|---|---|---|---|
| Plot | Mean | Q90 | | Plot | Mean | Q90 | | Plot | Mean | Q90 | | Plot | Mean | Q90 |
| BF1 | 18.3 | 27.1 | | BJ3 | 15.0 | 20.1 | | BR1 | 13.2 | 16.1 | | BO2 | 10.1 | 11.5 |
| BF2 | 17.3 | 28.3 | | BJ4 | 14.5 | 21.5 | | BR2 | 13.4 | 14.8 | | BO3 | 11.0 | 12.1 |
| BF3 | 18.0 | 28.5 | | BJ5 | 16.4 | 22.3 | | BR3 | 13.2 | 14.9 | | BO4 | 10.4 | 11.3 |
| BF4 | 18.6 | 26.5 | | BJ6 | 16.0 | 21.4 | | BR4 | 13.7 | 16.0 | | BO5 | 10.5 | 11.7 |
| HF1 | 20.9 | 29.6 | | HJ1 | 16.6 | 22.2 | | HR1 | 16.3 | 19.8 | | HO1 | 13.4 | 15.1 |
| HF2 | 20.8 | 29.4 | | HJ2 | 14.7 | 20.7 | | HR2 | 14.5 | 17.9 | | HO2 | 12.7 | 14.0 |
| HF3 | 23.0 | 32.5 | | HJ3 | 15.1 | 19.7 | | HR3 | 14.6 | 19.2 | | HO3 | 11.7 | 13.1 |
| HF4 | 22.4 | 34.3 | | HJ4 | 14.9 | 19.8 | | HR4 | 13.8 | 17.8 | | HO4 | 11.0 | 12.5 |

## b) Canopy undersampling modelling: above-belowground comparison

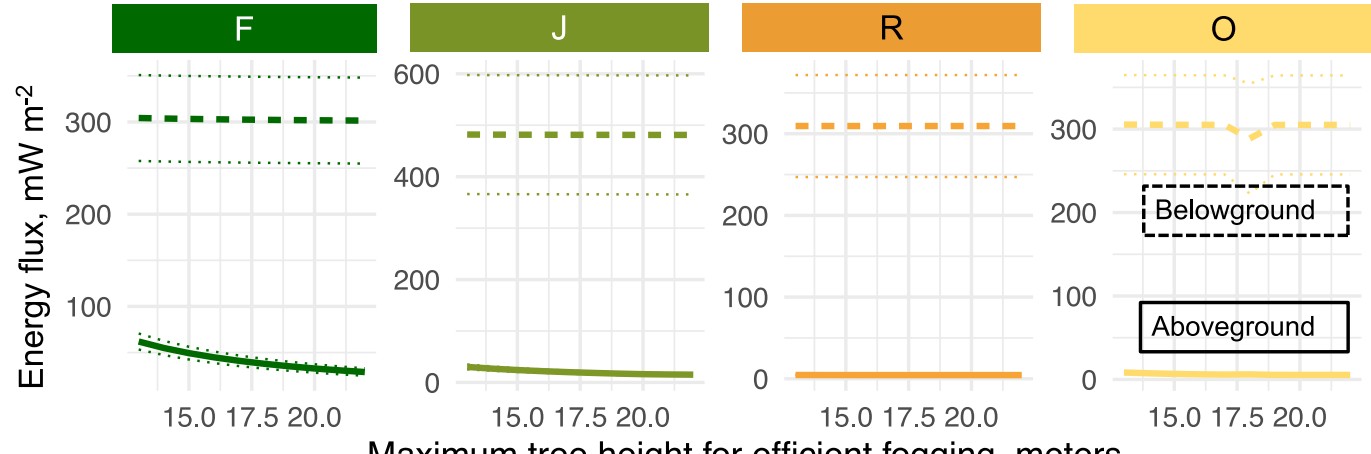

## c) Canopy undersampling modelling: land-use comparison (aboveground only)

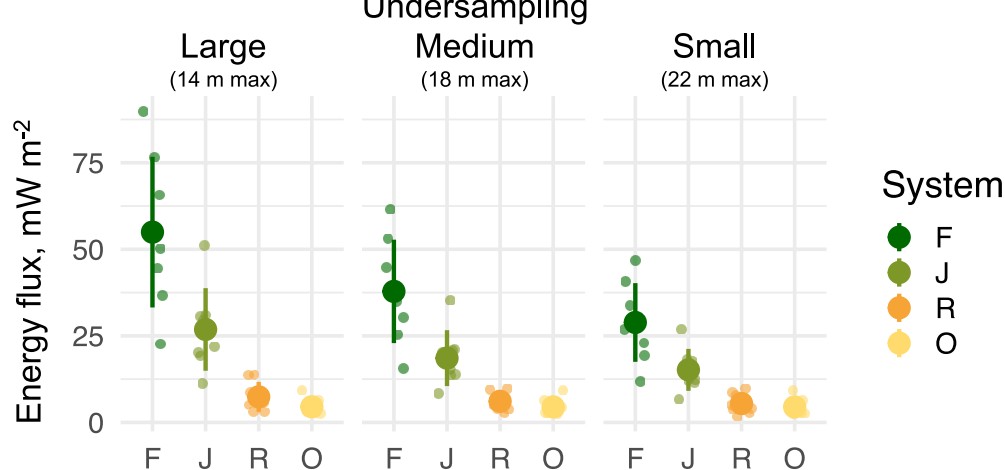

**Extended Data Fig. 2 | Sensitivity analysis of the canopy fogging undersampling.** Effect of canopy arthropod undersampling on the energy fluxes in four land-use systems (dark green – forest, light green – jungle rubber, orange – rubber, yellow – oil palm; n = 8 sites per system, standard error is shown as the variation measure). We tested how the maximum tree height for efficient fogging affected our results. Average and 90th quantile (Q90) of tree height were calculated for each plot (a). The quantiles were further used to estimate bias in energy flux estimations aboveground (solid line) and belowground (dashed line), assuming that fogging works efficiently to assess canopy arthropods up to a certain tree height (models were run for each additional metre starting from 14 m as a 'large bias' and up to 22 m as a 'small bias'; b). Land-use effect was more pronounced if we assumed a large bias due to higher trees in rainforest, than in plantations (n = 8, 1 SD variation; c).

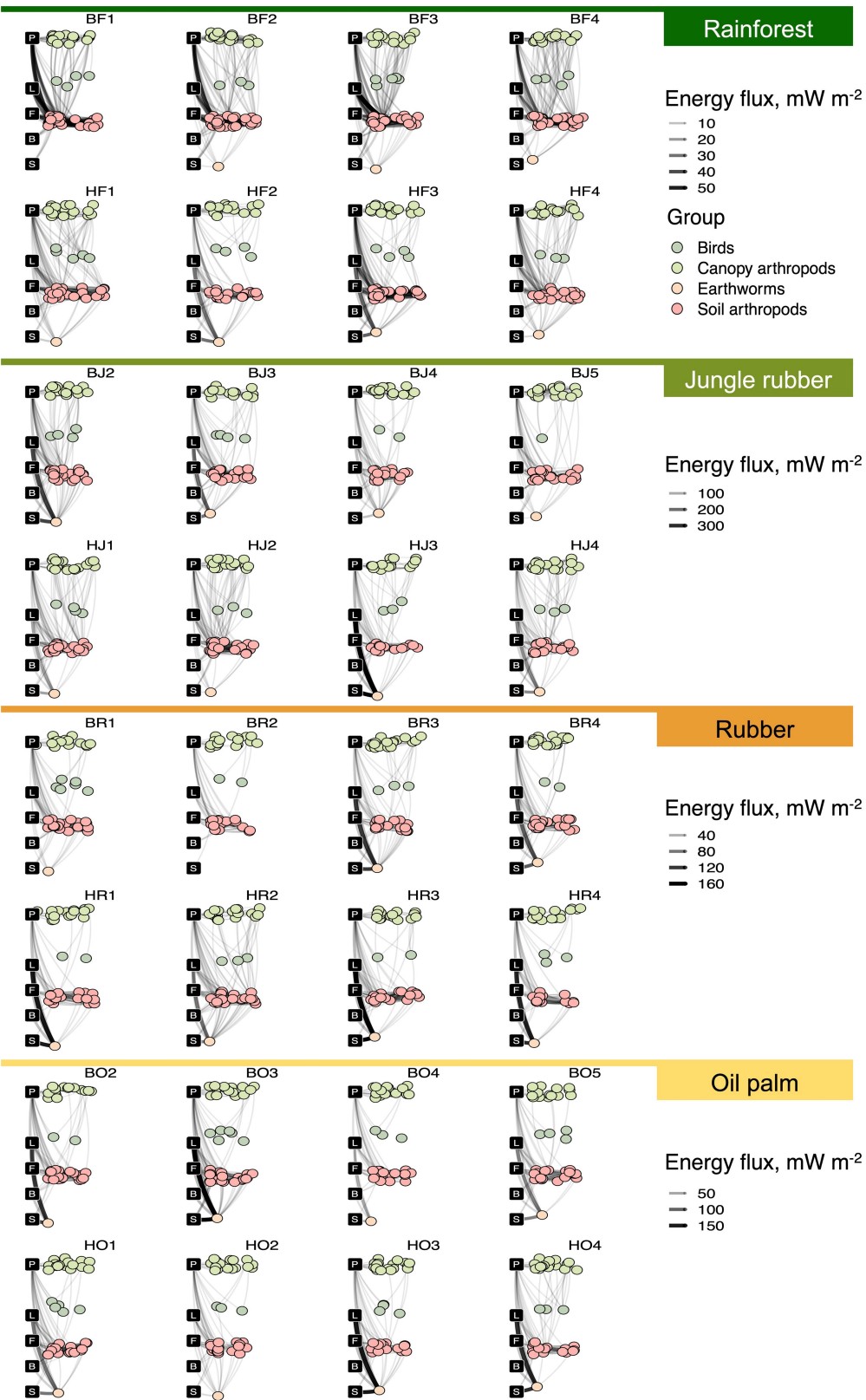

**Extended Data Fig. 3 | Reconstructed food webs in each land-use system from the main survey in 2013.** Eight food webs per system representing eight sites are shown. "B" and "H" in the site codes refer to the "Bukit Duabelas" and "Harapan" regions, correspondingly. Food-web nodes include basal resources displayed with black labels (living plants – P, plant litter – L, fungi – F, bacteria – B, soil organic matter – S) and consumer trophic guilds shown with circles. Consumer nodes are clustered in four major groups according to their vertical distribution and ecological niches (canopy arthropods – light green, birds – dark green, soil arthropods – light red, earthworms – beige). Horizontal distribution of consumer nodes represent trophic positions (trophic level increases from left to right). Connecting lines on the food-web diagram represent energy fluxes in mW m⁻² (represented by the thickness of the lines).

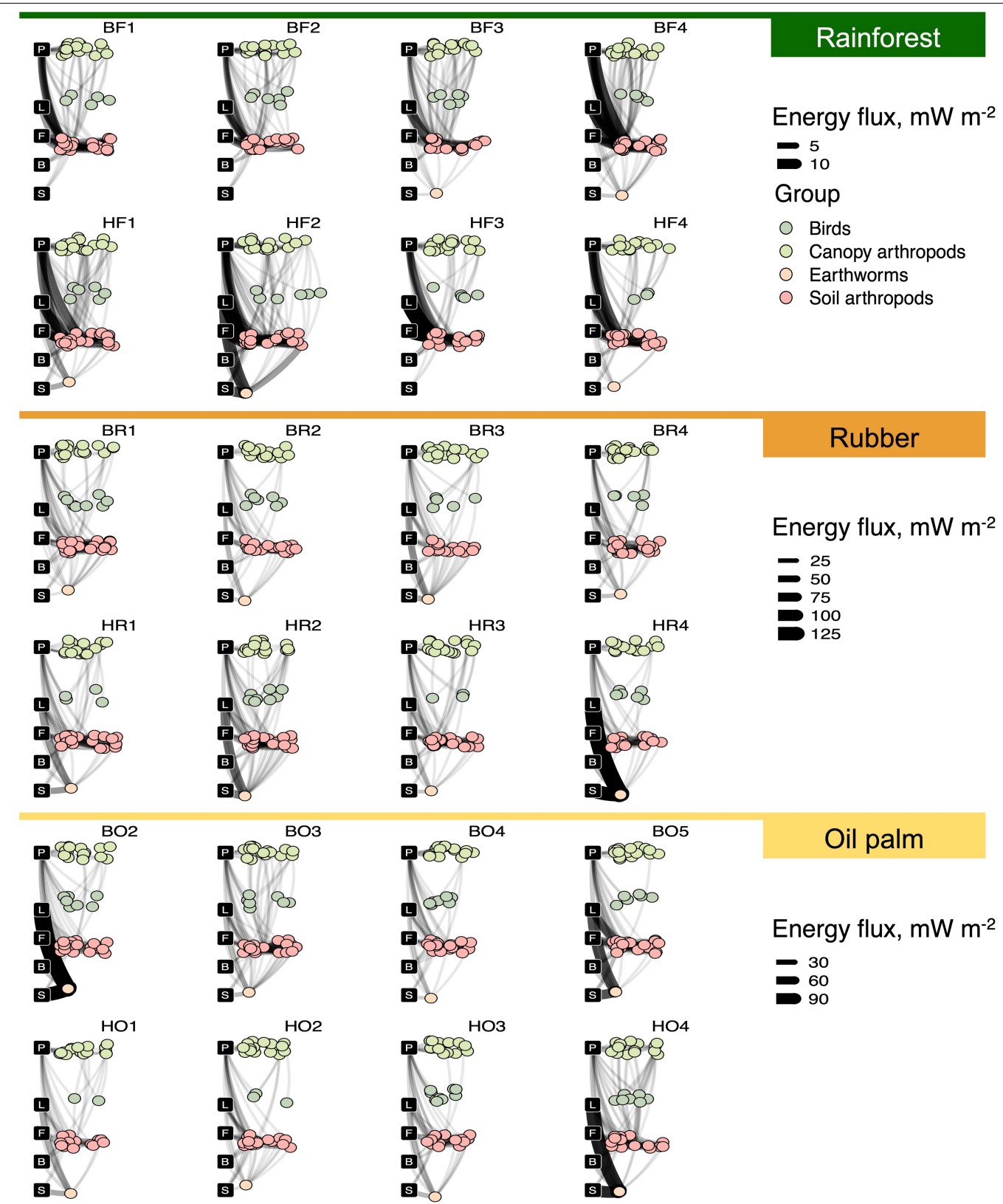

**Extended Data Fig. 4 | Reconstructed food webs in each land-use system from the validation survey in 2016–2017.** Eight food webs per system representing eight sites are shown. "B" and "H" in the site codes refer to the "Bukit Duabelas" and "Harapan" regions, respectively. Food-web nodes include basal resources displayed with black labels (living plants – P, plant litter – L, fungi – F, bacteria – B, soil organic matter – S) and consumer trophic guilds shown with circles. Consumer nodes are clustered in four major groups according to their vertical distribution and ecological niches (canopy arthropods – light green, birds – dark green, soil arthropods – light red, earthworms – beige). Horizontal distribution of consumer nodes represent trophic positions (trophic level increases from left to right). Connecting lines on the food-web diagram represent energy fluxes in mW m⁻² (represented by the thickness of the lines).

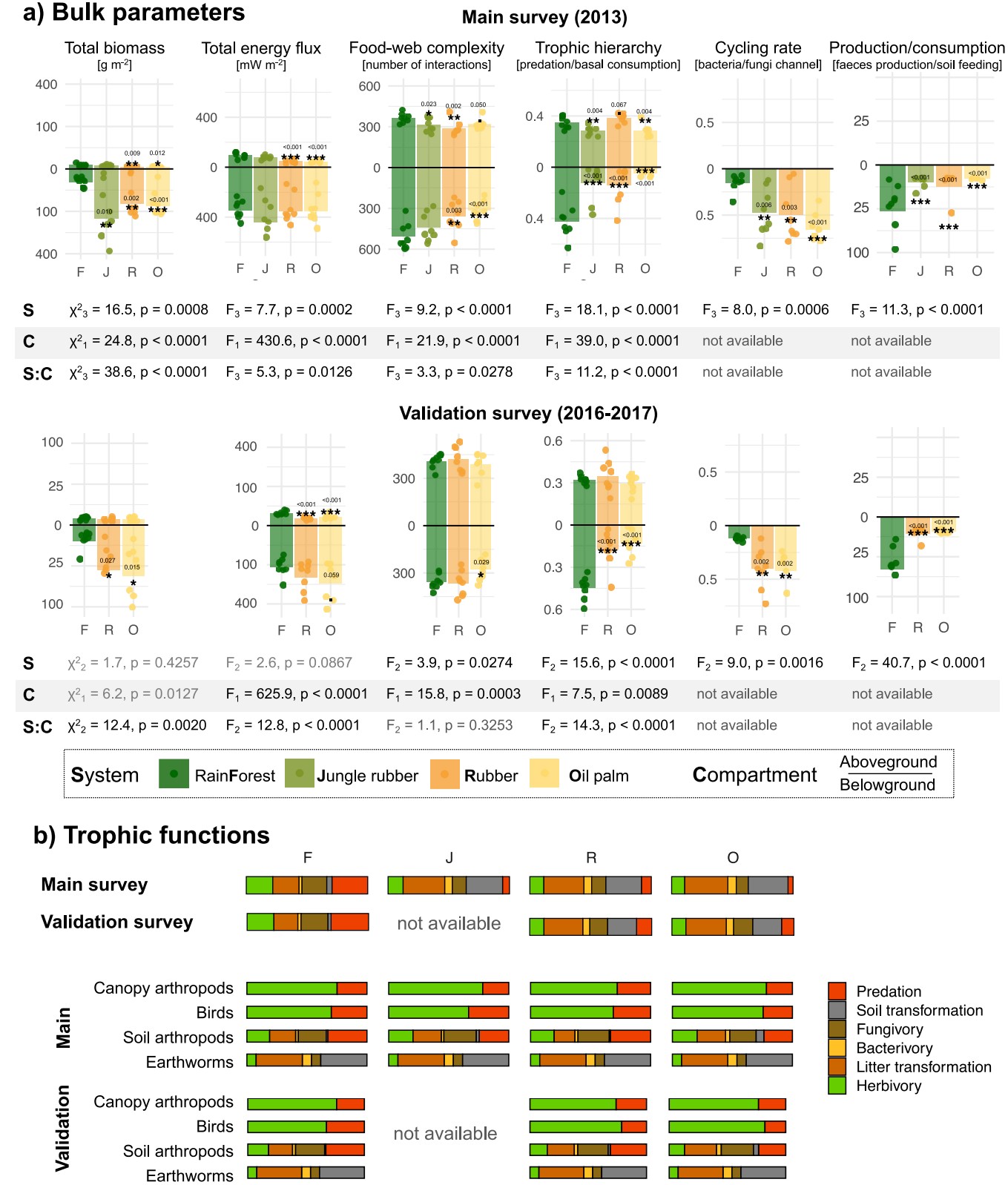

## a) Bulk parameters

### Main survey (2013)

| | Total biomass [g m⁻²] | Total energy flux [mW m⁻²] | Food-web complexity [number of interactions] | Trophic hierarchy [predation/basal consumption] | Cycling rate [bacteria/fungi channel] | Production/consumption [faeces production/soil feeding] |
|---|---|---|---|---|---|---|
| **S** | $\chi^2_3$ = 16.5, p = 0.0008 | $F_3$ = 7.7, p = 0.0002 | $F_3$ = 9.2, p < 0.0001 | $F_3$ = 18.1, p < 0.0001 | $F_3$ = 8.0, p = 0.0006 | $F_3$ = 11.3, p < 0.0001 |
| **C** | $\chi^2_1$ = 24.8, p < 0.0001 | $F_1$ = 430.6, p < 0.0001 | $F_1$ = 21.9, p < 0.0001 | $F_1$ = 39.0, p < 0.0001 | not available | not available |
| **S:C** | $\chi^2_3$ = 38.6, p < 0.0001 | $F_3$ = 5.3, p = 0.0126 | $F_3$ = 3.3, p = 0.0278 | $F_3$ = 11.2, p < 0.0001 | not available | not available |

### Validation survey (2016-2017)

| | | | | | | |
|---|---|---|---|---|---|---|
| **S** | $\chi^2_2$ = 1.7, p = 0.4257 | $F_2$ = 2.6, p = 0.0867 | $F_2$ = 3.9, p = 0.0274 | $F_2$ = 15.6, p < 0.0001 | $F_2$ = 9.0, p = 0.0016 | $F_2$ = 40.7, p < 0.0001 |
| **C** | $\chi^2_1$ = 6.2, p = 0.0127 | $F_1$ = 625.9, p < 0.0001 | $F_1$ = 15.8, p = 0.0003 | $F_1$ = 7.5, p = 0.0089 | not available | not available |
| **S:C** | $\chi^2_2$ = 12.4, p = 0.0020 | $F_2$ = 12.8, p < 0.0001 | $F_2$ = 1.1, p = 0.3253 | $F_2$ = 14.3, p < 0.0001 | not available | not available |

**System** — RainForest (dark green), Jungle rubber (light green), Rubber (orange), Oil palm (yellow)

**Compartment** — Aboveground / Belowground

## b) Trophic functions

**Predation** (red), **Soil transformation** (grey), **Fungivory** (dark yellow/olive), **Bacterivory** (yellow), **Litter transformation** (brown/orange), **Herbivory** (green)

**Extended Data Fig. 5 | Validation of the main survey from 2013 with the results of the validation survey from 2016–2017.** Effects of land use and above/belowground compartment identity on bulk food-web parameters compared between the main survey in 2013 and the validation survey in 2016–2017 (a). Each point is a site, bars represent means. Colours denote land-use systems (dark green – forest, light green – jungle rubber, orange – rubber, yellow – oil palm; n = 8 sites per system). Note square root scale in biomass, total energy flux and carbon balance charts. Asterisks mark significant differences between means for the given parameter above- or belowground from that in rainforest (two-tailed; ***p < 0.001, **p < 0.01, *p < 0.05). Effects of land-use system and above/belowground on the tested parameters are given below the corresponding bar charts. Average trophic functions for each major group of consumers and for the food web in total are summarized as stacked proportional bar charts (b). Refer to Figs. 2 and 3 in the main text for more detailed explanations. Note that jungle rubber was not assessed in 2016–2017.

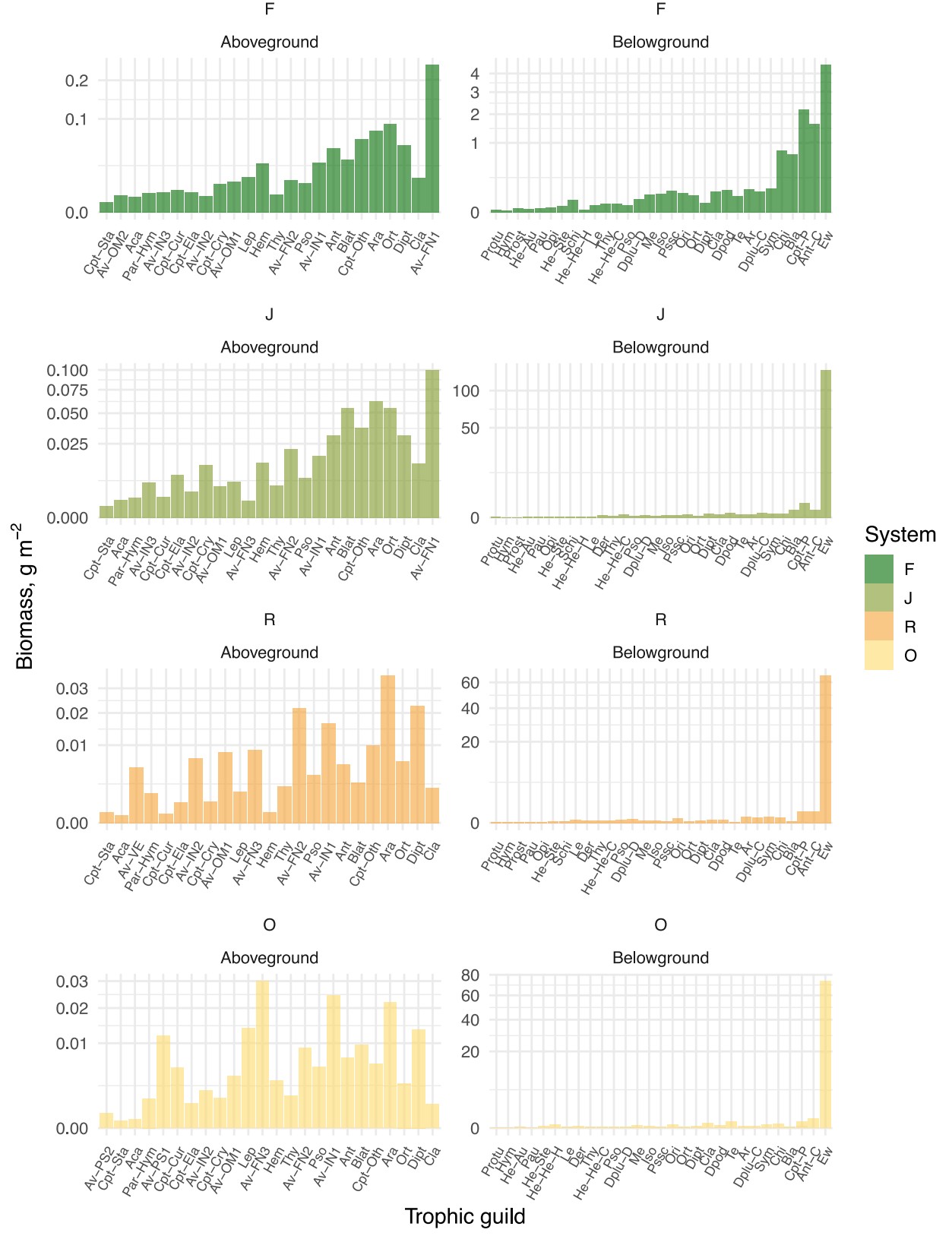

**Extended Data Fig. 6 | Biomasses of trophic guilds in different land-use systems.** Guilds of aboveground (birds, canopy arthropods) and belowground (earthworms and soil arthropods) animals are shown separately and ordered according to average biomass across systems. Mean fresh biomasses from the main survey in 2013 are displayed with bars. Full definitions and abbreviations (for example Ew – earthworms, Av – birds) of the trophic guilds are given in the data tables available from Figshare (see Data availability).

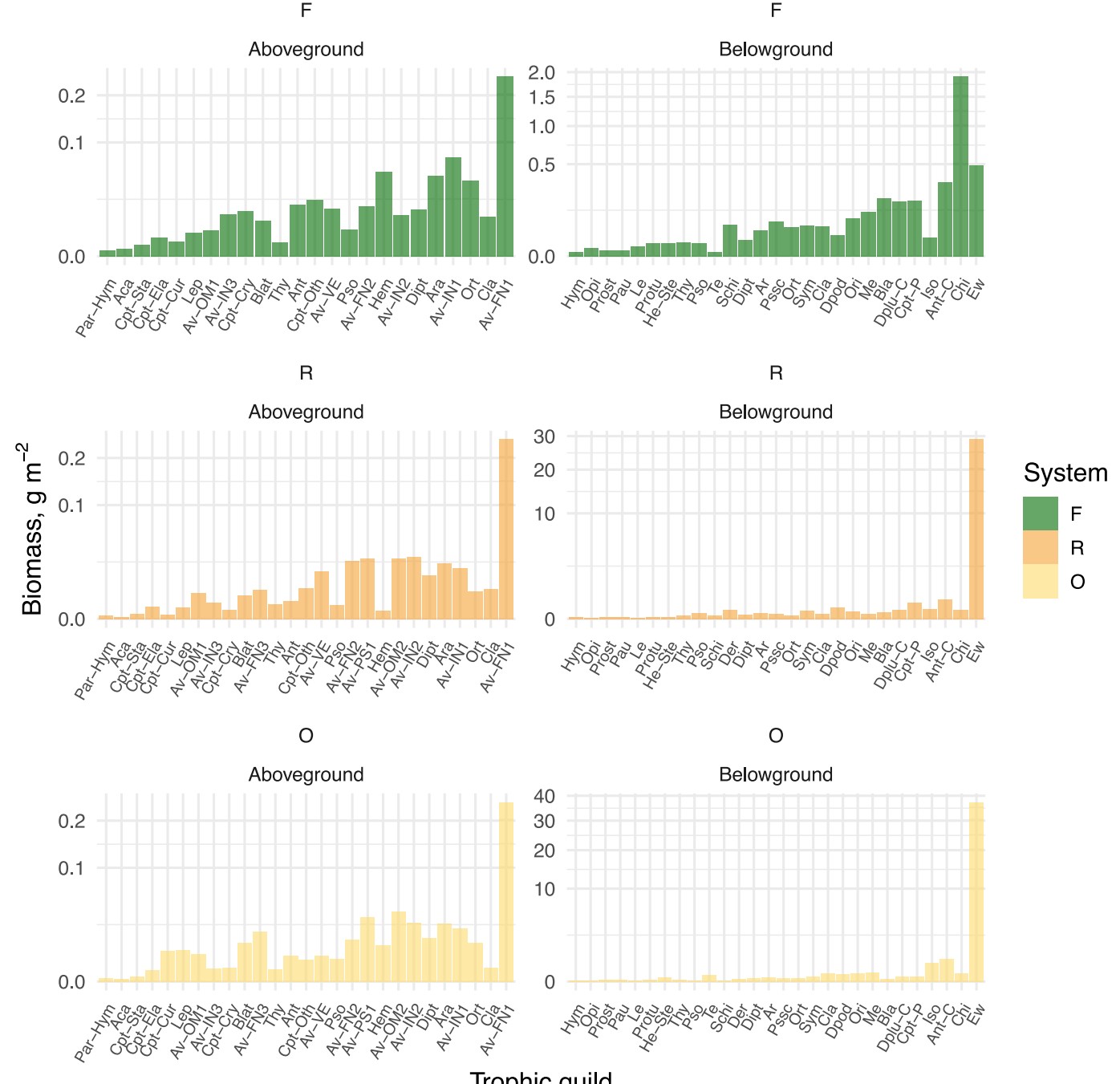

**Extended Data Fig. 7 | Biomasses of trophic guilds in different land-use systems in the validation survey.** Guilds of aboveground (birds, canopy arthropods) and belowground (earthworms and soil arthropods) animals are shown separately and ordered according to average biomass across systems. Mean fresh biomasses from the validation survey in 2016–2017 are displayed with bars. Full definitions and abbreviations (for example Ew – earthworms, Av – birds) of the trophic guilds are available from Figshare (see Data availability).

**Extended Data Table 1 | Effects of land use on the energy flux distribution across major animal groups and trophic functions**

Our validation survey showed that the total energy flux and biomass in rainforest was ca. 2.5 times lower than that in the main survey due to a lower abundance of earthworms, ants, and other dominant group (Extended Data Figs. 6 and 7). This may be a consequence of the drought that followed the El Niño events in 2015 the study region[88]. Despite this difference in the absolute values, the proportion of aboveground fluxes in the total energy flux was nearly the same (6.8% in the main and 8.1% in the validation survey; Extended Data Fig. 5). The proportion of birds in the total canopy energy flux in rainforest was also similar in both surveys (7.4% in the main and 6.6% in the validation survey). Earthworms were responsible for an average of 5.5% of the energy flux in rainforest and for 52-54% of the energy flux across plantations; both numbers were lower than in the main survey, but the strong differences among land-use types remained, and other energetically important groups were similar between the main and validation surveys (Extended Data Figs. 6 and 7). The belowground energy flux was manifold higher than the aboveground across land-use systems, and the total aboveground energy flux was reduced by -53 to -67% in both monoculture plantation types in comparison to rainforest (-75 to -79% in the main survey; Extended Data Fig. 5). We also confirmed a decline in the belowground predation/consumption ratio in monoculture plantations (-62 to -71%), and a moderate decline aboveground in oil palm (-8%), but not in rubber plantations (+8%). Similar changes in trophic functions were recorded in both surveys: total herbivory and fungivory were lower, while total bacterivory, soil feeding, and bacteria/fungi energy flux ratio were higher in monoculture plantations than in rainforest. However, in contrast to the main survey, our validation survey was unable to detect a significant reduction in the number of trophic interactions in plantation systems (except oil palm belowground; Extended Data Fig. 5). Overall, we have validated energetic dominance of the belowground over the aboveground energy channel, canopy arthropods over birds, energetic decline in canopies, and reallocation of energy to belowground food webs in plantations, and shifts in trophic functions, but were not able to validate food-web simplification across above- and belowground compartments.

Results of linear mixed-effect models testing the effect of land-use system, region (two geographical areas) and either animal group (canopy arthropods, birds, soil arthropods, earthworms) or trophic function (see Fig. 2), on energy fluxes in food webs. Chi-squared, degrees of freedom (d.f.) and significance (p) were approximated using Wald Chi-squared tests Type II (*car* package). Models for groups and functions were run using different summaries of energy fluxed by corresponding grouping variables. Models for the main and validation survey were run and are presented separately. Note that jungle rubber is not included in the validation survey.

**Extended Data Table 2 | Regression coefficients used to calculate group-specific metabolic rates**

| Animal groups | a | E | ln($i_0$) |
|---|---|---|---|
| Arachnida | -0.4347 ± 0.0211 | 0.7093 ± 0.0357 | 20.297 ± 1.427 |
| Chilopoda | -0.4419 ± 0.0313 | 0.803 ± 0.0434 | 23.919 ± 1.739 |
| Clitellata | -0.1993 ± 0.0309 | 0.4433 ± 0.0461 | 9.784 ± 1.861 |
| Insecta | -0.2411 ± 0.0232 | 0.6574 ± 0.0388 | 19.026 ± 1.548 |
| Isopoda | -0.4455 ± 0.0422 | 0.6867 ± 0.0494 | 18.81 ± 1.957 |
| Mesostigmata | -0.3095 ± 0.0885 | 0.3793 ± 0.0175 | 6.255 ± 7.124 |
| Oribatida | -0.3206 ± 0.0372 | 0.7061 ± 0.0636 | 18.527 ± 2.564 |
| Progoneata | -0.4287 ± 0.0366 | 0.67 ± 0.0487 | 18.105 ± 1.940 |
| Prostigmata | -0.3401 ± 0.0675 | 0.4125 ± 0.1530 | 6.652 ± 6.189 |
| Endothermic vertebrates | -0.29 | 0.69 | 19.5 |

Metabolic rates in Watt per gram ($Me$) were calculated using the formula:[76] $ln(Me) = ln(i_0) + a\ ln(M) - E(1/kT)$, where $i_0$ is a normalization factor, $a$ is the allometric exponent (per unit mass), $M$ is the fresh body mass in gram, $E$ is the activation energy, $k$ is the Boltzmann's constant ($8.62 * 10^{-5}$ eV K$^{-1}$) and $T$ is the environmental temperature in Kelvin. Standard errors, whenever available, are given as a measure of variation. Regression coefficients for vertebrates were taken from Brown et al. [11], all other coefficients from Ehnes et al. [82].

# Reporting Summary

## Statistics

For all statistical analyses, confirm that the following items are present in the figure legend, table legend, main text, or Methods section.

| n/a | Confirmed | |
|---|---|---|
| ☐ | ☒ | The exact sample size (*n*) for each experimental group/condition, given as a discrete number and unit of measurement |
| ☐ | ☒ | A statement on whether measurements were taken from distinct samples or whether the same sample was measured repeatedly |
| ☐ | ☒ | The statistical test(s) used AND whether they are one- or two-sided *Only common tests should be described solely by name; describe more complex techniques in the Methods section.* |
| ☐ | ☒ | A description of all covariates tested |
| ☐ | ☒ | A description of any assumptions or corrections, such as tests of normality and adjustment for multiple comparisons |
| ☐ | ☒ | A full description of the statistical parameters including central tendency (e.g. means) or other basic estimates (e.g. regression coefficient) AND variation (e.g. standard deviation) or associated estimates of uncertainty (e.g. confidence intervals) |
| ☐ | ☒ | For null hypothesis testing, the test statistic (e.g. *F*, *t*, *r*) with confidence intervals, effect sizes, degrees of freedom and *P* value noted *Give P values as exact values whenever suitable.* |
| ☒ | ☐ | For Bayesian analysis, information on the choice of priors and Markov chain Monte Carlo settings |
| ☒ | ☐ | For hierarchical and complex designs, identification of the appropriate level for tests and full reporting of outcomes |
| ☒ | ☐ | Estimates of effect sizes (e.g. Cohen's *d*, Pearson's *r*), indicating how they were calculated |

*Our web collection on statistics for biologists contains articles on many of the points above.*

## Software and code

Policy information about availability of computer code

| Data collection | All data were collected from invertebrate samples and bird observations using manual counting and identification. No software was used during the data collection. BioSounds was used as a platform to get identification on the bird audio recordings (Darras, K., Pérez, N., Mauladi & Hanf-Dressler, T. BioSounds: an open-source, online platform for ecoacoustics. F1000Res. 9, 1224 (2020)) |
|---|---|
| Data analysis | Data analysis was implemented in R v4.2.0 with R studio interface v1.4.1103 (RStudio, PBC). The following packages were used: lme4 v1.1-33, tidyverse v2.0.0, plyr v1.8.8, reshape v0.8.9, reshape2 v1.4.4, fluxweb v0.2.0, readxl v1.4.2, ggrepel v0.9.3, Food-web reconstruction code is openly available from Figshare (https://doi.org/10.6084/m9.figshare.24648438). Linear models are specified in the Extended Data Table 4. |

For manuscripts utilizing custom algorithms or software that are central to the research but not yet described in published literature, software must be made available to editors and reviewers. We strongly encourage code deposition in a community repository (e.g. GitHub). See the Nature Portfolio guidelines for submitting code & software for further information.

## Data

Policy information about availability of data

All manuscripts must include a data availability statement. This statement should provide the following information, where applicable:
- Accession codes, unique identifiers, or web links for publicly available datasets
- A description of any restrictions on data availability
- For clinical datasets or third party data, please ensure that the statement adheres to our policy

Raw data used in the analysis are available from Figshare: https://doi.org/10.6084/m9.figshare.24648438. The following datasets were used for bird identification and assignment of traits to invertebrates: Xeno-Canto online bird call database (http://xeno-canto.org/) and EltonTraits (Wilman, H. et al. EltonTraits 1.0: Species-level foraging attributes of the world's birds and mammals. Ecology 95, 2027–2027, 2014); Feeding habits review: Potapov, A. M. et al. Feeding habits and multifunctional classification of soil-associated consumers from protists to vertebrates. Biol. Rev. Camb. Philos. Soc. 97, 1057–1117 (2022); Stable isotope data: Zhou, Z., Krashevska, V., Widyastuti, R., Scheu, S. & Potapov, A. Tropical land use alters functional diversity of soil food webs and leads to monopolization of the detrital energy channel. Elife 11, (2022)

## Human research participants

Policy information about studies involving human research participants and Sex and Gender in Research.

| | |
|---|---|
| Reporting on sex and gender | N/A |
| Population characteristics | N/A |
| Recruitment | N/A |
| Ethics oversight | N/A |

Note that full information on the approval of the study protocol must also be provided in the manuscript.

# Field-specific reporting

Please select the one below that is the best fit for your research. If you are not sure, read the appropriate sections before making your selection.

☐ Life sciences        ☐ Behavioural & social sciences        ☒ Ecological, evolutionary & environmental sciences

For a reference copy of the document with all sections, see nature.com/documents/nr-reporting-summary-flat.pdf

# Ecological, evolutionary & environmental sciences study design

All studies must disclose on these points even when the disclosure is negative.

| | |
|---|---|
| Study description | The study explores differences between land-use systems (rainforest, jungle rubber, rubber, oil palm), and food-web compartments (aboveground, belowground), having a factorial design. In total, 32 sites were assessed, 8 in each system. |
| Research sample | The study analyses canopy and soil arthropods, birds, and earthworms in Jambi province, Sumatra, Indonesia. |
| Sampling strategy | The sampling was done on 8 independent sites (replicates) per system in two landscapes. No procedure was used to predetermine the sampling size. |
| Data collection | Authors of the paper applied a combination of collection methods to assess bird (point counts, sound recorders), canopy arthropod (fogging), soil arthropod and earthworm (heat extraction) communities. |
| Timing and spatial scale | The main sampling was done during May-October 2013 in two landscapes (around 'Harapan' and 'Bukit Duabelas' national parks) in Jambi province, Sumatra, Indonesia. Additional validation survey was done on 24 out of the 32 sampling sites in 2016-2017. Sampling sites were distributed over an area with a diameter of ca. 80 km. |
| Data exclusions | No data were excluded from the analysis |
| Reproducibility | We performed validation survey four years after the main survey on a subset of sites, which confirmed most of our conclusions. R code and statistical model specifications are openly available allowing to reproduce the data analysis. |
| Randomization | Sites, subplots within sites, and samples within subplots were randomly established taken. For more information see http://rstb.royalsocietypublishing.org/lookup/doi/10.1098/rstb.2015.0275 |
| Blinding | This is an observational study dealing with animal biodiversity. Same people assessed different sites. |

Did the study involve field work?  ☒ Yes  ☐ No

# Field work, collection and transport

| Field conditions | The climate is tropical humid with two peak rainy seasons around March and December, and a dryer period during July-August. Average annual temperature in the area is 26.7 ± 0.2°C, annual precipitation is 2235 ± 381 mm |
|---|---|
| Location | The study was carried out in Jambi province, Sumatra, Indonesia |
| Access & import/export | The permits for collection and export of the samples were granted by the Indonesian Ministry of Forestry (PHKA), Directorate General of Nature Resources and Ecosystem Conservation (KSDAE), and the Indonesian Institute of Sciences (LIPI). For soil invertebrates: collection permit no. S.07/KKH-2/2013 issued by PHKA, and export permit by LIPI (register file no. 24/SI/MZB/IV/2014) and PHKA (no. 125/KKH-5/TRP/2014). For canopy arthropods: collection Permit No. S.710/KKH-2/2013 issued by PHKA based on recommendation No. 2122/IPH.1/KS.02/X/2013 by LIPI, and export permit SK.61/KSDAE/SET/KSA.2/3/2019 issued by KSDAE based on LIPI recommendation B1885/IPH.1/KS.02.04/ VII/2017. For birds: field observations were carried out by Indonesian colleagues, so no research or collection permit was required. The research permit (number 211/SIP/FRP/SM/VI/2012) was recommended by LIPI and issued by PHKA. No bird collection or exporting was carried out. For the validation survey, the following research permits were used: canopy arthropods: 131/SIP/FRP/E5/Dit.KI/V/2017, birds: 386/SIP/FRP/E5/Dit.KI/XI/2016, soil invertebrates: 2841/IPH.1/KS.02.04/ X/2016 (collection permit), canopy height and tree stand properties: 42/EXT/SIP/FRP/E5/Dit.KI/VIII/2016 and 56/EXT/SIP/FRP/E5/ Dit.KI/IX/2017 |
| Disturbance | Work on the study sites was implemented with care, to minimize disturbance. Whenever possible, manipulations with samples were done in a laboratory, outside the field sampling areas. |

# Reporting for specific materials, systems and methods

We require information from authors about some types of materials, experimental systems and methods used in many studies. Here, indicate whether each material, system or method listed is relevant to your study. If you are not sure if a list item applies to your research, read the appropriate section before selecting a response.

## Materials & experimental systems

| n/a | Involved in the study |
|---|---|
| ☒ | ☐ Antibodies |
| ☒ | ☐ Eukaryotic cell lines |
| ☒ | ☐ Palaeontology and archaeology |
| ☐ | ☒ Animals and other organisms |
| ☒ | ☐ Clinical data |
| ☒ | ☐ Dual use research of concern |

## Methods

| n/a | Involved in the study |
|---|---|
| ☒ | ☐ ChIP-seq |
| ☒ | ☐ Flow cytometry |
| ☒ | ☐ MRI-based neuroimaging |

## Animals and other research organisms

Policy information about studies involving animals; ARRIVE guidelines recommended for reporting animal research, and Sex and Gender in Research

| Laboratory animals | Study did not involve laboratory animals |
|---|---|
| Wild animals | Only invertebrate animals (arthropods and earthworms) were collected and killed using ethanol during the study. This was necessary to assess biomass and community composition. We collected canopy arthropod communities using insecticide fogging, soil arthropod and earthworm communities using Kempson extractors and assessed bird communities using point counts and audio recorders. Presumably several hundreds of species (mainly unidentified) were collected without selection for age or strains. |
| Reporting on sex | Sex was not considered in the study |
| Field-collected samples | Collected soil samples were transported in the lab within 2-3 days for heat extraction. All invertebrates were stored in 70-80% ethanol solution. No field-collected environmental samples were used in this study. |
| Ethics oversight | No ethical approval was required. The study did not involve vertebrate animal capturing and killing |

Note that full information on the approval of the study protocol must also be provided in the manuscript.

