## [Peer Review File · Nature]

Manuscript Title: Rainforest transformation reallocates energy from green to brown food webs

Reviewer Comments & Author Rebuttals

Reviewer Reports on the Initial Version:

Referees' comments:

Referee #1 (Remarks to the Author):

One of the major threats to global biodiversity is the transformation of tropical rainforests into agriculture-oriented land-use systems like agroforestry and mono-culture plantations. Obviously, the effects of such land-use changes are numerous. Land-use change has direct effects on the aboveground vegetation and primary productivity, and along with this on the belowground productivity in terms of altered root turn-over and root exudates. These changes in productivity will affect almost all groups of organisms at various trophic levels, including herbivores and predators aboveground, and microbes and fauna belowground, and along with this the total energy that comes available for the above- and belowground food webs with various consequences for the fluxes among the trophic groups within the food webs. Hence, observing, measuring and understanding the complex of changes along land-use change is an enormous task, and hence offers at the same time a great scientific challenge. The study by Potatov takes up this challenge by taking an ecosystem approach. The study focuses on the aboveground as well as the belowground ecosystem compartment and the linkages between the two. The study also includes the analysis of the effects on the above- and belowground food web structure and function by measuring population sizes of various groups of organisms and inferring the fluxes among these groups using metabolic parameters. The study focuses on three kinds of land-use transformations, i.e., from tropical forest into rubber agroforestry, and into monoculture plantations of rubber and oil palm. The paper is very well written and has some nice clarifying figures. The results are scientifically interesting and societally relevant.

Given this the manuscript has the potential to be published in a high-quality scientific journal. To make the paper acceptable for publication in Nature I have though the following questions, suggestions and recommendations to revise the paper.

1. The choice of selecting (only) two groups of organisms per ecosystem compartment as representative examples, i.e., arthropods and birds aboveground and arthropods and earthworms belowground is understandable, one cannot measure everything. But I think these particular choices can be more strongly justified. For example, earthworms are an excellent choice as they occupy a central position in belowground ecosystem functioning ('ecosystem engineers') as well as in the soil food webs by eating almost everything ('ecophores'). Also, the choice for arthropods makes sense as most arthropods can be found at various trophic levels, e.g., microbivores, herbivores, predators, and within these trophic levels they can have a very diverse diet. In this way composition and abundances of the arthropods may also reflect the composition of the food web as a whole.

2. Some hypotheses, including the main one, are a little unsatisfactory formulated. Instead of an 'un-directional' formulation in terms of 'differences' I recommend formulations in terms of a priori predictions or 'increases' or 'decreases' as much as possible.

3. In Figure 2. we see a large increase in earthworms after transformation, 'compensating' for a large decrease in arthropods. I did not find an explanation for this decrease, but maybe I overlooked something. Has this something to do with disturbance?

4. Table 1. For tables in the main text, I prefer that they give more specific information in terms of which group, what biomass or energy flux. Levels of significance might be given in the text, so that such a table can be part of the supplementary materials/extended data.

5. Regarding the predator-declines, the observed increase in mono-culture rubber is seen as an exception. But isn't it one out of only 3 (land-use changes), or even one out of 2 (monocultures)?

6. Finally, the results are clearly described and nicely presented in the figures. The (possible) explanations are adequate. There is something though that I miss in the explanation and that is that it is likely that the land transformations have led to fast, and almost direct changes aboveground, and slow and indirect changes belowground. Aboveground, the vegetation is replaced quickly and so will be the faunal groups that depend on the vegetation. Belowground, rapid changes can be expected in the rhizosphere through altered root and exudate composition, and in the composition of the rhizosphere organisms, but changes in the detrital food web may take much longer. The age of the transformed systems (8-17 years) seems adequate to have a complete transformation aboveground but maybe not for the belowground compartment. Hence, the aboveground compartment may reflect the present, but the belowground compartment may reflect also the history. This might relate to some observed effect I think, such as the finding that reduction in the total energy flux goes along with losses in the aboveground biodiversity, and not (or less) in the belowground biodiversity. And also, that the brown food web showed a higher resilience than green food webs. I am not sure, but it might be interesting when the authors take this notion into account.

Peter de Ruiter

Referee #2 (Remarks to the Author):

The paper presents an observational and comparative study to construct partial food webs within native and managed tropical forests in Indonesia and infer energy fluxes among groups of organisms within the systems.

The paper presents multiple goals and questions: 1) description of the aboveground and belowground portions of the plant-arthropod-earthworm-bird components of the native and managed forests, that include estimates of energy flow within the systems based on estimates of the biomasses of organisms sampled, 2) a comparison of the webs and estimates across the different forest and management types, and 3) a generalization of the implications of their findings.

The results are incremental in terms of the impacts of the findings on food web responses to forest management and land use.

The paper uses a familiar energetic modeling framework to estimate energy flow. The results reveal some interesting and familiar outcomes. First, the authors note that tropical systems are known for their diversity and productivity. The energy flow descriptions revealed "... that most of the energy in rainforests is channeled to the brown animal web." This is consistent with what has been reported in most terrestrial ecosystems. Likewise, the loss of both predators and total energy flux to higher trophic levels, the shift from the 'slower' plant and fungal to the 'fast' plant and bacterial channels, and the internal reallocation of energy flow in the managed systems compared to the native forest is consistent with similar comparisons cropping, grazing and forested systems in temperate regions. The authors recognize this, but in the discussion and significance of the results present speculation. The stability of the systems, the long-term consequences of the observed changes in the food webs on carbon sequestration, ties to the green-brown imbalance hypothesis, and the implications of the results to management are brought up, but not discussed in much detail.

Methods and supplemental materials should be expanded. The portion of the methods section that describes how the energy fluxes were estimated draws from summary and synthesis papers, rather than primary sources. Including more details on the approach would help. Details on the design and sampling could be expanded as well. How many times were the plots sampled? The energy flux estimates are made under the assumption that the system is at steady-state. For studies like this multiple samples over a growing season are made to capture the steady-state. If a single sample date or multiple samples over narrow time window is taken then the time frame that the estimates are capturing is limited. For comparative purposes this is fine, but it does constrain the conclusions that can be made.

Referee #3 (Remarks to the Author):

The manuscript by Potapov et al. compares above- and belowground foodweb structure and fluxes across 4 different tropical land use systems.

Their main findings are that 1) in rainforest the belowground foodweb is larger in terms of biomass and energy flux than the aboveground foodweb, 2) compared, to rainforests, plantations have a similar energy flux, but somewhat less foodweb interactions, relatively less predators belowground, a higher dominance by earthworms and bacteria dominated carbon cycling.

The strengths of the manuscript are 1) a fascinating study system (foodwebs), 2) a rather complete description of the foodweb (both above and belowground, birds, canopy arthropods, and belowground arthropods and worms), and 3) the comparison of rainforest with common forms of commodity plantations.

The weaknesses are that 1) for an interested outsider it is rather difficult to follow what exactly is

being reported, 2) energy fluxes are inferred from biomass stocks rather than from measured fluxes, 3) biodiversity loss was assumed to underlie the patterns but was not measured, 4) relative short time intervals and soil volumes have been sampled, and the question how representative this is.

Please find my major and minor comments below. I hope they are of help to improve the manuscript.

With kind regards,
Lourens Poorter

MAJOR COMMENTS

1. ENERGY FLUX IS INFERRED RATHER THAN MEASURED. I am a tropical forest ecologists working on functional ecology of plants and land use change. I am therefore not familiar with foodweb research, but to be honest, I had an extremely difficult time to understand and grasp what is going on. As an interested reader I expect at least that a manuscript for Nature is directed to a generalist interested audience. So maybe this reflects my ignorance with the field, but I really do not understand how you can infer energy fluxes from standing stocks.

1.1. What I understand is that you first quantified abundance and fresh mass for each taxon (how many taxa were unidentified?),

1.2 then estimated for each taxon its feeding guild (for how many taxa is this unknown, and how many of these taxa feed on different food sources or guilds, and if so, how do you know the relative share of each food source/guild?).

1.3 I can imagine that based on this you can create an interaction food web, but up to here is for me what you really can say with your data. You then infer energy fluxes using a multichannel foodweb approach (I595-624) which is based on many unknown assumptions. I guess for this you need to know the metabolic rate based on the biomass. These equations are presented in Extended Data Table 3. But what is the r^2 of these equations, and if you backtransform this how much scatter do you get, so how reliable are these estimates?

1.4 Is it then assumed that the biomass of all these functional groups is constant? And based on the metabolic rate you assume they need to assume a certain amount of biomass needs to be consumed, assuming a specific conversion energy? And then consumed animals are replenished out of the blue because their biomass is assumed to be constant? And if a guild is omnivorous and feeds on several other guilds, how do you infer how much biomass was consumed from other guilds? Apparently you assume average body mass values for each guild, but how much body mass variation is there within guilds and how does that determine the fluxes? You see, I am totally lost, and I have a hard time to believe that you are inferring something meaningful. I trust your values for total biomass, and foodweb complexity (Fig. 3A,C) but not for total energy flux, predation consumption ratio, and bacteria/fungi ratio (Fig 3B,D,E). I personally would confine your story to biomass and foodweb complexity and forget about the rest,

2. UNCLEAR WHAT YOU ARE REPORTING. Again, I am a laymen, and unfortunately had really a hard time to follow what you did.

2.1 Use less jargon, define all better, and use easier terms. You talk about the 'green' and 'brown' foodweb. Why not simply 'aboveground' and 'belowground' (=litter+ground+soil) foodweb. You talk about 'strata'. I thought about forest strata but why not simply saying above/belowground. Why are resources only living plants, leaf litter, fungi, bacteria, soil organic matter, and not the animals that are predated? What is the 'total green energy flux' (I98). Do you sum up everything that these groups eat amongst themselves (so the sum of the energy flux of all pairwise interactions? It took me some time to realize that the 5 resources are consumed (Fig 1) and that each of these consumers is consumed by others. Explain this better with an example in the text. What is each dot in Fig 1. A taxonomic group (hornbills, warblers) or a functional group). What are they eating in this green web? Fruits? Leaves?

2.2 It would be nice if you would report the biomass of each of the 4 animal groups, and their energy flux in the same graph. Or did you show that somewhere and did I miss it?

3. BIODIVERSITY LOSS IS ASSUMED TO DRIVE PATTERNS BUT NEVER REPORTED. Your justification starts with the impact of biodiversity loss for foodwebs structure and resilience, and a lot of your explanations of the foodweb patterns found is based on biodiversity loss. You never report biodiversity values with your data. It would be nice if you could provide at least data on the richness of the taxa, how this varies across the 4 land use systems, as that would make the narrative more convincing and it would make it a more complete and round story

4. HOW REPRESENTATIVE AND ACCURATE ARE YOUR DATA. I have the feeling that the effort has been massive to collect these data, but the question is how accurate these are, as bird, arthropod density can vary over the seasons and year, and a lot in space.

4.1 You measured birds for each plot 3x20 minutes, and acoustically for 4 days. Although I appreciate that you included birds, this is REALLY very little (and I guess too little). I assume that frugivores are there when plants fruit in specific parts of the season, and that bird density and activity depend on the weather pattern, and that many birds can be migratory or less vocal when breeding. If I walk 4 days in my own neighbourhood watching birds, then the bird composition varies a lot over the weeks

4.2 Canopy arthropods were estimated with fogging. How high did your fog go? I assume your rainforest canopy is 30-40 m tall? I do not see how the sensitivity analysis that you report can check for species that you did not measure because the fog did not go high enough

4.3 Soil arthropods were measured in 3 subplots per plot (it is unclear if there are 3 samples per subplot). So you sampled 16cmwidth x 48 length x 5 cm deep. How is this going to be representative for a plot of 50x50 m?? I can imagine that the spatial turnover of these micro and meso organisms is tremendous. What is the pairwise sorenson similarity between the 3 samples? I really can not imagine how you can realistically quantify the soil arthropod community with such an approach (although the effort has been massive).

4.4 The biomass and energy flux of birds and arthropods is surprisingly low, you say (this is one of your main conclusions). So what about canopy and ground mammals such as monkeys, deer, reptiles, amphibians. I guess you have missed out a lot of the aboveground mass of these big animals, and hence, their energy fluxes.

5. WHAT ARE THE IMPLICATIONS? You show that compared to rainforest, plantations have a similar energy flux but a different foodweb structure. So what? Is this 'good' because the energy flux is the same? Is this 'bad' because the composition differs from the control? Are the systems 'worse' because they have less predators, and more worms and bacteria carbon cycling? Or are they simply different and are still functioning. ? Why would a foodweb with 280 interactions instead of 300 be less resilient? It still seems quite complex to me. So I am not sure what the implications are, apart from the fact that they function differently.

MINOR COMMENTS

1. Better describe your land use systems. What tree sizes? What densities. I have no clue what a jungle rubber system is.

2. L101. You state that the pervading idea is that tropical ecosystems are a green world. Why? Because most of the vegetation biomass is aboveground? Please explain and provide a reference. I thought that Odum (1969) Science had predicted that old growth forests are a brown detritus based food web

3. L242. Why does this only apply for rubber? Palms also have a more simple canopy structure

4. Rubber agroforests have a higher pH and therefore more worms. Were these rubber agroforests all by coincidence on a different soil type, or did the vegetation modify the soil conditions?

5. L538. Please report how many species/taxa and individuals you measured per plot (means and the range across plots). Do the same for the canopy arthropods and the soil arthropods and worms. That gives a better understanding about your data.

6. The discussion was sometimes difficult to follow. You report many patterns and give many different explanations. I got a bit lost. Please select your 3 main take home messages and focus on those

Author Rebuttals to Initial Comments:

Referees' comments and replies

Referee #1 (Remarks to the Author):

One of the major threats to global biodiversity is the transformation of tropical rainforests into agriculture-oriented land-use systems like agroforestry and mono-culture plantations. Obviously, the effects of such land-use changes are numerous. Land-use change has direct effects on the aboveground vegetation and primary productivity, and along with this on the belowground productivity in terms of altered root turn-over and root exudates. These changes in productivity will affect almost all groups of organisms at various trophic levels, including herbivores and predators aboveground, and microbes and fauna belowground, and along with this the total energy that comes available for the above- and belowground food webs with various consequences for the fluxes among the trophic groups within the food webs. Hence, observing, measuring and understanding the complex of changes along land-use change is an enormous task, and hence offers at the same time a great scientific challenge. The study by Potatov takes up this challenge by taking an ecosystem approach. The study focuses on the aboveground as well as the belowground ecosystem compartment and the linkages between the two. The study also includes the analysis of the effects on the above- and belowground food web structure and function by measuring population sizes of various groups of organisms and inferring the fluxes among these groups using metabolic parameters. The study focuses on three kinds of land-use transformations, i.e., from tropical forest into rubber agroforestry, and into monoculture plantations of rubber and oil palm. The paper is very well written and has some nice clarifying figures. The results are scientifically interesting and societally relevant.

Given this the manuscript has the potential to be published in a high-quality scientific journal. To make the paper acceptable for publication in Nature I have though the following questions, suggestions and recommendations to revise the paper.

Response: Thank you for this evaluation and for highlighting the relevance of our paper, which we very much appreciate. We have thoroughly addressed all your helpful points below.

1. The choice of selecting (only) two groups of organisms per ecosystem compartment as representative examples, i.e., arthropods and birds aboveground and arthropods and earthworms belowground is understandable, one cannot measure everything. But I think these particular choices can be more strongly justified. For example, earthworms are an excellent choice as they occupy a central position in belowground ecosystem functioning ('ecosystem engineers') as well as in the soil food webs by eating almost everything ('ecophores'). Also, the choice for arthropods makes sense as most arthropods can be found at various trophic levels, e.g., microbivores, herbivores, predators, and within these trophic levels they can have a very diverse diet. In this way composition and abundances of the arthropods may also reflect the composition of the food web as a whole.

Response: Thank you for pointing this out. We now justify our animal group choices in the introduction (LL 85-89): “Our group selection represents the majority of the animal biomass (arthropods, earthworms), including ecosystem engineers (earthworms, ants) and animals at different trophic levels – from detritivores, microbivores, and herbivores, to top predators (e.g., spiders, birds) – thus well representing the composition of the food web as a whole”

2. Some hypotheses, including the main one, are a little unsatisfactory formulated. Instead of an ‘un-directional’ formulation in terms of ‘differences’ I recommend formulations in terms of a priori predictions or ‘increases’ or ‘decreases’ as much as possible.

Response: Our main hypothesis was intentionally formulated generically due to the complexity of our design and little room for justification of all hypotheses in the general introduction. However, we agree with the reviewer that specification of hypotheses is helpful and will better guide readers. We now added ‘directions’ in the main hypothesis (LL 93-97): “Our main hypothesis was that there are different keystone animal groups that channel most of the energy in rainforest and plantations, and that energy distribution changes with land use across strata (more energy allocated to aboveground food webs in plantations), trophic levels (low predation in plantations) and basal resources (disproportionally high herbivory in plantations).” Please, note that we provide justifications and specific hypotheses in the sections “Green to brown shift with habitat transformation”, “Predation decline with rainforest transformation”, and “Changes in trophic functions with habitat transformation”. We now also specified the hypothesis in the latter section (LL 285-288): “We hypothesised that the dominant trophic functions would change with land use, indicating different carbon pathways at the ecosystem scale – specifically, we expected higher utilisation of primary basal food resources (i.e., plants and detritus) in plantations due to lower microbial biomass.”

3. In Figure 2. we see a large increase in earthworms after transformation, ‘compensating’ for a large decrease in arthropods. I did not find an explanation for this decrease, but maybe I overlooked something. Has this something to do with disturbance?

Response: We now added potential explanations for the observed arthropod decline (LL 193-198): “The increase in the earthworm-associated energy flux was mirrored by a decline in the soil arthropod-associated energy flux (Fig. 1). Earthworms may negatively affect soil and litter arthropods through direct and indirect trophic relationships and environmental modifications, but the arthropod decline may also have been due to reduced leaf litter input and reduced soil organic carbon and nitrogen in plantations.”

4. Table 1. For tables in the main text, I prefer that they give more specific information in terms of which group, what biomass or energy flux. Levels of significance might be given in the text, so that such a table can be part of the supplementary materials/extended data.

Response: We agree that the table is not very informative in the main text. We did the following: (1) Table 1 was moved to the supplement (new Extended Data Table 2); (2) We added results of the same analysis from the validation survey to this table (details about the

validation survey are provided in the response to reviewer #3); (3) significance is now reported in the text.

5. Regarding the predator-declines, the observed increase in mono-culture rubber is seen as an exception. But isn't it one out of only 3 (land-use changes), or even one out of 2 (monocultures)?

Response: Indeed, our results cover too few crops for such generalisations. In the revised text, we do not refer anymore to predation in rubber monocultures as an "exception" and focussed our conclusions on the different outcomes of land-use types for trophic functions (LL 341-345): "...(3) land-use change is associated with a decline in predation and an increase in relative herbivory both above- and belowground in jungle rubber and oil palm, however, the high predation in rubber suggests that crop choices can have predictable outcomes for trophic functions in food webs"

6. Finally, the results are clearly described and nicely presented in the figures. The (possible) explanations are adequate. There is something though that I miss in the explanation and that is that it is likely that the land transformations have led to fast, and almost direct changes aboveground, and slow and indirect changes belowground. Aboveground, the vegetation is replaced quickly and so will be the faunal groups that depend on the vegetation. Belowground, rapid changes can be expected in the rhizosphere through altered root and exudate composition, and in the composition of the rhizosphere organisms, but changes in the detrital food web may take much longer. The age of the transformed systems (8-17 years) seems adequate to have a complete transformation aboveground but maybe not for the belowground compartment. Hence, the aboveground compartment may reflect the present, but the belowground compartment may reflect also the history. This might relate to some observed effect I think, such as the finding that reduction in the total energy flux goes along with losses in the aboveground biodiversity, and not (or less) in the belowground biodiversity. And also, that the brown food web showed a higher resilience than green food webs. I am not sure, but it might be interesting when the authors take this notion into account.

Response: Thank you for this interesting suggestion. Indeed, high resilience and legacy effects can play a role in the observed different responses of brown vs green food webs. We now discuss this in the main text (LL 223-226): "This finding may be due to a delayed impact of land-use change on below- than on aboveground biodiversity, as recently shown for temperate grasslands, which could be explained by legacy effects due to high inertia of soils"; (LL 360-363): "Improving belowground habitat structure via mulching and reducing herbicide use could be sufficient to partly restore soil biodiversity and energetic balance in belowground food webs. However, it may take time for the effects of these measures to become visible due to high historical inertia of the soil system."

Peter de Ruiter

Referee #2 (Remarks to the Author):

The paper presents an observational and comparative study to construct partial food webs within native and managed tropical forests in Indonesia and infer energy fluxes among groups of organisms within the systems.

The paper presents multiple goals and questions: 1) description of the aboveground and belowground portions of the plant-arthropod-earthworm-bird components of the native and managed forests, that include estimates of energy flow within the systems based on estimates of the biomasses of organisms sampled, 2) a comparison of the webs and estimates across the different forest and management types, and 3) a generalization of the implications of their findings.

The results are incremental in terms of the impacts of the findings on food web responses to forest management and land use.

1. The paper uses a familiar energetic modeling framework to estimate energy flow. The results reveal some interesting and familiar outcomes. First, the authors note that tropical systems are known for their diversity and productivity. The energy flow descriptions revealed "... that most of the energy in rainforests is channeled to the brown animal web." This is consistent with what has been reported in most terrestrial ecosystems. Likewise, the loss of both predators and total energy flux to higher trophic levels, the shift from the 'slower' plant and fungal to the 'fast' plant and bacterial channels, and the internal reallocation of energy flow in the managed systems compared to the native forest is consistent with similar comparisons cropping, grazing and forested systems in temperate regions. The authors recognize this, but in the discussion and significance of the results present speculation. The stability of the systems, the long-term consequences of the observed changes in the food webs on carbon sequestration, ties to the green-brown imbalance hypothesis, and the implications of the results to management are brought up, but not discussed in much detail.

Response: We are grateful for the thorough evaluation of our study and the helpful comments. We agree that our study supports several patterns observed before in temperate ecosystems, such as the dominance of brown over green food webs, loss of predators, and increase in fast energy channels. However, these observations are novel for tropical food webs and we also report novel and nuanced patterns: (1) Despite it being known that >80% of primary production is channelled as detritus in forests¹, and that biomass of (belowground) microbes greatly exceeds that of animals², the energy going into green (canopy) vs brown *animal* food webs in the tropics, to our knowledge, has not been quantified; (2) The loss of predators is not universal, but strongly depends on the land-use choices, which opens the possibility for manipulations/predictions of energy fluxes and trophic functions. We also show that green and brown food webs may respond very differently to land-use changes. While green food webs suffer energetic losses with little restructuring in terms of trophic functions, brown food webs retain the same amount of energy (over the time of investigation), while having distinct trophic functioning. We now highlight these points in the introduction (LL 84-85 and 92-93), and conclusions (LL 333-374). In the discussion, we mention similarities to previous research, but focus on novel

aspects. We also considerably expanded the section with implications of our results in the conclusions (LL 350-374).

1. Just Cebrian, 'Patterns in the Fate of Production in Plant Communities', *The American Naturalist* 154, no. 4 (1999): 449–68, <https://doi.org/10.1086/303244>.
2. Yinon M. Bar-On, Rob Phillips, and Ron Milo, 'The Biomass Distribution on Earth', *Proceedings of the National Academy of Sciences* 115, no. 25 (2018): 6506–11, <https://doi.org/10.1073/pnas.1711842115>.

2. Methods and supplemental materials should be expanded. The portion of the methods section that describes how the energy fluxes were estimated draws from summary and synthesis papers, rather than primary sources. Including more details on the approach would help. Details on the design and sampling could be expanded as well. How many times were the plots sampled? The energy flux estimates are made under the assumption that the system is at steady-state. For studies like this multiple samples over a growing season are made to capture the steady-state. If a single sample date or multiple samples over narrow time window is taken then the time frame that the estimates are capturing is limited. For comparative purposes this is fine, but it does constrain the conclusions that can be made.

Response: We agree with these points. We expanded the methods and the supplementary materials to detail the assumptions and steps behind the food-web reconstruction and energy flux estimations. See details in our responses to Reviewer #3. We also provided the full R code allowing for reproduction of these calculations. Our main analysis is based on the one-time-point sampling in 2013. We believe that even a snapshot can capture the strong differences among the studied land-use systems. Seasonal dynamics are less pronounced in the tropics than in temperate ecosystems and most invertebrate groups are active throughout the year with limited density variation^{3,4}. However, to account for the limited temporal scope of our study, we now repeated our analysis using another dataset from a subsequent survey four years after the main survey (in 2016-2017). This analysis does not include jungle rubber because this system was not assessed in the second 'validation' survey. This survey showed overall lower animal biomasses, but validated energetic dominance of the brown over the green energy channel, canopy arthropods over birds, energetic decline in canopies and reallocation of energy to belowground food webs in plantations, and shifts in trophic functions. However, in contrast to the main survey, in our validation survey we were not able to detect a significant reduction in the number of trophic interactions in plantation systems (except oil palm belowground). This is why we removed this pattern from the abstract and the conclusions. This information is now briefly presented in the text (LL 324-330) "This validation survey showed lower estimates of the absolute biomass and energy flux, but validated energetic dominance of the belowground over the aboveground energy flux, canopy arthropods over birds, energetic decline in canopies, reallocation of energy to belowground food webs in plantations, and shifts in trophic functions, but did not validate the general loss trophic links across above- and belowground compartments (Extended Data Text 1; Extended Data Figs. 5, 6 and 7; Extended Data Table 3).", and details are provided in the supplement including new Extended Data Fig. 5. In addition, we now mention potential soil legacy effects in the discussion and the conclusions (LL 223-226 and 360-363). Since we focus on comparisons rather than absolute values, we believe that our conclusions are robust.

3. Valentyna Krashevska et al., 'Land Use Change Shifts and Magnifies Seasonal Variations of the Decomposer System in Lowland Tropical Landscapes', *Ecology and Evolution* 12, no. 6 (2022), <https://doi.org/10.1002/ece3.9020>.
4. Amanda Mawan et al., 'Response of Arboreal Collembola Communities to the Conversion of Lowland Rainforest into Rubber and Oil Palm Plantations', *BMC Ecology and Evolution* 22, no. 1 (2022): 144, <https://doi.org/10.1186/s12862-022-02095-6>.

Referee #3 (Remarks to the Author):

The manuscript by Potapov et al. compares above- and belowground foodweb structure and fluxes across 4 different tropical land use systems.

Their main findings are that 1) in rainforest the belowground foodweb is larger in terms of biomass and energy flux than the aboveground foodweb, 2) compared, to rainforests, plantations have a similar energy flux, but somewhat less foodweb interactions, relatively less predators belowground, a higher dominance by earthworms and bacteria dominated carbon cycling.

The strengths of the manuscript are 1) a fascinating study system (foodwebs), 2) a rather complete description of the foodweb (both above and belowground, birds, canopy arthropods, and belowground arthropods and worms), and 3) the comparison of rainforest with common forms of commodity plantations.

The weaknesses are that 1) for an interested outsider it is rather difficult to follow what exactly is being reported, 2) energy fluxes are inferred from biomass stocks rather than from measured fluxes, 3) biodiversity loss was assumed to underlie the patterns but was not measured, 4) relative short time intervals and soil volumes have been sampled, and the question how representative this is.

Response: Thank you for the thorough assessment and suggestions on how to improve our manuscript. We appreciate that you positively scored our comprehensive assessment of tropical communities, which we believe is among unique features of the study. Below we provide comments on the mentioned weaknesses of the manuscript:

(1) We now clarify our methodological approach early in the introduction (LL 102-108): "We further used these data to reconstruct food webs at each site, comprising 62 trophic guilds across all animal groups (Extended Data Table 1), and used metabolic regressions to estimate metabolic rates of each guild per unit biomass. Steady-state food-web modelling, which assumes that energetic losses are compensated by lower trophic levels, was used to calculate energy fluxes and to quantify the distribution of energy and consumption of different resources (living plants, litter, bacteria, fungi, soil organic matter, other animals) in aboveground and belowground food webs". We also provided additional clarifications throughout the text and in the methods to make our story easier to follow.

(2) Our study investigates animal food webs and it was not feasible to have an ecosystem-scale measurement of animal respiration and animal-driven processes such as total and

group-specific predation, herbivory, microbivory etc. without inference of energy fluxes from animal biomasses. We are not aware of methods allowing for in-situ measurements of animal energy fluxes at this scale. Metabolic regressions is a generic approach based on extensive empirical data, which is now widely used to quantify food-web functioning across ecosystems and wide ranges of body sizes/taxa (some literature is listed below).

3) We did not report biodiversity loss in our study since it has been repeatedly reported in previous publications, also from our study sites. We provided a list of selected references below. Unfortunately, species-level identification is not available for about half of the arthropod taxa included in our food-web calculations. This is ongoing work that for our region, takes years, if not decades, of taxonomic effort. We added quantification of already reported animal biodiversity declines to the text (LL 203-206): “Tropical land-use change has been found to result in 20-70% decline in species richness in arthropods, birds, and other taxa. Our findings show that this species decline is accompanied by fundamental functional changes in food webs and energy distribution in tropical ecosystems”.

4) Indeed, our main analysis is based on a snapshot animal assessment. To test the robustness of our results, we now repeated our analysis using an independent dataset from the ‘validation survey’ four years after the main survey. Details are described below and in the responses to Reviewer #2.

Detailed replies to all comments are given below.

Literature supporting the metabolic ecology/regressions approach:

- James H. Brown et al., ‘Toward a Metabolic Theory of Ecology’, *Ecology* 85, no. 7 (2004): 1771–89, <https://doi.org/10.1890/03-9000>.
- Richard M. Sibly, James H. Brown, and Astrid Kodric-Brown, *Metabolic Ecology: A Scaling Approach* (UK: Wiley-Blackwell, 2012).
- Andrew D. Barnes et al., ‘Energy Flux: The Link between Multitrophic Biodiversity and Ecosystem Functioning’, *Trends in Ecology & Evolution* 33, no. 3 (2018): 186–97, <https://doi.org/10.1016/j.tree.2017.12.007>.
- A. D. Barnes et al., ‘Biodiversity Enhances the Multitrophic Control of Arthropod Herbivory’, *Science Advances* 6, no. 45 (2020): eabb6603, <https://doi.org/10.1126/sciadv.abb6603>.

Selected literature showing species diversity decline in monoculture plantations (also now cited in the results and the methods):

- Ingo Grass et al., ‘Trade-Offs between Multifunctionality and Profit in Tropical Smallholder Landscapes’, *Nature Communications* 11, no. 1 (2020): 1186, <https://doi.org/10.1038/s41467-020-15013-5>.
- Anton M. Potapov et al., ‘Functional Losses in Ground Spider Communities Due to Habitat-Structure Degradation under Tropical Land-Use Change’, *Ecology* 101, no. 3 (2020): e02957, <https://doi.org/10.1002/ecy.2957>.
- Winda Ika Susanti et al., ‘Conversion of Rainforest into Oil Palm and Rubber Plantations Affects the Functional Composition of Litter and Soil Collembola’, *Ecology and Evolution* 11 (2021): 10686–708, <https://doi.org/10.1002/ece3.7881>.
- Amanda Mawan et al., ‘Response of Arboreal Collembola Communities to the Conversion of Lowland Rainforest into Rubber and Oil Palm Plantations’, *BMC Ecology and Evolution* 22, no. 1 (2022): 144, <https://doi.org/10.1186/s12862-022-02095-6>.

- Daniel Ramos et al., 'Rainforest Conversion to Rubber and Oil Palm Reduces Abundance, Biomass and Diversity of Canopy Spiders', *PeerJ* 10 (2022): e13898, <https://doi.org/10.7717/peerj.13898>.
- Rizky Nazarreta et al., 'Rainforest Conversion to Smallholder Plantations of Rubber or Oil Palm Leads to Species Loss and Community Shifts in Canopy Ants (Hymenoptera: Formicidae)', *Myrmecological News*, 2020, https://doi.org/10.25849/MYRMECOL.NEWS_030:175.
- Azru Azhar et al., 'Rainforest Conversion to Cash Crops Reduces Abundance, Biomass and Species Richness of Parasitoid Wasps in Sumatra, Indonesia', *Agricultural and Forest Entomology*, 2022, afe.12512, <https://doi.org/10.1111/afe.12512>.
- Walesa Edho Prabowo et al., 'Bird Responses to Lowland Rainforest Conversion in Sumatran Smallholder Landscapes, Indonesia', *PLOS ONE* 11, no. 5 (2016): e0154876, <https://doi.org/10.1371/journal.pone.0154876>.
- Kasmiatun et al., 'Rainforest Conversion to Smallholder Cash Crops Leads to Varying Declines of Beetles (Coleoptera) on Sumatra', *Biotropica* 55, no. 1 (2023): 119–31, <https://doi.org/10.1111/btp.13165>.

Please find my major and minor comments below. I hope they are of help to improve the manuscript.

With kind regards,
Lourens Poorter

MAJOR COMMENTS

1. ENERGY FLUX IS INFERRED RATHER THAN MEASURED. I am a tropical forest ecologist working on functional ecology of plants and land use change. I am therefore not familiar with foodweb research, but to be honest, I had an extremely difficult time to understand and grasp what is going on. As an interested reader I expect at least that a manuscript for *Nature* is directed to a generalist interested audience. So maybe this reflects my ignorance with the field, but I really do not understand how you can infer energy fluxes from standing stocks.

Response: We estimated fluxes from the standing biomass stocks and food-web topology (potential energy transfers). This approach is commonly referred to as 'steady-state food-web modelling' and described in detail in recent literature. These literature references are provided here and in the manuscript text; we also answer more specific comments in our responses below and added details to the methods.

- Andrew D. Barnes et al., 'Energy Flux: The Link between Multitrophic Biodiversity and Ecosystem Functioning', *Trends in Ecology & Evolution* 33, no. 3 (2018): 186–97, <https://doi.org/10.1016/j.tree.2017.12.007>.
- Malte Jochum et al., 'For Flux's Sake: General Considerations for Energy Flux Calculations in Ecological Communities', *Ecology and Evolution*, 14 (2021): ece3.8060, <https://doi.org/10.1002/ece3.8060>.

1.1. What I understand is that you first quantified abundance and fresh mass for each taxon (how many taxa were unidentified?),

Response: Correct. In total, 418 birds, 366,975 canopy arthropods, and 50,401 soil invertebrates were observed/collected and included in the main survey. Fresh masses were taken from the literature for birds (species-specific), and calculated using measured individual body sizes from each sample and each taxon (trophic guild) and allometric size-mass regressions for arthropods and earthworms. This information is now detailed in the methods (sections on birds, canopy arthropods, and soil invertebrates). We have species-level identifications for approximately half of the taxa. However, all taxa were identified at least to the order level. Species-level identifications were not presented in our study because it is out of our scope and these data are presented elsewhere (please see the list of papers above). Besides, with most of the species being not scientifically described in several taxa (e.g. parasitoid wasps), we can tell little from the (morpho)species name about the ecology of species at the current state of knowledge.

1.2 then estimated for each taxon its feeding guild (for how many taxa is this unknown, and how many of these taxa feed on different food sources or guilds, and if so, how do you know the relative share of each food source/guild?).

Response: Correct. All animals were allocated to general feeding guilds following existing knowledge and literature, as well as empirical data on stable isotope composition published/under review elsewhere^{5,6} (see the sections on birds, canopy arthropods, and soil invertebrates in the methods). Feeding guilds were defined by feeding preferences (relative shares of each food source/prey) to the five main resources and predation capability. Since our feeding guild classification is relatively rough, in most cases we were able to assign large taxa to guilds. We extrapolated general knowledge on the trophic ecology of high-rank taxa (e.g. Chrysomelidae are herbivores, Staphylinidae and Araneae are predators, earthworms feed on soil, while Collembola feed on fungi) to all collected individuals within these taxa assuming phylogenetic signal in trophic niches and because information on the feeding preferences of most tropical invertebrate species is lacking. See ref 7 for justification of this approach. We now added more details on this to the methods (LL 628-632, 657-667 and 686-691). To test if our conclusions are robust to the relative shares of each food source in the diet, we ran a sensitivity analysis, which is presented in the Extended Data Fig. 1.

5. Zhou, Z. *et al.* Tropical land use alters functional diversity of soil food webs and leads to monopolization of the detrital energy channel, *ELife* 11 (2022): e75428, <https://doi.org/10.7554/eLife.75428>.
6. Pollierer, M. M. *et al.* Rainforest conversion to plantations fundamentally alters energy fluxes and functions in canopy arthropod food webs, *under revision in Ecology Letters*
7. Anton M. Potapov, Stefan Scheu, and Alexei V. Tiunov, 'Trophic Consistency of Supraspecific Taxa in Belowground Invertebrate Communities: Comparison across Lineages and Taxonomic Ranks', *Functional Ecology* 33, no. 6 (2019): 1172–83, <https://doi.org/10.1111/1365-2435.13309>.

1.3 I can imagine that based on this you can create an interaction food web, but up to here is for me what you really can say with your data. You then infer energy fluxes using a multichannel foodweb approach (1595-624) which is based on many unknown assumptions. I guess for this you need to know the metabolic rate based on the biomass. These equations are presented in Extended Data Table 3. But what is the r^2 of these equations, and if you backtransform this how much scatter do you get, so how reliable are these estimates?

Response: This is correct, the multichannel food-web reconstruction approach is based on several (simple) assumptions. These assumptions are based on food-web theory and were repeatedly supported in existing literature. Further details and references are now added to the methods “Foodweb reconstruction” section (see also the literature below; LL 712-732): “Generic rules of food-web reconstruction based on food-web theory were used to infer weighted trophic interactions among all nodes with the following assumptions: (1) There are phylogenetically inherited differences in feeding preferences for various basal resources and predation capability among soil animal taxa that define their feeding interactions (reflected as resource preferences in the raw data table); (2) predator–prey interactions are primarily defined by the optimum predator-prey mass ratio (PPMR) – typically, a predator is larger than its prey, but certain predator traits (hunting traits and behavior, parasitic lifestyle) can considerably modify the optimum PPMR. We measured body mass distribution overlap for each potential pair of predator and prey in each food web to determine the most plausible trophic interactions; (3) strength of the trophic interaction between predator and prey is defined by the overlap in their spatial niches related to vertical differentiation, with greater overlap leading to stronger interactions (i.e. no overlap among specialized canopy and soil arthropods); (4) predation is biomass-dependent – due to higher encounter rate, predators will preferentially feed on prey that are locally abundant; (5) strength of the trophic interaction between predator and prey can be considerably reduced by prey protective traits – prey with physical, chemical or behavioral protection are consumed less. All these assumptions are applied together to infer the most plausible trophic interaction matrix. For example, feeding preferences of omnivorous nodes to basal resources or other invertebrates were assigned based on literature (assumption 1), while prey selection among other invertebrates was based on size, spatial niche, total biomass, and protection of prey (assumptions 2-5).” The full food-web reconstruction R script is now given in the Supplement. Moreover, we test if our reconstruction assumptions affect our main conclusions using a sensitivity analysis (Extended Data Fig. 1) - and they do not. The r^2 values of metabolic regressions are typically high (>95%)⁸ if calculated for a wide range of body masses. Considering the wide size range of taxa included in our study (from micrograms to kilograms), we believe that this has a negligible effect on our results. We now added the standard errors for metabolic coefficients in the Extended Data Table 4 (Extended Data Table 3 in the initial version). These standard errors are generally small, except for mite groups (Mesostigmata, Oribatida, Prostigmata). However, mites accounted only for a very small proportion of energy flux in our study. Overall, we agree that our metabolic estimates can be biased especially in terms of absolute values, therefore, we focus on comparisons rather than on absolute values in our main conclusions.

8. James H. Brown et al., ‘Toward a Metabolic Theory of Ecology’, *Ecology* 85, no. 7 (2004): 1771–89, <https://doi.org/10.1890/03-9000>.

Literature justifying our food-web assumptions:

- Potapov, A. M. Multifunctionality of belowground food webs: resource, size and spatial energy channels. *Biol. Rev. Camb. Philos. Soc.* 97, 1691-1711 (2022).
- Pierre-Marc Brousseau, Dominique Gravel, and I. Tanya Handa, 'Trait Matching and Phylogeny as Predictors of Predator-Prey Interactions Involving Ground Beetles', ed. Maud Ferrari, *Functional Ecology* 32, no. 1 (2018): 192–202.
- Potapov, A. M., Scheu, S. & Tiunov, A. V. Trophic consistency of supraspecific taxa in below-ground invertebrate communities: Comparison across lineages and taxonomic ranks. *Funct. Ecol.* 33, 1172–1183 (2019).
- Brose, U. et al. Foraging theory predicts predator-prey energy fluxes. *J. Anim. Ecol.* 77, 1072–1078 (2008).
- Owen L. Petchey et al., 'Size, Foraging, and Food Web Structure', *Proceedings of the National Academy of Sciences* 105, no. 11 (2008): 4191–96.
- Brose, U. et al. Predator traits determine food-web architecture across ecosystems. *Nat Ecol Evol* 3, 919–927 (2019).
- Gauzens, B. et al. fluxweb : An R package to easily estimate energy fluxes in food webs. *Methods Ecol. Evol.* 10, 270–279 (2019).
- Peschel, K., Norton, R., Scheu, S. & Maraun, M. Do oribatid mites live in enemy-free space? Evidence from feeding experiments with the predatory mite *Pergamasus septentrionalis*. *Soil Biology and Biochemistry* 38, 2985–2989 (2006).

1.4 Is it then assumed that the biomass of all these functional groups is constant? And based on the metabolic rate you assume they need to assume a certain amount of biomass needs to be consumed, assuming a specific conversion energy? And then consumed animals are replenished out of the blue because their biomass is assumed to be constant? And if a guild is omnivorous and feeds on several other guilds, how do you infer how much biomass was consumed from other guilds? Apparently you assume average body mass values for each guild, but how much body mass variation is there within guilds and how does that determine the fluxes? You see, I am totally lost, and I have a hard time to believe that you are inferring something meaningful. I trust your values for total biomass, and foodweb complexity (Fig. 3A,C) but not for total energy flux, predation consumption ratio, and bacteria/fungi ratio (Fig 3B,D,E). I personally would confine your story to biomass and foodweb complexity and forget about the rest,

Response: The 'steady-state food-web modelling' approach is based on the assumption that energetic losses (metabolic rates and losses due to predation) are compensated by lower trophic levels in account for food assimilation efficiencies. We now detailed our explanations in the methods (LL 744-754): "The energy flux to each node was calculated from per-biomass metabolism, accounting for assimilation efficiencies (proportion of energy from food that is metabolized by the consumer) and losses to predation assuming a steady-state energetic system (i.e., energetic losses from each node are compensated by the lower trophic levels) ... Then, we applied the *fluxing* function to the reconstructed interaction networks which delivered energy flux estimations among all food-web nodes." We also introduce this approach with our hypotheses in the introduction (LL 102-106): "We further used these data to reconstruct food webs at each site, comprising 62 trophic guilds across all animal groups (Extended Data Table 1), and used metabolic regressions to estimate metabolic rates of each guild per unit biomass. Steady-state food-web modelling, which

assumes that energetic losses are compensated by lower trophic levels, was used to calculate energy fluxes...”.

For omnivorous guilds, we assign food-web preferences based on the assumptions listed above (the “Foodweb reconstruction” section). Details are added to the methods (LL 729-732): “For example, feeding preferences of omnivorous nodes to basal resources or other invertebrates were assigned based on literature (assumption 1), while prey selection among other invertebrates was based on size, spatial niche, total biomass, and protection of prey (assumptions 2-5).”

Indeed, we use the average body mass values for each guild, but we also account for body mass variations. Body masses and their variations were measured for each taxon/guild in each plot/food web and were used to determine potential predator-prey interactions (using Predator-Prey Mass Ratios). We now specify this in the Foodweb reconstruction section (LL 719-721): “We measured body mass distribution overlap for each potential pair of predator and prey in each food web to determine the most plausible trophic interactions”.

To test the validity of our steady-state modelling approach and account for the limited time scope of our study, we now repeated our analysis using an independent dataset from the ‘validation survey’ four years later after the main survey. This survey showed overall similar results to our main survey. Details are provided in the response to Reviewer #2, in the text and supplement. We thus believe that our energy-flux based conclusions are robust and would like to keep them.

2. UNCLEAR WHAT YOU ARE REPORTING. Again, I am a laymen, and unfortunately had really a hard time to follow what you did.

2.1 Use less jargon, define all better, and use easier terms. You talk about the ‘green’ and ‘brown’ foodweb. Why not simply ‘aboveground’ and ‘belowground’ (=litter+ground+soil) foodweb. You talk about ‘strata’. I thought about forest strata but why not simply saying above/belowground. Why are resources only living plants, leaf litter, fungi, bacteria, soil organic matter, and not the animals that are predated? What is the ‘total green energy flux’ (I98). Do you sum up everything that these groups eat amongst themselves (so the sum of the energy flux of all pairwise interactions? It took me some time to realize that the 5 resources are consumed (Fig 1) and that each of these consumers is consumed by others. Explain this better with an example in the text. What is each dot in Fig 1. A taxonomic group (hornbills, warblers) or a functional group). What are they eating in this green web? Fruits? Leaves?

Response: Thank you for these suggestions. We aligned terminology in the text to make it easier to read. Initially, we used ‘green’ and ‘brown’ terms as concise and intuitive synonyms for ‘aboveground’ and ‘belowground’ to save space. However, we agree that uniform use of a single term will make the study clearer. Thus, we now use ‘aboveground’ and ‘belowground’ throughout the text with few exceptions. We would like to keep ‘green’ and ‘brown’ in the title, because otherwise our title will not fit the Nature formatting requirements (75 characters with spaces). We also kept ‘green’ and ‘brown’ if they were used as special terms (e.g. ‘green-brown imbalance hypothesis’). Finally, we also kept ‘green’ and ‘brown’ in the abstract and introduction to synonymise these terms to aboveground’ and ‘belowground’. Similarly, we replaced ‘strata’ with above- and belowground compartments throughout the text. We also now define ‘aboveground/belowground energy flux’ in the text (LL 214-216).

We use living plants, leaf litter, fungi, bacteria, soil organic matter as the main 'basal resources', linked to different ecosystem-level processes (primary/secondary production, C transformation, nutrient cycling). We now listed animals as another 'resource' in the introduction (LL 107-108) and replaced 'Resources' by 'Basal resources' in Figure 1, other figure captions, and throughout the text. We further edited Fig. 1 to improve clarity. Resources and trophic level are labelled now in the image. In Fig. 1 each point represents a trophic guild. Taxonomic names are given as examples. We added further explanation to the caption. Feeding on specific plant organs is not specified in our model. This trophic function (herbivory) represents consumption of living plant/algae material, i.e., contribution of the primary production to the energy flux.

2.2 It would be nice if you would report the biomass of each of the 4 animal groups, and their energy flux in the same graph. Or did you show that somewhere and did I miss it?

Response: These data are visualised in Fig. 2 using the trophic network figure (line size = fluxes, node size = biomasses) and are given in numbers for rainforest in the text (section 'Aboveground and belowground rainforest food webs'). Total above- and belowground energy fluxes and biomasses across the four land-use systems are presented in Fig. 3 panels a b. We would like to avoid adding more illustrations to stay concise.

3. BIODIVERSITY LOSS IS ASSUMED TO DRIVE PATTERNS BUT NEVER REPORTED. Your justification starts with the impact of biodiversity loss for foodwebs structure and resilience, and a lot of your explanations of the foodweb patterns found is based on biodiversity loss. You never report biodiversity values with your data. It would be nice if you could provide at least data on the richness of the taxa, how this varies across the 4 land use systems, as that would make the narrative more convincing and it would make it a more complete and round story

Response: As we described in our first response, biodiversity loss in plantations in comparison to rainforest has been repeatedly reported in previous publications, also from our study sites. While we have species-level data for about a half of the studied animal groups, we believe that reporting these data is beyond the scope of our paper. Illustrating biodiversity loss with changes in the number of trophic guilds will be misleading since (1) trophic guild definition does not align well with taxonomic classifications; (2) virtually all trophic guilds in our study are present across land-use systems (but are changing in biomass and many of them have lower species richness in plantations). We now specified observed species richness losses (in %) in the text, but prefer to keep this information in the form of references to published literature.

4. HOW REPRESENTATIVE AND ACCURATE ARE YOUR DATA. I have the feeling that the effort has been massive to collect these data, but the question is how accurate these are, as bird, arthropod density can vary over the seasons and year, and a lot in space.

4.1 You measured birds for each plot 3x20 minutes, and acoustically for 4 days. Although I appreciate that you included birds, this is REALLY very little (and I guess too little). I assume

that frugivores are there when plants fruit in specific parts of the season, and that bird density and activity depend on the weather pattern, and that many birds can be migratory or less vocal when breeding. If I walk 4 days in my own neighbourhood watching birds, then the bird composition varies a lot over the weeks

Response: We agree that our data cannot represent all bird species at the study sites. Unfortunately, gathering more detailed data was not feasible during the main survey. However, functional composition of communities is typically more stable than species composition⁹; i.e. despite species turnover, different bird species will perform similar roles in the food web as large-sized generalistic invertebrate feeders, or plant (seed/fruit) feeders. Thus, we believe that it is justified to retain birds in our food web model. To test the robustness of our findings, we analysed data from an independent 'validation survey', which showed overall similar results (Extended Data Text 1).

9. Anton M. Potapov et al., 'Functional Losses in Ground Spider Communities Due to Habitat-Structure Degradation under Tropical Land-Use Change', *Ecology* 101, no. 3 (2020): e02957, <https://doi.org/10.1002/ecy.2957>.

4.2 Canopy arthropods were estimated with fogging. How high did your fog go? I assume your rainforest canopy is 30-40 m tall? I do not see how the sensitivity analysis that you report can check for species that you did not measure because the fog did not go high enough

Response: According to our visual observations, fogging probably was very efficient in plantations, but could not adequately reach the highest canopies in rainforest. We didn't conduct a stratified fogging and thus cannot report community turnover in the canopies. We agree that our sensitivity analysis cannot account for vertical turnover of functional groups in canopies, however, existing studies showed overall comparable arthropod composition within rainforest canopies¹⁰. We now mention this limitation in the results (LL 124-127): 'This analysis suggested that the real energy flux aboveground (assuming uniform community composition within canopies) could be 62.0 ± 24.5 mW m⁻² in the most-severe undersampling scenario...'. This limitation, however, cannot undermine our main conclusions because it could still not explain the 14-fold above-belowground difference in energy flux.

10. Roman J Dial et al., 'Arthropod Abundance, Canopy Structure, and Microclimate in a Bornean Lowland Tropical Rain Forest', *Biotropica* 38, no. 5 (2006): 643–52.

4.3 Soil arthropods were measured in 3 subplots per plot (it is unclear if there are 3 samples per subplot). So you sampled 16cmwidth x 48 length x 5 cm deep. How is this going to be representative for a plot of 50x50 m?? I can imagine that the spatial turnover of these micro and meso organisms is tremendous. What is the pairwise sorenson similarity between the 3 samples? I really can not imagine how you can realistically quantify the soil arthropod community with such an approach (although the effort has been massive).

Response: Correct, each plot was represented by three soil samples (one per subplot), 16x16 cm in surface area. Despite this area being rather small in comparison to the size of the plot, such sampling schemes are very common in soil ecology due to high densities of soil invertebrates^{11,12}. For example, in previous studies, such a sampling design was sufficient to show systematic differences between, and similarities within land-use systems - both in tropical and temperate ecosystems (see list of references below). This is even more applicable in our case, since we look at the functional, rather than taxonomic composition of communities and we indeed see clear differences among land-use systems in our study. While we have heterogeneity within each plot (low pseudoreplication), relatively large true replication (n = 8 plots per system) compensate for this heterogeneity in our cross-land-use-type comparison. Which is confirmed by our validation survey.

11. Patrick Lavelle et al., 'Soil Macroinvertebrate Communities: A World-wide Assessment', *Global Ecology and Biogeography* 31, no. 7 (2022): 1261–76, <https://doi.org/10.1111/geb.13492>.
12. Anton M. Potapov et al., 'Global Monitoring of Soil Animal Communities Using a Common Methodology', *Soil Organisms* 94, no. 1 (2022): 55–68, <https://doi.org/10.25674/so94iss1id178>.

Literature comparing different land-use systems with low per plot replication:

- Maria A. Tsiafouli et al., 'Intensive Agriculture Reduces Soil Biodiversity across Europe', *Global Change Biology* 21, no. 2 (2015): 973–85, <https://doi.org/10.1111/gcb.12752>.
- Melanie M. Pollierer et al., 'Diversity and Functional Structure of Soil Animal Communities Suggest Soil Animal Food Webs to Be Buffered against Changes in Forest Land Use', *Oecologia*, (2021), <https://doi.org/10.1007/s00442-021-04910-1>.
- Winda Ika Susanti et al., 'Conversion of Rainforest into Oil Palm and Rubber Plantations Affects the Functional Composition of Litter and Soil Collembola', *Ecology and Evolution* 11 (2021): 10686–708, <https://doi.org/10.1002/ece3.7881>.
- Anton M. Potapov et al., 'Functional Losses in Ground Spider Communities Due to Habitat-Structure Degradation under Tropical Land-Use Change', *Ecology* 101, no. 3 (2020): e02957, <https://doi.org/10.1002/ecy.2957>.

4.4 The biomass and energy flux of birds and arthropods is surprisingly low, you say (this is one of your main conclusions). So what about canopy and ground mammals such as monkeys, deer, reptiles, amphibians. I guess you have missed out a lot of the aboveground mass of these big animals, and hence, their energy fluxes.

Response: We agree and discuss this limitation in our results and the 'Caveats' section (LL 160-166 and 318-323). Although we are missing several vertebrate groups, our assessment captures the energetic core of the community since arthropod biomass greatly exceeds that of vertebrates¹³, and arthropods also have a higher metabolism per body mass (smaller size). Despite the fact that we can't prove this using data from our sites, we are convinced that this conclusion is robust.

13. Yinon M. Bar-On, Rob Phillips, and Ron Milo, 'The Biomass Distribution on Earth', Proceedings of the National Academy of Sciences 115, no. 25 (2018): 6506–11, <https://doi.org/10.1073/pnas.1711842115>.

5. WHAT ARE THE IMPLICATIONS? You show that compared to rainforest, plantations have a similar energy flux but a different foodweb structure. So what? Is this 'good' because the energy flux is the same? Is this 'bad' because the composition differs from the control? Are the systems 'worse' because they have less predators, and more worms and bacteria carbon cycling? Or are they simply different and are still functioning. ? Why would a foodweb with 280 interactions instead of 300 be less resilient? It still seems quite complex to me. So I am not sure what the implications are, apart from the fact that they function differently.

Response: We believe that these changes are rather 'bad' since they are associated with previously reported losses in the total biodiversity and decline in many ecosystem functions (e.g. ref 14). Intuitively, a system where >50% of the total energy available to animals is channelled through invasive earthworms, is not 'healthy'. However, we prefer to avoid statements reflecting our attitude to the observed changes. Our results specifically point to the necessity of complex above-below assessments of (tropical) ecosystems to understand human effects. We focus on the energetic mechanisms and related functions that are associated with restructuring of biodiversity to inform predictions and guide actions. We now considerably expanded the last paragraph in conclusions focussing on potential implications of our results for ecosystem management (LL 350-374): "It is well documented that tropical land-use change results in animal biodiversity losses both above- and belowground. We show here that there are also associated changes in aboveground vs belowground food-web structure, basal resource consumption, and energy fluxes are distinctly different between the above- and belowground realm. We suggest that restoration and management practices in the tropics that alter the energetic balance across ecosystem compartments, taxa, size classes, and trophic levels, need to be more strongly considered and trialled. Plantations, especially oil palm, are very productive, but the available energy for maintaining multitrophic biodiversity is disproportionately low, which is associated with re-allocation of energy fluxes to basal trophic levels in belowground food webs. These findings reveal a large energetic potential for restoring animal biodiversity in plantations. Improving belowground habitat structure via mulching and reducing herbicide use could be sufficient to partly restore soil biodiversity and energetic balance in belowground food webs. However, it may take time for the effects of these measures to become visible due to high historical inertia of the soil system. Aboveground, measures directly affecting vegetation are needed. For example, increasing canopy complexity by planting trees within monoculture plantations, and designing diverse landscapes could provide more ecological niches likely resulting in re-allocation of more energy to aboveground food webs. In the absence of restoration measures, intensive tropical land use may foster earthworm invasion belowground, further depletion of soil organic stocks, and increase risks of aboveground pest outbreaks. This is likely to result in intensification of fertilizer, herbicide, and pesticide use. Experimental studies exploring the effect of restoration measures on the energy distribution and trophic functions of food webs across above- and belowground compartments of tropical ecosystems will be crucial for better management of the energy of tropical ecosystems to sustain tropical biodiversity and ecosystem services."

14. Claudia Dislich et al., 'A Review of the Ecosystem Functions in Oil Palm Plantations, Using Forests as a Reference System', *Biological Reviews* 92, no. 3 (2017): 1539–69, <https://doi.org/10.1111/brv.12295>.

MINOR COMMENTS

1. Better describe your land use systems. What tree sizes? What densities. I have no clue what a jungle rubber system is.

Response: We added information on the average tree height and tree density in each studied land-use system in the 'Study region and design' section.

2. L101. You state that the pervading idea is that tropical ecosystems are a green world. Why? Because most of the vegetation biomass is aboveground? Please explain and provide a reference. I thought that odum (1969) *Science* had predicted that old growth forests are a brown detritus based food web

Response: Rainforest canopies host diverse communities of vertebrate and invertebrate animals species and arthropod biomass (see e.g. 15). It is known that >80% of primary production is allocated to detritus in forest ecosystems. However, to our knowledge there is no comparison of energy flux in animal food webs in tropical aboveground versus belowground ecosystem compartments. We now added an explanation in the text (LL 117-118): "These figures are in contrast to the pervading idea of tropical ecosystems as a 'green world' with very high animal biomass in canopies."

15. Martin D. F. Ellwood and William A. Foster, 'Doubling the Estimate of Invertebrate Biomass in a Rainforest Canopy', *Nature* 429, no. 6991 (2004): 549–51, <https://doi.org/10.1038/nature02560>.

3. L242. Why does this only apply for rubber? Palms also have a more simple canopy structure

Response: We agree that simple canopy stature cannot explain the difference in predation between rubber and oil palm. Since the high predation in rubber canopies was mainly associated with a large biomass of diptera (blood-sucking gnats and mosquitoes), it may be explained by the presence of small water bodies (rubber sap collection buckets) in this system that can host dipteran larvae. However, this speculation needs to be tested in experiments. We added this alternative explanation in the text (LL 268-271).

4. Rubber agroforests have a higher pH and therefore more worms. Were these rubber agroforests all by coincidence on a different soil type, or did the vegetation modify the soil conditions?

Response: Increase in pH with rainforest transformation to agricultural systems are caused by ash inputs after rainforest burning and liming of plantation systems. Soil type was similar

across land-use systems in Harapan (loamy Acrisols) and Bukit Duabelas (clayey Acrisols) regions¹⁶

16. Kara Allen et al., 'Spatial Variability Surpasses Land-Use Change Effects on Soil Biochemical Properties of Converted Lowland Landscapes in Sumatra, Indonesia', *Geoderma* 284 (2016): 42–50, <https://doi.org/10.1016/j.geoderma.2016.08.010>.

5. L538. Please report how many species/taxa and individuals you measured per plot (means and the range across plots). Do the same for the canopy arthropods and the soil arthropods and worms. That gives a better understanding about your data.

Response: We now provide tables with overview of collected individuals for each animal group in the Supplementary material. We also report the total numbers of collected individuals in 2013 and 2016-2017 in the methods.

6. The discussion was sometimes difficult to follow. You report many patterns and give many different explanations. I got a bit lost. Please select your 3 main take home messages and focus on those

Response: We now added explanations in our results to improve clarity. We also simplified sections "Aboveground to belowground shift with habitat transformation" and "Predation decline with rainforest transformation". In our conclusions we give 4 main bullet points and now added a paragraph on implications of our results (LL 333-374).

Reviewer Reports on the First Revision:

Referees' comments:

Referee #1 (Remarks to the Author):

I am overall happy with the way the authors have addressed my comments.

Regarding the justification of the choice for microarthropods and earthworms, I am fine with that addition, but – very little remark - suggest to leave out 'well'; is somewhat 'safer' as groups like protists and nematodes are not in.

The reformulated hypotheses still have 'changes' and 'differences' in the formulation, but I like them much better, in particular (LL287-288) "...we expected higher utilisation of primary basal food resources (i.e., plants and detritus) in plantations due to lower microbial biomass.

Furthermore, I am happy with the explanations for the observed arthropod decline (LL 193-198), the changes in Table1/AppTable2 and the paragraph on possible legacy effects.

Referee #2 (Remarks to the Author):

The authors addressed many of the questions, concerns and suggests that were raised. There are at least three main threads in the paper. The first two are well presented, the third and most important, is weakly supported as presented.

The first thread focuses on the energetic organizations of birds and arthropods in a native tropical forest and three plantation types (former native forests) in the region. Methods and sample size notwithstanding, the dominance of the 'brown' versus green web in descriptions of these tropical systems is apparent and novel. The authors point out the novelty of the results in the revision and in their response to my comment in the review.

The second focuses on the differences and similarities in the energetic organization across the system types. The more interesting results being the shift in the away from belowground arthropods to the invasive earthworms. The title – Rainforest transformation reallocates energy from green to brown food webs – and summary section - in rainforest >90% of the total animal energy flux was channelled by arthropods in soil and canopy, while in plantations >50% of the energy was allocated to annelids (earthworms) are explicit on this, while the conclusion section (conclusion 3) is less so.

The third thread, and most important in terms of interest and impact, focuses on the potential consequences of these changes on carbon storage in tropical systems (a point raised in the summary - Our study shows that tropical land-use change drives consistent decline of multitrophic energy flux aboveground, and intensification of soil organic matter consumption joint [sic] with nearly complete elimination of energy flux to predators belowground, which coincide with previously reported soil carbon stocks depletion - and in conclusion 4 - belowground food webs in plantations rely on different basal resources than those in rainforest, promoting faster energy channels.) This is one of the more interesting conclusions as it links changes in the energetic organizations of the food webs

in the plantations to the faster energy channels to depleted carbon stocks in the systems. The connection being made to the faster energy channel is tenuous as presented. Given that this is a reconstructed food web parameterized to estimate energy flux via standing biomass and physiologies, from C flux, it certainly would be possible to provide some estimates of C respiration emanating from each of the belowground channels. This would strengthen the conclusion that the faster channel is operating in the plantations. Conclusion 4 further states that – [t]hese changes are associated with previously observed depletion of carbon stocks¹⁷, but the causal relationships here still need to be tested. What was the magnitude of the observed changes in carbon stocks, and could the observed shift belowground towards the earthworms account for the losses? Attempting to answer this question using the data and analytical approach in place would move the results beyond an interesting observation and comparative study to a more impactful study in terms of our understanding of food web responses to tropical forest management and land use.

Referee #3 (Remarks to the Author):

The manuscript by Potapov compares above- and belowground foodweb structure and fluxes across 4 different tropical land use systems. Their main findings are that 1) in rainforest the belowground foodweb is larger in terms of biomass and energy flux than the aboveground foodweb, 2) compared, to rainforests, plantations have a similar energy flux, but somewhat less foodweb interactions, relatively less predators belowground, a higher dominance by earthworms and bacteria dominated carbon cycling.

As I said in my earlier review, the strengths of the manuscript are 1) a fascinating study system (foodwebs), 2) a rather complete description of the foodweb (both above and belowground, birds, canopy arthropods, and belowground arthropods and worms), and 3) the comparison of rainforest with common forms of commodity plantations.

The revised version of the manuscript has been much improved, is interesting, and a pleasure to read. The description of the food web construction and energy flux estimates is much more clear, the overarching hypothesis and the detailed hypothesis per section are much better explained and justified, the structure is easier to follow through the headers and more clear take home messages, and I especially appreciate that it is now more clearly explained what forest conversion towards commodity plantation means for ecosystem functioning, and how the obtained insights on energy fluxes can be more hands on used to improve plantation design and management to mitigate the negative effects. I congratulate the authors therefore with the overall result.

I am not a food network specialist so I hope that the other two reviewers really have checked if the network construction is fine and if the assumptions are solid (their reviews were rather brief and more focused on the interpretation of the results than on the methods).

I found the responses to my comments overall satisfactory and convincing, but it was often unclear what exact changes were made in the manuscript. Please put below each of my previous comments

with quotes how and where you changed things in the manuscript. You should not only convince me in the cover letter, but also the future readers of the manuscript, so make sure that these considerations/justifications are included in the revised manuscript. For example. Say in the ms that species level identification is not available for half of the arthropod taxa, be clear that unidentified taxa did not affect the result, that focus on comparisons rather than absolute flux values the conclusions should be robust, put in the text that the r^2 of the metabolic regressions is high (>95%), and by including a wide range of taxa body mass mistakes will have little impact, that your sensitivity analysis does not account for vertical turnover amongst groups, that functional composition of communities is more stable than species composition, as different species will perform similar roles, etc.

Please find my major and minor comments below. I hope they are of help to improve the manuscript.

With kind regards,
Lourens Poorter

MAJOR COMMENTS.

CAUSE-EFFECT. The cause-effect relationship between biodiversity and energy flux and distribution is unclear (for example line 62-60). Does biodiversity (loss) drive changes in energy distribution, or does energy distribution allow for a species richness and composition? Please explain more and better throughout the manuscript

“PERVADING IDEA OF TROPICAL FORESTS AS A GREEN WORLD”. It is unclear why you state that everybody thinks that tropical forests are a green world with very high animal biomass in the canopies. Why? Because most of the vegetation biomass is aboveground? Please explain better me this feels like a strawman’s argument of your paper. The paper you cite (Ellwood & Foster 2004) simply try to estimate invertebrate biomass in rainforest canopies and makes the point that basket ferns contribute a lot to this. They nowhere mention that rainforests should have less soil arthropods. . Please better cite Odum (1969) Science that old growth forests should have a brown detritus based food web, and that you now (for the first time?) have confirmed this for complex tropical rainforest

SENSITIVITY TO SPECIES LOSS. You argue that plantations are more sensitive to species loss. Please can you discuss this better? The number of interactions declines from ca 480 (I can not read the scale well) in rainforest to 320 in oilpalm plantations (Fig 3c) sounds pretty diverse to me and there should be a lot of redundancy. So please do not only discuss the relative declines but also the absolute number, as this indicates how sensitive the system is to species/functional group loss. Also, if the total belowground energy flux is the same in plantations as in rainforest, so why do you argue that the plantation system is more vulnerable to species losses (L228-237) when they have lost already so many species and the plantations are still doing the same energy flux wise. Please justify/discuss better.

MINOR COMMENTS

L32. Explain what type of plantations

L32 'pervading idea'. Do you mean 'pervasive'? And

L44-46. This is unclear. "which coincide with previously reported soil carbon stock depletion" seems to be a bit disconnected from the 3 patterns you describe in the lead sentence. What do you mean with "multistrata functioning" and 'call for managing their energy for sustainable land use' is vague. Make it more concrete.

L50. Replace 'expansion' by 'conversion towards'

L74-78. A good justification!

L86-88. A good justification. Please also include the other reviewers argument that especially arthropods are relevant for this because they reflect multiple trophic levels

L91. Explain what jungle rubber agroforestry is

L93-97. Nice that you included the direction of your hypotheses, but also explain in one sentence why you expect that direction, so that the reader is on the same page as you

L104-108 Good that you explain in a nutshell your approach. Please clarify what you mean with 'steady state' (both the biomass distribution over the compartments and the energy fluxes are stable)? Also discuss somewhere in the methods or the caveats that you assume stability but but your system is not stable (aboveground it goes fast, but belowground there are many legacies), and if 14-18yr old plantations can be considered to be in a stable state or still developing. Also explain better what you mean with 'energy losses are compensated by lower trophic levels'

L114. State explicitly that the total flux is the sum of all links

L129. "Tropical trees allocate twice as much primary production belowground in the form of litter and root biomass as they do aboveground". This statement is wrong. Primary production is the formation of new material, so by definition it is not the formation of abscised, senesced material (litter).

Fig 1. Nice fig! Say in legend that what the horizontal displacement of the group at the same trophic level means. Is this just for visualization?> I would expect that all primary consumers are aligned at the same horizontal position, and they are not. Also explain what the position on the horizontal axis means (beyond primary consumers). Is this a continuum? Do spiders have really a higher trophic level than warblers? Please also connect the name of the groups with a line to the dot, as it is now unclear where the group names belong to

L160. Nice that you justify that your bird biomass estimates are realistic.

Fig. 2. What do you mean with 'herbivory'? Does that also include root feeding? Now there is a link from plants to soil arthropods. I assume that they do not feed on green leaves or phloem but eat roots? Please be clear somewhere in the methods. Explain in legend if the plus/minus value refers to SD or SE. How come that the numbers of the 4 groups do not add up to the total value. For rainforests the 4 values add up to 321 rather than 317, for rubber to 345 rather than 314, and for oil palm to 315 rather than 310.

L187/ If I see the large SD in Fig 2 (494+-325) I can not imagine that jungle rubber has a higher energy flux than the others. And where does the Chi² 79.9 come from? In Fig 3b it says 7.7, or is that a different test?

L192. Explain that the higher pH is due to burning and ash.

L193. But earthworms can also consume SOM in rainforest. Do you suggest that there is less SOM in

rainforest and why?

L195. "Earthworms may negatively affect soil and litter arthropods through direct and indirect trophic relationships and environmental modifications". This is totally unclear. Please explain better, as this is an important result:

L227. Resilience is the product of resistance and recovery. Be explicit if belowground foodwebs are more resilient because they are more resistant (e.g., less affected by aboveground fire and conversion), or because they recover faster (but if they have a fast recovery then legacies should be less important for the soil compartment...

L228. Why should there belowground be less specialized links. Please explain

The weaknesses are that 1) for an interested outsider it is rather difficult to follow what exactly is being reported, 2) energy fluxes are inferred from biomass stocks rather than from measured fluxes, 3) biodiversity loss was assumed to underlie the patterns but was not measured, 4) relative short time intervals and soil volumes have been sampled, and the question how representative this is.

L256. Per unit of PLANT biomass? Please add if so.

L264. Please check phrasing. According to me the Fig says that in plantations the predation in the canopy and the soil are generally lower compared to rainforest (except for the rubber canopy, where it is similar)

L287. Explain why there is a lower microbial biomass in plantations

L302-306. Unclear. Explain why higher herbivory would lead to accelerated energy processing, and why that would reduce stability.

L319. Be explicit what groups you missed. Monkeys, ground dwelling mammals like deer

L327-328. Explain why reduced trophic links was not confirmed in the same sites a few years later (more time has passed since disturbance?)

L332-347. Clear conclusions and clear take home messages! Well done!

L349-373. A very good and more concrete discussion of the implications of your findings. Well done!

L583. {Please replace 'primary' by 'mature forest'. Primary forest is often interpreted as without human intervention. And humans have been historically everywhere in the rainforest. Primary and degraded are also in contradiction with each other

L586. Indicate trees > what diameter at breast height

L586-589. Mean height is not informative, as a tall forest with lots of regeneration would have on average a small height. Better report canopy height (90th percentile, like in your appendix)

L628. Why are nectar feeding and fruit feeding combined? These seem to be rather different strategies? And fruits contain seeds, so how can you allocate species to different feeding groups?

L629. What is multistrata. Explain. And define 'canopy

L626. Indicate to how many species your 418 bird occurrences belonged

L610-698. Better discuss here that sample sizes are short or small, that there is a lot of spatial heterogeneity, and that despite the tropics being relatively constant there is still seasonal variation associated with the drier and wetter season. Then put your arguments that you have in your rebuttal letter why this would not be a problem. Now you simply do not mention it. Better acknowledge that this does play a role, but why it will affect your conclusions to a limited extent.

L659. How do you get to 17 guilds? $12+4$ beetles=16?

L668. Explain that this is a random sample of body masses, and that you include this spread in your energy calculations

L746. Discuss if the assumption of stability is logic in a successional system (disturbed forest, 16y old plantations)

L746. Why do you assume that bottom resources are limitless. Please explain

L770-772. Do all these subcompartments add up to the total, or do the two approaches yield different results

L778. What random slopes? What was the independent, and what was the dependent here?

L774-792. Explain this way better. For me it is unclear what exactly were the predictor variables and the response variables in your model, and why you used these tests.

Author Rebuttals to First Revision:

Referees' comments (revision)

Referee #1

I am overall happy with the way the authors have addressed my comments.

Regarding the justification of the choice for microarthropods and earthworms, I am fine with that addition, but – very little remark - suggest to leave out 'well'; is somewhat 'safer' as groups like protists and nematodes are not in.

The reformulated hypotheses still have 'changes' and 'differences' in the formulation, but I like them much better, in particular (LL287-288) "...we expected higher utilisation of primary basal food resources (i.e., plants and detritus) in plantations due to lower microbial biomass. Furthermore, I am happy with the explanations for the observed arthropod decline (LL 193-198), the changes in Table1/AppTable2 and the paragraph on possible legacy effects.

Response: Thank you for positive feedback to our revised text. As suggested, we now leave out 'well' from the sentence in the introduction (replaced with 'reliably reflecting'). We have also added a few more improvements according to the comments of Reviewers #2 and #3.

Referee #2

The authors addressed many of the questions, concerns and suggests that were raised. There are at least three main threads in the paper. The first two are well presented, the third and most important, is weakly supported as presented.

The first thread focuses on the energetic organizations of birds and arthropods in a native tropical forest and three plantation types (former native forests) in the region. Methods and sample size notwithstanding, the dominance of the 'brown' versus green web in descriptions of these tropical systems is apparent and novel. The authors point out the novelty of the results in the revision and in their response to my comment in the review.

The second focuses on the differences and similarities in the energetic organization across the system types. The more interesting results being the shift in the away from belowground arthropods to the invasive earthworms. The title – Rainforest transformation reallocates energy from green to brown food webs – and summary section - in rainforest >90% of the total animal energy flux was channelled by arthropods in soil and canopy, while in plantations >50% of the energy was allocated to annelids (earthworms) are explicit on this, while the conclusion section (conclusion 3) is less so.

The third thread, and most important in terms of interest and impact, focuses on the potential consequences of these changes on carbon storage in tropical systems (a point raised in the summary - Our study shows that tropical land-use change drives consistent decline of multitrophic energy flux aboveground, and intensification of soil organic matter consumption joint [sic] with nearly complete elimination of energy flux to predators belowground, which

coincide with previously reported soil carbon stocks depletion - and in conclusion 4 - belowground food webs in plantations rely on different basal resources than those in rainforest, promoting faster energy channels.) This is one of the more interesting conclusions as it links changes in the energetic organizations of the food webs in the plantations to the faster energy channels to depleted carbon stocks in the systems. The connection being made to the faster energy channel is tenuous as presented. Given that this is a reconstructed food web parameterized to estimate energy flux via standing biomass and physiologies, from C flux, it certainly would be possible to provide some estimates of C respiration emanating from each of the belowground channels. This would strengthen the conclusion that the faster channel is operating in the plantations. Conclusion 4 further states that – [t]hese changes are associated with previously observed depletion of carbon stocks¹⁷, but the causal relationships here still need to be tested. What was the magnitude of the observed changes in carbon stocks, and could the observed shift belowground towards the earthworms account for the losses? Attempting to answer this question using the data and analytical approach in place would move the results beyond an interesting observation and comparative study to a more impactful study in terms of our understanding of food web responses to tropical forest management and land use.

Response: Thank you for nicely summarising our main messages. To be more explicit with point #2, we have now rephrased the conclusion section as follows: "(2) animal communities in tropical canopies suffer higher total energetic losses due to rainforest transformation than those in belowground food webs, but the energy in belowground food webs in plantations is reallocated from functionally diverse arthropod communities to invasive earthworms" (LL 386-389). To directly link our results to the changes in soil carbon stocks with land-use change, we have now calculated an additional food-web derived index of 'Carbon balance', representing the ratio of production to consumption of soil organic matter by soil animal communities. Specifically, we calculated the ratio between the production of faeces (i.e., unassimilated food) and the consumption of soil organic matter by all soil invertebrates. This ratio was much higher in rainforest and clearly supported the link between changes in energy flux distribution and carbon balance. However, since the effect of faeces production on soil organic matter production and stabilisation is indirect, it is difficult to convert it to a quantitative measure at the ecosystem scale and over time. We believe that this would be an exciting question to address with a dynamic ecosystem-level modelling including environmental, plant, and microbial data. However, this falls beyond the scope of our study. We now refer to these new information in the abstract (LL 38-43), results (LL 337-348), conclusions (LL 392-396), and methods (LL 842-851):

[Abstract] "Land-use change led to a consistent decline in multitrophic energy flux aboveground, while belowground food webs responded with reducing energy flux to higher trophic levels down to -90% and shifting the energetic balances from slow (fungal) to fast (bacterial) energy channels, and from faeces production towards consumption of soil organic matter. This coincides with previously reported soil carbon stock depletion."

[Results] "To quantify animal effects on soil carbon stocks, we calculated the ratio between the production of faeces (i.e. unassimilated food) and the consumption of soil organic matter by all soil invertebrates. It has been shown that conversion of plant materials into faeces by soil invertebrates increases microbial biomass production, which is the key process contributing to soil organic matter formation and stabilisation. Supporting the link between the belowground food-web structure and net carbon loss in plantations, we found that the production-to-consumption ratio decreased by more than 75% from 27.6 ± 29.6 in rainforest

to 3.8 ± 2.9 in jungle rubber, 6.2 ± 10.4 in rubber and 2.3 ± 0.3 in oil palm plantations (Fig. 3f). Although quantification of these animal effects over time requires dynamic ecosystem-level modelling, our analysis suggests that changes in energy flux distribution due to habitat transformation have large functional consequences for carbon cycling.”

[Conclusions] “...(4) belowground food webs in plantations rely on different basal resources than those in rainforest, promoting faster energy channeling and shifting carbon balance from production to consumption of soil organic matter. These changes are associated with previously observed depletion of carbon stocks¹⁷, but the magnitude of animal effects in this context remains to be tested.”

[Methods] “...(6) ratio between the production of faeces and the consumption of soil organic matter (a proxy for soil organic matter/carbon balance). The latter indicator is novel and is based on three main lines of evidence: (i) conversion of plant material into faeces by soil invertebrates increases microbial biomass production; (ii) microbial biomass production is the key process contributing to soil organic matter formation and stabilisation; (iii) consumption of soil organic matter by invertebrates (a sum of outgoing fluxes from soil organic matter) leads to consumption of associated microbial biomass and thus has opposite effect to the first two. To calculate the production of faeces, we multiplied all energy fluxes to inverted assimilation efficiency and summed them up, thus quantifying all unassimilated food in the food web.”

Referee #3

The manuscript by Potapov compares above- and belowground foodweb structure and fluxes across 4 different tropical land use systems. Their main findings are that 1) in rainforest the belowground foodweb is larger in terms of biomass and energy flux than the aboveground foodweb, 2) compared, to rainforests, plantations have a similar energy flux, but somewhat less foodweb interactions, relatively less predators belowground, a higher dominance by earthworms and bacteria dominated carbon cycling.

As I said In my earlier review, the strengths of the manuscript are 1) a fascinating study system (foodwebs), 2) a rather complete description of the foodweb (both above and belowground, birds, canopy arthropods, and belowground arthropods and worms), and 3) the comparison of rainforest with common forms of commodity plantations.

The revised version of the manuscript has been much improved, is interesting, and a pleasure to read. The description of the food web construction and energy flux estimates is much more clear, the overarching hypothesis and the detailed hypothesis per section are much better explained and justified, the structure is easier to follow through the headers and more clear take home messages, and I especially appreciate that it is now more clearly explained what forest conversion towards commodity plantation means for ecosystem functioning, and how the obtained insights on energy fluxes can be more hands on used to improve plantation design and management to mitigate the negative effects. I congratulate the authors therefore with the overall result.

Response: Thank you for positive feedback on the revised manuscript.

I am not a food network specialist so I hope that the other two reviewers really have checked if the network construction is fine and if the assumptions are solid (their reviews were rather brief and more focused on the interpretation of the results than on the methods).

I found the responses to my comments overall satisfactory and convincing, but it was often unclear what exact changes were made in the manuscript. Please put below each of my previous comments with quotes how and where you changed things in the manuscript. You should not only convince me in the cover letter, but also the future readers of the manuscript, so make sure that these considerations/justifications are included in the revised manuscript. For example. Say in the ms that species level identification is not available for half of the arthropod taxa, be clear that unidentified taxa did not affect the result, that focus on comparisons rather than absolute flux values the conclusions should be robust, put in the text that the r^2 of the metabolic regressions is high (>95%), and by including a wide range of taxa body mass mistakes will have little impact, that your sensitivity analysis does not account for vertical turnover amongst groups, that functional composition of communities is more stable than species composition, as different species will perform similar roles, etc.

Please find my major and minor comments below. I hope they are of help to improve the manuscript.

With kind regards,
Lourens Poorter

Response: We are sorry for being not always explicit about the changes made. We ensured that all points mentioned in our answers below are also addressed in the manuscript. We have uploaded the version with tracked changes for transparency. We also revised our responses in the previous revision round. In the cases where no changes were made, the suggested information is now added to the manuscript and these changes are also specified in the separate section at the end of our replies with quotations and line numbers. The energy flux approach has been proven as a powerful method in food-web ecology and assumptions behind the network reconstructions were thoroughly tested via sensitivity analyses. This may explain why there are no strong concerns from other reviewers regarding the overall methodological framework.

MAJOR COMMENTS.

CAUSE-EFFECT. The cause-effect relationship between biodiversity and energy flux and distribution is unclear (for example line 62-60). Does biodiversity (loss) drive changes in energy distribution, or does energy distribution allows for a species richness and composition? Please explain more and better throughout the manuscript

Response: Thank you for raising this important question. Biodiversity and energy flux are often correlated, but the cause-effect relationship is complex. In the framework of our study, we consider two main mechanisms. First, changes in the total energy flux (cause) may pose

limits to the number of species in the ecosystem (effect). Second, changes in consumer communities induced by environmental shifts or other reasons (i.e. biodiversity = cause), change the distribution of energy, which restructures food webs and thus energy fluxes (effect). These changes may consequently feed back to community shifts (i.e. there is a feedback loop on the biodiversity distribution across compartments). We agree that these interrelationships were not clearly communicated in the introduction. We have now rephrased the second paragraph of the introduction and focussed it on this question (the above-belowground topic was moved to a separate paragraph below). Here is the updated text (LL 61-76):

“Losses of animal diversity may be explained by reduced primary ecosystem productivity and by changes in the structure of consumer communities and interactions within, as has been shown in studies on the impacts of invasive species, climate, or other environmental changes. Energy, as a common currency that sustains life, can impose limits on the total number of species in an ecosystem, while community shifts change energy pathways through ecological networks (energy flux), which is closely associated with the distribution of biodiversity across different trophic levels and ecosystem compartments. For instance, under tropical land-use change, large declines in the number of species correlated with a simultaneous reduction in total energy flux in litter invertebrate communities¹³, demonstrating that biodiversity loss is associated with a loss in available energy. In soil, however, a similar decline in biodiversity was not associated with reduced total energy flux, but with a redistribution of energy across the food web. This indicates that biodiversity loss is associated with exclusion of specific functional groups, rebalancing the system energetically. Disentangling total available energy changes from shifts in its distribution may help us to determine appropriate measures for restoration of ecosystem functioning.”

“PERVADING IDEA OF TROPICAL FORESTS AS A GREEN WORLD”. It is unclear why you state that everybody thinks that tropical forests are a green world with very high animal biomass in the canopies. Why? Because most of the vegetation biomass is aboveground? Please explain better me this feels like a strawmen’s argument of your paper. The paper you cite (Ellwood & Foster 2004) simply try to estimate invertebrate biomass in rainforest canopies and makes the point that basket ferns contribute a lot to this. They nowhere mention that rainforests should have less soil arthropods. . Please better cite Odum (1969) Science that old growth forests should have a brown detritus based food web, and that you now (for the first time?) have confirmed this for complex tropical rainforest

Response: We appreciate the seminal paper of Odum about ecosystem development which proposes that mature ecosystems channel energy mainly through the detrital pathway. Yet, we see some conceptual differences with our data. The detrital pathway dominates in virtually any terrestrial ecosystem (e.g. Cebrian 1999 <https://doi.org/10.1086/303244>), but 95% of the energy in the detrital pathway is assumed to be processed by microorganisms (e.g. Petersen and Luxton 1982 doi 10.2307/3544689). Therefore, energetic dominance of soil animals over canopy animals in rainforest is not trivial. None-the-less, there is a perception of rainforests as a green world, which is clearly seen in the research focus on aboveground tropical biodiversity. The cited paper of Ellwood and Foster 2004 is just an example of invertebrate biomass estimation from epiphytic ferns, which was published in Nature, indicating high interest in the estimation of invertebrate biomass in canopies. At the same time, none of the estimations of the tropical soil invertebrate biomass were published

in high-rank interdisciplinary journals to our knowledge. To tone down this message, we have replaced 'pervading idea' with 'widespread perception' – as evidenced by the heavy focus on canopy fauna. We also clarified our arguments in the text and added relevant references:

In the abstract: "In contrast to the widespread perception of tropical ecosystems as a 'green world', our results indicate that most of the energy in rainforests is channelled to the belowground animal web."

In the results (LL 136-140): "These figures contrast with the widespread perception of tropical ecosystems as a 'green world' and question the current research focus on tropical animal biomass aboveground (including publications in Nature25). This energetic dominance of soil over canopy animals in rainforest is unexpected because c. 95% of the energy channelled belowground is assumed to be processed by microorganisms."

SENSITIVITY TO SPECIES LOSS. You argue that plantations are more sensitive to species loss. Please can you discuss this better? The number of interactions declines from ca 480 (I can not read the scale well) in rainforest to 320 in oilpalm plantations (Fig 3c) sounds pretty diverse to me and there should be a lot of redundancy. So please do not only discuss the relative declines but also the absolute number, as this indicates how sensitive the system is to species/functional group loss. Also, if the total belowground energy flux is the same in plantations as in rainforest, so why do you argue that the plantation system is more vulnerable to species losses (L228-237) when they have lost already so many species and the plantations are still doing the same energy flux wise. Please justify/discuss better.

Response: We agree that the sensitivity to potential species losses was not clearly explained. Neither was it strongly supported by our results. To avoid emphasising results that are not robust, we have removed this argument. Instead, we now conclude that soil animal communities in plantations are able to sustain comparable energy flux, despite a very different food-web structure (LL 263-267): "Therefore, soil animal communities in plantations rely on few strong interactions, reflecting documented losses of biodiversity and multifunctionality, but nevertheless process a similar amount of energy as soil animal communities in rainforests. This demonstrates a remarkable adaptability of belowground food-web functioning to perturbations."

MINOR COMMENTS

L32. Explain what type of plantations

Response: Specified now in L34.

L32 'pervading idea'. Do you mean 'pervasive'? And

Response: Replaced now with 'widespread perception'.

L44-46. This is unclear. “which coincide with previously reported soil carbon stock depletion” seems to be a bit disconnected from the 3 patterns you describe in the lead sentence. What do you mean with “multistrata functioning” and ‘call for managing their energy for sustainable land use’ is vague. Make it more concrete.

Response: The abstract has been largely restructured to accommodate the requested changes across new comments and to emphasise new results on the carbon balance. We have also made the concluding sentence clearer, although space limitations mean that we cannot include more details. Here is the updated second part of the abstract (LL 38-47): “...Land-use change led to a consistent decline in multitrophic energy flux aboveground, while belowground food webs responded with reduced energy flux to higher trophic levels, down to -90%, and with shifts in energetic balances from slow (fungal) to fast (bacterial) energy channels and from faeces production towards consumption of soil organic matter. This coincides with previously reported soil carbon stock depletion. Our study shows that previously documented animal biodiversity declines with tropical land-use change are associated with vast energetic and functional restructuring in food webs. We highlight the pervasive role of land-use choices on multitrophic and multistrata functioning of tropical ecosystems and call for multidimensional restoration measures to manage energy for sustainable land use.”

L50. Replace ‘expansion’ by ‘conversion towards’

Response: Replaced, thank you.

L74-78. A good justification!

L86-88. A good justification. Please also include the other reviewers argument that especially arthropods are relevant for this because they reflect multiple trophic levels

Response: Thank you. We modified the sentence as follows: “Our group selection represents the majority of the animal biomass (arthropods, earthworms), including ecosystem engineers (earthworms, ants) and animals at different trophic levels – from detritivores, microbivores, and herbivores (various arthropod groups), to top predators (e.g., spiders, birds) ”.

L91. Explain what jungle rubber agroforestry is

Response: Explained now: “...including jungle rubber (selectively logged rainforest with planted rubber trees)...“

L93-97. Nice that you included the direction of your hypotheses, but also explain in one sentence why you expect that direction, so that the reader is on the same page as you

Response: We have now added short justifications in the introduction (LL 102-111): “Our main hypothesis was that there are different keystone animal groups that channel most of

the energy in rainforest and plantations, and that energy distribution changes with land use across strata (more energy allocated to aboveground food webs in plantations because plantation management commonly aims to maximize aboveground production), trophic levels (low predation in plantations because monocultures sustain lower predation than diverse plant communities) and basal resources (disproportionally high herbivory in plantations because of a lower predation pressure and monodominant plant species).” Detailed justification is provided in the individual sections later in the text.

L104-108 Good that you explain in a nutshell your approach. Please clarify what you mean with ‘steady state’ (both the biomass distribution over the compartments and the energy fluxes are stable)? Also discuss somewhere in the methods or the caveats that you assume stability but your system is not stable (aboveground it goes fast, but belowground there are many legacies), and if 14-18yr old plantations can be considered to be in a stable state or still developing. Also explain better what you mean with ‘energy losses are compensated by lower trophic levels’

Response: To explain better our approach and highlight the assumption of steady state, we modified the text as given below.

Introduction (LL 113-126): “To test our hypotheses, we estimated abundance and biomass of canopy arthropods using fogging, birds using audio recorders and point counts, and soil arthropods and earthworms using high-gradient heat extractors across 32 sites representing rainforests and plantations²⁴. We linked collected body mass and biomass data to literature data on traits and feeding preferences of taxa to define 62 trophic guilds across all animal groups (Extended Data Table 1) and reconstruct food-web topologies at each site. We further used steady-state food-web modelling, which assumes that energetic demands of each trophic guild (including metabolic rate, losses during the food assimilation and consumption by higher trophic levels) are compensated by taking energy from lower trophic levels. Metabolic rates of each guild per unit biomass were estimated from body masses using metabolic regressions and multiplied by the observed biomasses. Resulting energy fluxes were used as quantitative measures of the distribution of energy and consumption of different resources (living plants, litter, bacteria, fungi, soil organic matter, other animals) in aboveground and belowground food webs^{12,13}.”

Discussion (LL 362-367): “Finally, our plantation systems were 14-18 years old and were unlikely to be at a stable state, especially considering higher rates of change in the aboveground than in the belowground ecosystem compartment. We therefore call for studies evaluating tropical land-use systems in the longer term. To prove the generality of our findings, we performed another survey at the same sites (except jungle rubber) in 2016-2017....”

L114. State explicitly that the total flux is the sum of all links

Response: Added: “The total aboveground energy flux (sum of all energy fluxes to canopy arthropods and birds) was...”

L129. “Tropical trees allocate twice as much primary production belowground in the form of litter and root biomass as they do aboveground”. This statement is wrong. Primary production is the formation of new material, so by definition it is not the formation of abscised, senesced material (litter).

Response: The sentence was modified for clarity: “(1) tropical trees allocate twice as much produced organic matter belowground, in the form of litter and root biomass, as they store aboveground”

Fig 1. Nice fig! Say in legend that what the horizontal displacement of the group at the same trophic level means. Is this just for visualization?> I would expect that all primary consumers are aligned at the same horizontal position, and they are not. Also explain what the position on the horizontal axis means (beyond primary consumers). Is this a continuum? Do spiders have really a higher trophic level than warblers? Please also connect the name of the groups with a line to the dot, as it is now unclear where the group names belong to

Response: Thank you. Trophic level (position) is a continuous variable and spiders have a higher trophic level than warblers in our food webs. This can be explained by warbler’s preference for abundant prey that are large herbivorous arthropods in canopies (trophic position in general not correlated with body mass in terrestrial food webs: <https://www.journals.uchicago.edu/doi/10.1086/705811>). We have now added details to the caption: “...Nodes are ordered horizontally according to the trophic position (continuous variable; nodes were slightly jittered to avoid overlaps, but the general order remains) and vertically according to the ecosystem stratification (positions within the four major animal groups/colors are random).” The names and the nodes are not aligned because we used broad taxonomic groups to depict several nodes per trophic level each (and not to overwhelm the figure with guild names). We now clarify this in the caption: “Exemplary dominant taxonomic groups within the major trophic levels (primary consumers, omnivores and primary predators, top predators) are shown with text.”

L160. Nice that you justify that your bird biomass estimates are realistic.

Response: Thank you.

Fig. 2. What do you mean with ‘herbivory’? Does that also include root feeding? Now there is a link from plants to soil arthropods. I assume that they do not feed on green leaves or phloem but eat roots? Please be clear somewhere in the methods. Explain in legend if the plusminus value refers to SD or SE. How come that the numbers of the 4 groups do not add up to the total value. For rainforests the 4 values add up to 321 rather than 317, for rubber to 345 rather than 314, and for oil palm to 315 rather than 310.

Response: Herbivory includes feeding on leaves or roots. We added this information to the caption and the results (L 199). We have now explained in the caption that variation refers to 1 SD. We checked again the numbers on the figures and corrected inconsistencies. Updated numbers are provided in the revised figure 2 (small deviations in 1 unit is due to rounding up

the means of individual groups). There were no earthworms on two of the sampling sites and this was not accounted for in the per-group energy flux calculations. This led to higher total energy flux values while summing up the group-specific means. Based on this observation, we thoroughly checked all other calculations and detected a similar problem in per-group energy flux summaries and in the data behind the first linear model (that tested the interactive effects of land use, animal group, and trophic function). We have now replaced this model with two separate models - one testing the effect of land use in interaction with animal group and another one testing the effect of land use in interaction with trophic function. Results of these new models are presented in Extended Data Table 2 and updated statistics are added to the sections “Changes in energetic keystone groups with habitat transformation” and “Changes in trophic functions with habitat transformation”. We also recalculated values for Extended Data Fig. 6 and 7 and Extended Data Table 3 which are now slightly different from the previous versions. Other presented averages and figures are correct, and main analyses and conclusions were not affected by these changes. It is our unfortunate mistake and we are grateful that you have spotted this inconsistency.

L187/ If I see the large SD in Fig 2 (494+-325) I can not imagine that jungle rubber has a higher energy flux than the others. And where does the Chi2 79.9 come from? In Fig 3b it says 7.7, or is that a different test?

Response: The significant system effect was caused by an imbalanced-factor design in our initial model. As we describe in our previous reply, we have now split this model into two: with animal groups and with trophic functions. The overall effect of the system in both models on the total energy flux is not significant (LL 210-212): “The total animal energy flux was similar in rainforest and monoculture plantations (310-317 mW m⁻²), and was ca. 50% higher in jungle rubber, though the variation was very high (the total System effect was not significant; Fig. 2; Extended Data Table 2). “. However, the overall effect of system was (weakly) significant in the above/belowground model, probably because of the total energy flux decline aboveground (Fig. 3b). In the results and discussion we emphasise different responses of above- and belowground compartments, rather than the overall system effect, which, as you noted, is not robust (LL 245-249): “The belowground energy flux was higher than the aboveground in jungle rubber (ca. 30-fold), rubber (55-fold) and oil palm monocultures (68-fold), with an even higher difference in biomass (Fig. 3a,b). The total aboveground energy flux was reduced by -75 to -79% in both monoculture plantation types in comparison to rainforest.” The total energy flux effects were not included in the main messages of the study and thus the above mentioned information and the model change does not affect our conclusions. We are grateful that you have spotted this inconsistency.

L192. Explain that the higher pH is due to burning and ash.

Response: Added (L 218).

L193. But earthworms can also consume SOM in rainforest. Do you suggest that there is less SOM in rainforest and why?

Response: We now explicitly quantify the carbon balance using the SOM consumption versus the SOM (faeces) production. See our replies above. In the scope of our study, we are focussing on the animal-driven SOM trends.

L195. "Earthworms may negatively affect soil and litter arthropods through direct and indirect trophic relationships and environmental modifications". This is totally unclear. Please explain better, as this is an important result:

Response: We now specify these effects (LL 221-223). Here we refer mainly to the existing knowledge rather than our own results: "It is known that earthworms may negatively affect soil and litter arthropods through direct (consumption of small fauna) and indirect trophic interactions and environmental modifications (litter removal, microbial feeding)".

L227. Resilience is the product of resistance and recovery. Be explicit if belowground foodwebs are more resilient because they are more resistant (e.g., less affected by aboveground fire and conversion), or because they recover faster (but if they have a fast recovery then legacies should be less important for the soil compartment...

Response: In this sentence we are referring to a published paper where mainly resistance is mentioned as the underlying process (although not explicitly). We have now replaced 'resilience' with 'resistance'.

L228. Why should there belowground be less specialized links. Please explain [The weaknesses are that 1) for an interested outsider it is rather difficult to follow what exactly is being reported, 2) energy fluxes are inferred from biomass stocks rather than from measured fluxes, 3) biodiversity loss was assumed to underlie the patterns but was not measured, 4) relative short time intervals and soil volumes have been sampled, and the question how representative this is.]

Response: Less specialised links in belowground food webs have been reported in the literature (<http://doi.wiley.com/10.1111/oik.00865>). The suggested mechanism is that organisms cannot successfully search for specific food objects in the environment where mobility is restricted. We now mention this in the text (LL 254-257): "The pattern also fits the 'green-brown imbalance' hypothesis, which suggests a higher resistance of belowground than aboveground food webs due to a lower number of specialised links in the former (because of restricted mobility of organisms and thus food selection)." We assume that the text in [square brackets] has been accidentally copied from the first review, where we also addressed these potential weaknesses.

L256. Per unit of PLANT biomass? Please add if so.

Response: Correct. We specified 'plant biomass'.

L264. Please check phrasing. According to me the Fig says that in plantations the predation in the canopy and the soil are generally lower compared to rainforest (except for the rubber canopy, where it is similar)

Response: The sentence refers to rubber monocultures only. Similar (slightly higher) predation/primary consumption ratios are visible on Fig. 3d. We consider our phrasing correct.

L287. Explain why there is a lower microbial biomass in plantations

Response: Microbial biomass is primarily driven by easily accessible energy (e.g. root exudates) and nutrient supply (N and P availability), but may be related to a number of other factors. We prefer to avoid speculations in this text as the drivers of microbial biomass fall beyond the scope of our study.

L302-306. Unclear. Explain why higher herbivory would lead to accelerated energy processing, and why that would reduce stability.

Response: It has been suggested that stability of a system is supported by a balance between the fast (responsive) and the slow (inert) energy channels (<https://linkinghub.elsevier.com/retrieve/pii/S016953471100259X>). Herbivory can be envisioned as the 'fast' energy channel in comparison to the 'slow' detritivorous energy channel, because living biomass is directly entering the animal food web in the former, without microbial decomposition and partial stabilisation. However, this is merely a hypothesis. To stick to well established concepts, we now discuss only bacterivory-vs-fungivory in this context (originally referred to by Rooney and McCann), and rephrase the sentence for clarity (LL 332-336): "The likely increase in bacterivory therefore indicates that there is accelerated energy processing (faster turnover rates) in these systems. A shift from the naturally observed balance to fast energy channel-dominated food webs may destabilise the system and may accelerate depletion of carbon stocks, which has been observed in rubber and oil palm plantations". Herbivory is now discussed in the paragraph above mainly to demonstrate the pest control potential (LL 307-310): "Reduced natural pest control in oil palm is also supported by a lower predation-to-herbivory ratio (0.37 ± 0.16 in birds, 0.28 ± 0.05 in canopy arthropods, and 1.14 ± 0.63 in soil arthropods) in comparison to rainforest (0.64 ± 0.29 in birds, 0.34 ± 0.05 in canopy arthropods, and 1.95 ± 0.74 in soil arthropods)."

L319. Be explicit what groups you missed. Monkeys, ground dwelling mammals like deer

Response: We prefer to stay at the general scale of classes (amphibians, reptiles, mammals). Bats were highlighted because of their potential energetic importance. However, we cannot highlight all relevant groups (which are many) due to space limitations.

L327-328. Explain why reduced trophic links was not confirmed in the same sites a few years later (more time has passed since disturbance?)

Response: We added this potential explanation (LL 374-377): "Potentially, some trophic links were restored as plantations aged (from c. 15 in the main survey to c. 19 years old in the validation survey), but future plantation replanting will likely result in a second wave of biodiversity decline, which may lead to further food-web disassembly."

L332-347. Clear conclusions and clear take home messages! Well done!

L349-373. A very good and more concrete discussion of the implications of your findings. Well done!

Response: Thank you.

L583. {Please replace 'primary' by 'mature forest'. Primary forest is often interpreted as without human intervention. And humans have been historically everywhere in the rainforest. Primary and degraded are also in contradiction with each other

Response: The term "primary degraded lowland rainforest" comes from the publication "Margono, B. A., Potapov, P. V., Turubanova, S., Stolle, F. & Hansen, M. C. Primary forest cover loss in Indonesia over 2000–2012. *Nature Climate Change* 4, 730–735 (2014)." which we refer to. This term well describes our systems (<http://www.sciencedirect.com/science/article/pii/S0006320717303968>) and thus we prefer to keep the term.

L586. Indicate trees > what diameter at breast height

Response: All trees with the diameter at breast height ≥ 10 cm were recorded. We added this information to the methods (L 635).

L586-589. Mean height is not informative, as a tall forest with lots of regeneration would have on average a small height. Better report canopy height (90th percentile, like in your appendix)

Response: 90th percentile tree heights are reported in the methods now (LL 638-646).

L628. Why are nectar feeding and fruit feeding combined? These seem to be rather different strategies? And fruits contain seeds, so how can you allocate species to different feeding groups?

Response: Indeed, herbivory can be functionally different. However, disentangling these functions was out of the scope of our study. We focused on herbivory from an ecosystem point of view, defined as the consumption of living plant matter. This decision has been made based on our research questions and hypotheses.

L629. What is multistrata. Explain. And define 'canopy

Response: 'multistrata' refers to both the canopy and the ground; specified now in the text: "canopy, ground foraging or both". This classification refers to the Elton Traits 1.0 dataset (Wilman et al. 2014). Unfortunately, canopy is not clearly defined in their metadata. In the framework of our study, trophic interactions between 'canopy' birds and arthropods collected using canopy fogging are plausible. We now added this clarification to the text (LL 786-788): "...and full overlap between 'canopy' birds and arthropods collected using canopy fogging".

Wilman, H. et al. EltonTraits 1.0: Species-level foraging attributes of the world's birds and mammals. *Ecology* 95, 2027–2027 (2014).

L626. Indicate to how many species your 418 bird occurrences belonged

Response: In total we recorded 71 individual bird species. For more details on the community and species identities please refer to Prabowo et al. 2016 and Clough et al. 2016.

Prabowo, W.E., Darras, K., Clough, Y., Toledo-Hernandez, M., Arlettaz, R., Mulyani, Y.A., Tscharntke, T., 2016. Bird Responses to Lowland Rainforest Conversion in Sumatran Smallholder Landscapes, Indonesia. *PLOS ONE* 11, e0154876. <https://doi.org/10.1371/journal.pone.0154876>

Clough, Y., Krishna, V.V., Corre, M.D., Darras, K., et al., 2016. Land-use choices follow profitability at the expense of ecological functions in Indonesian smallholder landscapes. *Nature Communications* 7, 13137. doi:10.1038/ncomms13137

L610-698. Better discuss here that sample sizes are short or small, that there is a lot of spatial heterogeneity, and that despite the tropics being relatively constant there is still seasonal variation associated with the drier and wetter season. Then put your arguments that you have in your rebuttal letter why this would not be a problem. Now you simply do not mention it. Better acknowledge that this does play a role, but why it will affect your conclusions to a limited extent.

Response: We now added these details to the methods (LL 653-663): "Our assessment is a snapshot that cannot represent all animal species at the study sites. However, the functional composition of communities is typically more stable than the species composition; i.e. despite species turnover, different species will perform similar roles in the food web. This turnover, however, is expected to be moderate due to a limited seasonality at the study region, with a rainier period during December–March and a dryer period during July–August. Although we were not able to fully cover the spatial heterogeneity within each plot, our sampling design compensates for this with true replication of n = 8 plots per system. To account for the temporal variation, validate results of the main survey, and prove the generality of our findings, we did another independent survey with the same approach at the same sites (except jungle rubber, i.e. 24 plots) in 2016-2017."

L659. How do you get to 17 guilds? 12+4 beetles=16?

Response: There were 12 orders in total. Hymenoptera were divided into Formicidae and Braconidae, while Coleoptera were divided into Chrysomelidae, Curculionidae, Elateridae, Staphylinidae and 'other Coleoptera'. Total = 10+2+5 = 17. We have now modified the text to make it clearer (LL 712-721): "Arthropods were then sorted to 12 major arthropod orders (Acarina, Araneae, Blattodea, Coleoptera, Collembola, Diptera, Hemiptera, Hymenoptera, Lepidoptera, Orthoptera, Psocoptera and Thysanoptera). As large flying taxa such as Apoidea and Vespoidea in part actively evaded the insecticide fog at the time of application (J. Drescher, Pers. obs.), the order Hymenoptera in this study is represented by Formicidae (ants) and Braconidae (a family of parasitoid wasps), both of which were highly abundant in the samples. Additionally, four abundant beetle families with contrasting feeding strategies were analyzed separately from the rest of the order Coleoptera (henceforth termed 'other Coleoptera'), i.e. Chrysomelidae, Curculionidae, Elateridae and Staphylinidae."

L668. Explain that this is a random sample of body masses, and that you include this spread in your energy calculations

Response: An explanation was added: "up to 10 random individuals per sample per group to estimate the mean"

L746. Discuss if the assumption of stability is logic in a successional system (disturbed forest, 16y old plantations)

Response: We added an explanation to the methods (LL 811-817): "...assuming a steady-state energetic system (i.e., energetic losses from each node are compensated by the lower trophic levels; e.g. if herbivores are present in the system there is enough plant biomass to sustain them). Although the steady-state assumption is unlikely to be fully supported in most real-world ecosystems, this assumption allows for comparison of dominant energy processes across different ecosystems that are stable at the time of consideration (years) and thus was appropriate for our aims."

L746. Why do you assume that bottom resources are limitless. Please explain

Response: The logic of top-down energy flux calculation suggests that there are 3 g of herbivores per square metre in a community at a given time point, there is enough plant biomass to sustain these herbivores (unless the whole ecosystem collapses, which is not the case). See quotes in our previous reply.

L770-772. Do all these subcompartments add up to the total, or do the two approaches yield different results

Response: Similar to biomass, the total energy flux can be broken down to individual energy channels/subcompartments, which together sum up to the total energy flux. We prefer not to specify this detail in the text since it is not critical for understanding of the results.

L778. What random slopes? What was the independent, and what was the dependent here?

Response: The main model is now split into two models without random slopes and the paragraph was modified accordingly (LL 856-864): “To analyze the overall distribution of energy flux across animal groups and trophic functions, we first ran two mixed-effect models testing the effect of land-use system (rainforest, jungle rubber, rubber, oil palm), region (two regions included in the design), and either major animal group or trophic function on energy fluxes in food webs (the lme4 package). Two models were run separately because not all functions are performed by all groups. Chi-square, significance and degrees of freedom were approximated using Wald Chi-square tests (the car package). We allowed for random intercepts in animal groups and trophic functions depending on the plot to account for interdependence of food-web components in the same site...”. Full model syntaxes are given in Extended Data Table 5.

L774-792. Explain this way better. For me it is unclear what exactly where the predictor variables and the response variables in your model, and why you used these tests.

Response: The paragraph on statistical methods was considerably modified according to the changes listed in our previous replies and the new carbon balance variable. Here is the new version of the first part of this paragraph (LL 856-871): “To analyze overall distribution of energy flux across animal groups and trophic functions, we first ran two mixed-effect models testing the effect of land-use system (rainforest, jungle rubber, rubber, oil palm), region (two regions included in the design), and either major animal group or trophic function on energy fluxes in food webs (the lme4 package⁸¹). Two models were run separately for groups and functions because not all functions are performed by all groups. Chi-square, significance and degrees of freedom were approximated using Wald Chi-square tests (the car package⁸²). We allowed for random intercepts depending on the plot to account for interdependence of groups and functions in the same site. To test specific hypotheses related to changes in trophic functions (first – more energy allocated to aboveground food webs in plantations, second – lower predation in plantations, third – a shift in basal resource feeding and carbon cycling across land-use systems), generalized linear models were run for each of the four bulk food-web parameters calculated separately for above- and belowground food-web compartments (response variables: total biomass, total energy flux, number of trophic links and trophic hierarchy) and two indicators of carbon cycling in belowground food webs (response variables: bacteria-to-fungi ratio and faeces production-to-soil consumption ratio)...”. Full model syntaxes are given in Extended Data Table 5.

Specified replies to the comments of Reviewer #3 to the initially submitted manuscript

Response: Based on the revision of the previous comments, and accounting for the changes described above, the following additional edits were made in the text:

- 1) “Since species-level biology of tropical invertebrates is poorly known and we did not have species-level information for about 50% of the studied arthropods, traits were assigned to supraspecific taxa assuming their general trophic and functional consistency” (LL 772-774).
- 2) “...the used metabolic regressions typically have $R^2 > 95\%$ if calculated for a wide range of body masses...” (LL 806-807).
- 3) “Since the absolute estimates of the energy flux can be biased due to the above-mentioned assumptions and regression-based conversions, we focus mainly on comparisons in our main conclusions” (LL 824-826).
- 4) “Assuming the same density and community composition in the unassessed community, we multiplied canopy arthropod biomass by the ratio of unassessed to assessed community (LL 910-912; already included in the first revision)

All our other replies were accompanied by edits and referred to specific line numbers in the revised MS. Our complete response to the first round of comments is provided below. We have now added direct quotes [**highlighted with bold**] in the places where they were missing, while keeping the line numbers from the first revision of the MS (as originally given in these replies). The new version with changes tracked is provided with the clean revised version in the submission.

Referee #3 – first revision round (refined responses)

The manuscript by Potapov et al. compares above- and belowground foodweb structure and fluxes across 4 different tropical land use systems.

Their main findings are that 1) in rainforest the belowground foodweb is larger in terms of biomass and energy flux than the aboveground foodweb, 2) compared, to rainforests, plantations have a similar energy flux, but somewhat less foodweb interactions, relatively less predators belowground, a higher dominance by earthworms and bacteria dominated carbon cycling.

The strengths of the manuscript are 1) a fascinating study system (foodwebs), 2) a rather complete description of the foodweb (both above and belowground, birds, canopy arthropods, and belowground arthropods and worms), and 3) the comparison of rainforest with common forms of commodity plantations.

The weaknesses are that 1) for an interested outsider it is rather difficult to follow what exactly is being reported, 2) energy fluxes are inferred from biomass stocks rather than from

measured fluxes, 3) biodiversity loss was assumed to underlie the patterns but was not measured, 4) relative short time intervals and soil volumes have been sampled, and the question how representative this is.

Response: Thank you for the thorough assessment and suggestions on how to improve our manuscript. We appreciate that you positively scored our comprehensive assessment of tropical communities, which we believe is among unique features of the study. Below we provide comments on the mentioned weaknesses of the manuscript:

(1) We now clarify our methodological approach early in the introduction (LL 102-108): “We further used these data to reconstruct food webs at each site, comprising 62 trophic guilds across all animal groups (Extended Data Table 1), and used metabolic regressions to estimate metabolic rates of each guild per unit biomass. Steady-state food-web modelling, which assumes that energetic losses are compensated by lower trophic levels, was used to calculate energy fluxes and to quantify the distribution of energy and consumption of different resources (living plants, litter, bacteria, fungi, soil organic matter, other animals) in aboveground and belowground food webs”. We also provided additional clarifications throughout the text and in the methods to make our story easier to follow.

(2) Our study investigates animal food webs and it was not feasible to have an ecosystem-scale measurement of animal respiration and animal-driven processes such as total and group-specific predation, herbivory, microbivory etc. without inference of energy fluxes from animal biomasses. We are not aware of methods allowing for in-situ measurements of animal energy fluxes at this scale. Metabolic regressions is a generic approach based on extensive empirical data, which is now widely used to quantify food-web functioning across ecosystems and wide ranges of body sizes/taxa (some literature is listed below).

3) We did not report biodiversity loss in our study since it has been repeatedly reported in previous publications, also from our study sites. We provided a list of selected references below. Unfortunately, species-level identification is not available for about half of the arthropod taxa included in our food-web calculations. This is ongoing work that for our region, takes years, if not decades, of taxonomic effort. We added quantification of already reported animal biodiversity declines to the text (LL 203-206): “Tropical land-use change has been found to result in 20-70% decline in species richness in arthropods, birds, and other taxa. Our findings show that this species decline is accompanied by fundamental functional changes in food webs and energy distribution in tropical ecosystems”.

4) Indeed, our main analysis is based on a snapshot animal assessment. To test the robustness of our results, we now repeated our analysis using an independent dataset from the ‘validation survey’ four years after the main survey. Details are described below and in the responses to Reviewer #2.

Detailed replies to all comments are given below.

Literature supporting the metabolic ecology/regressions approach:

- James H. Brown et al., ‘Toward a Metabolic Theory of Ecology’, *Ecology* 85, no. 7 (2004): 1771–89, <https://doi.org/10.1890/03-9000>.
- Richard M. Sibly, James H. Brown, and Astrid Kodric-Brown, *Metabolic Ecology: A Scaling Approach* (UK: Wiley-Blackwell, 2012).
- Andrew D. Barnes et al., ‘Energy Flux: The Link between Multitrophic Biodiversity and Ecosystem Functioning’, *Trends in Ecology & Evolution* 33, no. 3 (2018): 186–97, <https://doi.org/10.1016/j.tree.2017.12.007>.

- A. D. Barnes et al., 'Biodiversity Enhances the Multitrophic Control of Arthropod Herbivory', *Science Advances* 6, no. 45 (2020): eabb6603, <https://doi.org/10.1126/sciadv.abb6603>.

Selected literature showing species diversity decline in monoculture plantations (also now cited in the results and the methods):

- Ingo Grass et al., 'Trade-Offs between Multifunctionality and Profit in Tropical Smallholder Landscapes', *Nature Communications* 11, no. 1 (2020): 1186, <https://doi.org/10.1038/s41467-020-15013-5>.
- Anton M. Potapov et al., 'Functional Losses in Ground Spider Communities Due to Habitat-Structure Degradation under Tropical Land-Use Change', *Ecology* 101, no. 3 (2020): e02957, <https://doi.org/10.1002/ecy.2957>.
- Winda Ika Susanti et al., 'Conversion of Rainforest into Oil Palm and Rubber Plantations Affects the Functional Composition of Litter and Soil Collembola', *Ecology and Evolution* 11 (2021): 10686–708, <https://doi.org/10.1002/ece3.7881>.
- Amanda Mawan et al., 'Response of Arboreal Collembola Communities to the Conversion of Lowland Rainforest into Rubber and Oil Palm Plantations', *BMC Ecology and Evolution* 22, no. 1 (2022): 144, <https://doi.org/10.1186/s12862-022-02095-6>.
- Daniel Ramos et al., 'Rainforest Conversion to Rubber and Oil Palm Reduces Abundance, Biomass and Diversity of Canopy Spiders', *PeerJ* 10 (2022): e13898, <https://doi.org/10.7717/peerj.13898>.
- Rizky Nazarreta et al., 'Rainforest Conversion to Smallholder Plantations of Rubber or Oil Palm Leads to Species Loss and Community Shifts in Canopy Ants (Hymenoptera: Formicidae)', *Myrmecological News*, 2020, https://doi.org/10.25849/MYRMECOL.NEWS_030:175.
- Azru Azhar et al., 'Rainforest Conversion to Cash Crops Reduces Abundance, Biomass and Species Richness of Parasitoid Wasps in Sumatra, Indonesia', *Agricultural and Forest Entomology*, 2022, afe.12512, <https://doi.org/10.1111/afe.12512>.
- Walesa Edho Prabowo et al., 'Bird Responses to Lowland Rainforest Conversion in Sumatran Smallholder Landscapes, Indonesia', *PLOS ONE* 11, no. 5 (2016): e0154876, <https://doi.org/10.1371/journal.pone.0154876>.
- Kasmiatun et al., 'Rainforest Conversion to Smallholder Cash Crops Leads to Varying Declines of Beetles (Coleoptera) on Sumatra', *Biotropica* 55, no. 1 (2023): 119–31, <https://doi.org/10.1111/btp.13165>.

Please find my major and minor comments below. I hope they are of help to improve the manuscript.

With kind regards,
Lourens Poorter

MAJOR COMMENTS

1. ENERGY FLUX IS INFERRED RATHER THAN MEASURED. I am a tropical forest ecologist working on functional ecology of plants and land use change. I am therefore not familiar with foodweb research, but to be honest, I had an extremely difficult time to understand and grasp what is going on. As an interested reader I expect at least that a manuscript for Nature is directed to a generalist interested audience. So maybe this reflects my ignorance with the field, but I really do not understand how you can infer energy fluxes from standing stocks.

Response: We estimated fluxes from the standing biomass stocks and food-web topology (potential energy transfers). This approach is commonly referred to as 'steady-state food-web modelling' and described in detail in recent literature. These literature references are provided here and in the manuscript text; we also answer more specific comments in our responses below and added details to the methods.

- Andrew D. Barnes et al., 'Energy Flux: The Link between Multitrophic Biodiversity and Ecosystem Functioning', *Trends in Ecology & Evolution* 33, no. 3 (2018): 186–97, <https://doi.org/10.1016/j.tree.2017.12.007>.
- Malte Jochum et al., 'For Flux's Sake: General Considerations for Energy Flux Calculations in Ecological Communities', *Ecology and Evolution*, 14 (2021): ece3.8060, <https://doi.org/10.1002/ece3.8060>.

1.1. What I understand is that you first quantified abundance and fresh mass for each taxon (how many taxa were unidentified?),

Response: Correct. In total, 418 birds, 366,975 canopy arthropods, and 50,401 soil invertebrates were observed/collected and included in the main survey. Fresh masses were taken from the literature for birds (species-specific), and calculated using measured individual body sizes from each sample and each taxon (trophic guild) and allometric size-mass regressions for arthropods and earthworms. This information is now detailed in the methods (sections on birds, canopy arthropods, and soil invertebrates). We have species-level identifications for approximately half of the taxa. However, all taxa were identified at least to the order level. Species-level identifications were not presented in our study because it is out of our scope and these data are presented elsewhere (please see the list of papers above). Besides, with most of the species being not scientifically described in several taxa (e.g. parasitoid wasps), we can tell little from the (morpho)species name about the ecology of species at the current state of knowledge.

1.2 then estimated for each taxon its feeding guild (for how many taxa is this unknown, and how many of these taxa feed on different food sources or guilds, and if so, how do you know the relative share of each food source/guild?).

Response: Correct. All animals were allocated to general feeding guilds following existing knowledge and literature, as well as empirical data on stable isotope composition published/under review elsewhere^{5,6} (see the sections on birds, canopy arthropods, and soil invertebrates in the methods). Feeding guilds were defined by feeding preferences (relative shares of each food source/prey) to the five main resources and predation capability. Since our feeding guild classification is relatively rough, in most cases we were able to assign large

taxa to guilds. We extrapolated general knowledge on the trophic ecology of high-rank taxa (e.g. Chrysomelidae are herbivores, Staphylinidae and Araneae are predators, earthworms feed on soil, while Collembola feed on fungi) to all collected individuals within these taxa assuming phylogenetic signal in trophic niches and because information on the feeding preferences of most tropical invertebrate species is lacking. See ref 7 for justification of this approach. We now added more details on this to the methods (LL 628-632, 657-667 and 686-691):

“Guilds were defined based on feeding preferences of species (5 levels: fruits and nectar, plants and seeds, invertebrates, vertebrates and scavenging, omnivores), spatial distribution (canopy, ground foraging or multistrata), and body masses; following information obtained from a public database. In total, 11 guilds were distinguished (Extended Data Table 1; Supplementary guild table).”

“Additionally, four abundant beetle families with contrasting feeding strategies were separately analyzed from the rest of the order Coleoptera (henceforth termed ‘other Coleoptera’), i.e. Chrysomelidae, Curculionidae, Elateridae and Staphylinidae. Arthropod taxa listed above were used as trophic guilds (17 in total; Extended Data Table 1), each assigned with feeding preferences to living plants or other invertebrates and vertebrates according to existing literature⁵⁵ and unpublished data on stable isotope composition measured in the collected animals. We extrapolated general knowledge on the trophic ecology of high-rank taxa (e.g., Chrysomelidae are herbivores while Staphylinidae are predators) to all collected individuals within these taxa assuming phylogenetic signal in trophic niches and because information on the feeding preferences of most tropical invertebrate species is lacking.”

“Soil invertebrate taxa are generally consistent in their trophic niches⁷¹. However, to reflect widespread omnivory, most of them were assigned to feed on multiple basal resources (living plants, litter, bacteria, fungi, soil organic matter, other invertebrates) based on existing knowledge⁵⁵ and stable isotope composition measured in the collected animals¹⁴.”

To test if our conclusions are robust to the relative shares of each food source in the diet, we ran a sensitivity analysis, which is presented in the Extended Data Fig. 1.

1. Zhou, Z. *et al.* Tropical land use alters functional diversity of soil food webs and leads to monopolization of the detrital energy channel, *ELife* 11 (2022): e75428, <https://doi.org/10.7554/eLife.75428>.
2. Pollierer, M. M. *et al.* Rainforest conversion to plantations fundamentally alters energy fluxes and functions in canopy arthropod food webs, *under revision in Ecology Letters*
3. Anton M. Potapov, Stefan Scheu, and Alexei V. Tiunov, ‘Trophic Consistency of Supraspecific Taxa in Belowground Invertebrate Communities: Comparison across Lineages and Taxonomic Ranks’, *Functional Ecology* 33, no. 6 (2019): 1172–83, <https://doi.org/10.1111/1365-2435.13309>.

1.3 I can imagine that based on this you can create an interaction food web, but up to here is for me what you really can say with your data. You then infer energy fluxes using a multichannel foodweb approach (1595-624) which is based on many unknown assumptions. I guess for this you need to know the metabolic rate based on the biomass. These equations

are presented in Extended Data Table 3. But what is the r^2 of these equations, and if you backtransform this how much scatter do you get, so how reliable are these estimates?

Response: This is correct, the multichannel food-web reconstruction approach is based on several (simple) assumptions. These assumptions are based on food-web theory and were repeatedly supported in existing literature. Further details and references are now added to the methods “Foodweb reconstruction” section (see also the literature below; LL 712-732): “Generic rules of food-web reconstruction based on food-web theory were used to infer weighted trophic interactions among all nodes with the following assumptions: (1) There are phylogenetically inherited differences in feeding preferences for various basal resources and predation capability among soil animal taxa that define their feeding interactions (reflected as resource preferences in the raw data table); (2) predator–prey interactions are primarily defined by the optimum predator-prey mass ratio (PPMR) – typically, a predator is larger than its prey, but certain predator traits (hunting traits and behavior, parasitic lifestyle) can considerably modify the optimum PPMR. We measured body mass distribution overlap for each potential pair of predator and prey in each food web to determine the most plausible trophic interactions; (3) strength of the trophic interaction between predator and prey is defined by the overlap in their spatial niches related to vertical differentiation, with greater overlap leading to stronger interactions (i.e. no overlap among specialized canopy and soil arthropods); (4) predation is biomass-dependent – due to higher encounter rate, predators will preferentially feed on prey that are locally abundant; (5) strength of the trophic interaction between predator and prey can be considerably reduced by prey protective traits – prey with physical, chemical or behavioral protection are consumed less. All these assumptions are applied together to infer the most plausible trophic interaction matrix. For example, feeding preferences of omnivorous nodes to basal resources or other invertebrates were assigned based on literature (assumption 1), while prey selection among other invertebrates was based on size, spatial niche, total biomass, and protection of prey (assumptions 2-5).” The full food-web reconstruction R script is now given in the Supplement. Moreover, we test if our reconstruction assumptions affect our main conclusions using a sensitivity analysis (Extended Data Fig. 1) - and they do not. The r^2 values of metabolic regressions are typically high (>95%)⁸ if calculated for a wide range of body masses. Considering the wide size range of taxa included in our study (from micrograms to kilograms), we believe that this has a negligible effect on our results. We now added the standard errors for metabolic coefficients in the Extended Data Table 4 (Extended Data Table 3 in the initial version). These standard errors are generally small, except for mite groups (Mesostigmata, Oribatida, Prostigmata). However, mites accounted only for a very small proportion of energy flux in our study. Overall, we agree that our metabolic estimates can be biased especially in terms of absolute values, therefore, we focus on comparisons rather than on absolute values in our main conclusions.

4. James H. Brown et al., ‘Toward a Metabolic Theory of Ecology’, *Ecology* 85, no. 7 (2004): 1771–89, <https://doi.org/10.1890/03-9000>.

Literature justifying our food-web assumptions:

- Potapov, A. M. Multifunctionality of belowground food webs: resource, size and spatial energy channels. *Biol. Rev. Camb. Philos. Soc.* 97, 1691-1711 (2022).

- Pierre-Marc Brousseau, Dominique Gravel, and I. Tanya Handa, 'Trait Matching and Phylogeny as Predictors of Predator-Prey Interactions Involving Ground Beetles', ed. Maud Ferrari, *Functional Ecology* 32, no. 1 (2018): 192–202.
- Potapov, A. M., Scheu, S. & Tiunov, A. V. Trophic consistency of supraspecific taxa in below-ground invertebrate communities: Comparison across lineages and taxonomic ranks. *Funct. Ecol.* 33, 1172–1183 (2019).
- Brose, U. et al. Foraging theory predicts predator-prey energy fluxes. *J. Anim. Ecol.* 77, 1072–1078 (2008).
- Owen L. Petchey et al., 'Size, Foraging, and Food Web Structure', *Proceedings of the National Academy of Sciences* 105, no. 11 (2008): 4191–96.
- Brose, U. et al. Predator traits determine food-web architecture across ecosystems. *Nat Ecol Evol* 3, 919–927 (2019).
- Gauzens, B. et al. fluxweb : An R package to easily estimate energy fluxes in food webs. *Methods Ecol. Evol.* 10, 270–279 (2019).
- Peschel, K., Norton, R., Scheu, S. & Maraun, M. Do oribatid mites live in enemy-free space? Evidence from feeding experiments with the predatory mite *Pergamasus septentrionalis*. *Soil Biology and Biochemistry* 38, 2985–2989 (2006).

1.4 Is it then assumed that the biomass of all these functional groups is constant? And based on the metabolic rate you assume they need to assume a certain amount of biomass needs to be consumed, assuming a specific conversion energy? And then consumed animals are replenished out of the blue because their biomass is assumed to be constant? And if a guild is omnivorous and feeds on several other guilds, how do you infer how much biomass was consumed from other guilds? Apparently you assume average body mass values for each guild, but how much body mass variation is there within guilds and how does that determine the fluxes? You see, I am totally lost, and I have a hard time to believe that you are inferring something meaningful. I trust your values for total biomass, and foodweb complexity (Fig. 3A,C) but not for total energy flux, predation consumption ratio, and bacteria/fungi ratio (Fig 3B,D,E). I personally would confine your story to biomass and foodweb complexity and forget about the rest,

Response: The 'steady-state food-web modelling' approach is based on the assumption that energetic losses (metabolic rates and losses due to predation) are compensated by lower trophic levels in account for food assimilation efficiencies. We now detailed our explanations in the methods (LL 744-754): "The energy flux to each node was calculated from per-biomass metabolism, accounting for assimilation efficiencies (proportion of energy from food that is metabolized by the consumer) and losses to predation assuming a steady-state energetic system (i.e., energetic losses from each node are compensated by the lower trophic levels) ... Then, we applied the *fluxing* function to the reconstructed interaction networks which delivered energy flux estimations among all food-web nodes." We also introduce this approach with our hypotheses in the introduction (LL 102-106): "We further used these data to reconstruct food webs at each site, comprising 62 trophic guilds across all animal groups (Extended Data Table 1), and used metabolic regressions to estimate metabolic rates of each guild per unit biomass. Steady-state food-web modelling, which assumes that energetic losses are compensated by lower trophic levels, was used to calculate energy fluxes..."

For omnivorous guilds, we assign food-web preferences based on the assumptions listed above (the “Foodweb reconstruction” section). Details are added to the methods (LL 729-732): “For example, feeding preferences of omnivorous nodes to basal resources or other invertebrates were assigned based on literature (assumption 1), while prey selection among other invertebrates was based on size, spatial niche, total biomass, and protection of prey (assumptions 2-5).”

Indeed, we use the average body mass values for each guild, but we also account for body mass variations. Body masses and their variations were measured for each taxon/guild in each plot/food web and were used to determine potential predator-prey interactions (using Predator-Prey Mass Ratios). We now specify this in the Foodweb reconstruction section (LL 719-721): “We measured body mass distribution overlap for each potential pair of predator and prey in each food web to determine the most plausible trophic interactions”.

To test the validity of our steady-state modelling approach and account for the limited time scope of our study, we now repeated our analysis using an independent dataset from the ‘validation survey’ four years later after the main survey. This survey showed overall similar results to our main survey. Details are provided in the response to Reviewer #2, in the text and supplement. We thus believe that our energy-flux based conclusions are robust and would like to keep them.

2. UNCLEAR WHAT YOU ARE REPORTING. Again, I am a laymen, and unfortunately had really a hard time to follow what you did.

2.1 Use less jargon, define all better, and use easier terms. You talk about the ‘green’ and ‘brown’ foodweb. Why not simply ‘aboveground’ and ‘belowground’ (=litter+ground+soil) foodweb. You talk about ‘strata’. I thought about forest strata but why not simply saying above/belowground. Why are resources only living plants, leaf litter, fungi, bacteria, soil organic matter, and not the animals that are predated? What is the ‘total green energy flux’ (I98). Do you sum up everything that these groups eat amongst themselves (so the sum of the energy flux of all pairwise interactions? It took me some time to realize that the 5 resources are consumed (Fig 1) and that each of these consumers is consumed by others. Explain this better with an example in the text. What is each dot in Fig 1. A taxonomic group (hornbills, warblers) or a functional group). What are they eating in this green web? Fruits? Leaves?

Response: Thank you for these suggestions. We aligned terminology in the text to make it easier to read. Initially, we used ‘green’ and ‘brown’ terms as concise and intuitive synonyms for ‘aboveground’ and ‘belowground’ to save space. However, we agree that uniform use of a single term will make the study clearer. Thus, we now use ‘aboveground’ and ‘belowground’ throughout the text with few exceptions. We would like to keep ‘green’ and ‘brown’ in the title, because otherwise our title will not fit the Nature formatting requirements (75 characters with spaces). We also kept ‘green’ and ‘brown’ if they were used as special terms (e.g. ‘green-brown imbalance hypothesis’). Finally, we also kept ‘green’ and ‘brown’ in the abstract and introduction to synonymise these terms to ‘aboveground’ and ‘belowground’. Similarly, we replaced ‘strata’ with above- and belowground compartments throughout the text. We also now define ‘aboveground/belowground energy flux’ in the text (LL 214-216):

“Thus, we initially hypothesized that belowground energy flux (i.e., sum of all energy fluxes belowground) would be stronger in rainforests, while aboveground energy flux (i.e., sum of all energy fluxes aboveground) would be stronger in plantations.”

We use living plants, leaf litter, fungi, bacteria, soil organic matter as the main ‘basal resources’, linked to different ecosystem-level processes (primary/secondary production, C transformation, nutrient cycling). We now listed animals as another ‘resource’ in the introduction (LL 107-108) and replaced ‘Resources’ by ‘Basal resources’ in Figure 1, other figure captions, and throughout the text. We further edited Fig. 1 to improve clarity. Resources and trophic level are labelled now in the image. In Fig. 1 each point represents a trophic guild. Taxonomic names are given as examples. We added further explanation to the caption. Feeding on specific plant organs is not specified in our model. This trophic function (herbivory) represents consumption of living plant/algae material, i.e., contribution of the primary production to the energy flux.

2.2 It would be nice if you would report the biomass of each of the 4 animal groups, and their energy flux in the same graph. Or did you show that somewhere and did I miss it?

Response: These data are visualised in Fig. 2 using the trophic network figure (line size = fluxes, node size = biomasses) and are given in numbers for rainforest in the text (section ‘Aboveground and belowground rainforest food webs’). Total above- and belowground energy fluxes and biomasses across the four land-use systems are presented in Fig. 3 panels a b. We would like to avoid adding more illustrations to stay concise.

3. BIODIVERSITY LOSS IS ASSUMED TO DRIVE PATTERNS BUT NEVER REPORTED. Your justification starts with the impact of biodiversity loss for foodwebs structure and resilience, and a lot of your explanations of the foodweb patterns found is based on biodiversity loss. You never report biodiversity values with your data. It would be nice if you could provide at least data on the richness of the taxa, how this varies across the 4 land use systems, as that would make the narrative more convincing and it would make it a more complete and round story

Response: As we described in our first response, biodiversity loss in plantations in comparison to rainforest has been repeatedly reported in previous publications, also from our study sites. While we have species-level data for about a half of the studied animal groups, we believe that reporting these data is beyond the scope of our paper. Illustrating biodiversity loss with changes in the number of trophic guilds will be misleading since (1) trophic guild definition does not align well with taxonomic classifications; (2) virtually all trophic guilds in our study are present across land-use systems (but are changing in biomass and many of them have lower species richness in plantations). We now specified observed species richness losses (in %) in the text, but prefer to keep this information in the form of references to published literature.

4. HOW REPRESENTATIVE AND ACCURATE ARE YOUR DATA. I have the feeling that the effort has been massive to collect these data, but the question is how accurate these are, as bird, arthropod density can vary over the seasons and year, and a lot in space.

4.1 You measured birds for each plot 3x20 minutes, and acoustically for 4 days. Although I appreciate that you included birds, this is REALLY very little (and I guess too little). I assume that frugivores are there when plants fruit in specific parts of the season, and that bird density and activity depend on the weather pattern, and that many birds can be migratory or less vocal when breeding. If I walk 4 days in my own neighbourhood watching birds, then the bird composition varies a lot over the weeks

Response: We agree that our data cannot represent all bird species at the study sites. Unfortunately, gathering more detailed data was not feasible during the main survey. However, functional composition of communities is typically more stable than species composition⁹; i.e. despite species turnover, different bird species will perform similar roles in the food web as large-sized generalistic invertebrate feeders, or plant (seed/fruit) feeders. Thus, we believe that it is justified to retain birds in our food web model. To test the robustness of our findings, we analysed data from an independent 'validation survey', which showed overall similar results (Extended Data Text 1).

5. Anton M. Potapov et al., 'Functional Losses in Ground Spider Communities Due to Habitat-Structure Degradation under Tropical Land-Use Change', *Ecology* 101, no. 3 (2020): e02957, <https://doi.org/10.1002/ecy.2957>.

4.2 Canopy arthropods were estimated with fogging. How high did your fog go? I assume your rainforest canopy is 30-40 m tall? I do not see how the sensitivity analysis that you report can check for species that you did not measure because the fog did not go high enough

Response: According to our visual observations, fogging probably was very efficient in plantations, but could not adequately reach the highest canopies in rainforest. We didn't conduct a stratified fogging and thus cannot report community turnover in the canopies. We agree that our sensitivity analysis cannot account for vertical turnover of functional groups in canopies, however, existing studies showed overall comparable arthropod composition within rainforest canopies¹⁰. We now mention this limitation in the results (LL 124-127): 'This analysis suggested that the real energy flux aboveground (assuming uniform community composition within canopies) could be 62.0 ± 24.5 mW m⁻² in the most-severe undersampling scenario...'. This limitation, however, cannot undermine our main conclusions because it could still not explain the 14-fold above-belowground difference in energy flux.

6. Roman J Dial et al., 'Arthropod Abundance, Canopy Structure, and Microclimate in a Bornean Lowland Tropical Rain Forest', *Biotropica* 38, no. 5 (2006): 643–52.

4.3 Soil arthropods were measured in 3 subplots per plot (it is unclear if there are 3 samples per subplot). So you sampled 16cm width x 48 length x 5 cm deep. How is this going to be representative for a plot of 50x50 m?? I can imagine that the spatial turnover of these micro and meso organisms is tremendous. What is the pairwise sorenson similarity between the 3

samples? I really can not imagine how you can realistically quantify the soil arthropod community with such an approach (although the effort has been massive).

Response: Correct, each plot was represented by three soil samples (one per subplot), 16x16 cm in surface area. Despite this area being rather small in comparison to the size of the plot, such sampling schemes are very common in soil ecology due to high densities of soil invertebrates^{11,12}. For example, in previous studies, such a sampling design was sufficient to show systematic differences between, and similarities within land-use systems - both in tropical and temperate ecosystems (see list of references below). This is even more applicable in our case, since we look at the functional, rather than taxonomic composition of communities and we indeed see clear differences among land-use systems in our study. While we have heterogeneity within each plot (low pseudoreplication), relatively large true replication (n = 8 plots per system) compensate for this heterogeneity in our cross-land-use-type comparison. Which is confirmed by our validation survey.

7. Patrick Lavelle et al., 'Soil Macroinvertebrate Communities: A World-wide Assessment', *Global Ecology and Biogeography* 31, no. 7 (2022): 1261–76, <https://doi.org/10.1111/geb.13492>.
8. Anton M. Potapov et al., 'Global Monitoring of Soil Animal Communities Using a Common Methodology', *Soil Organisms* 94, no. 1 (2022): 55–68, <https://doi.org/10.25674/so94iss1id178>.

Literature comparing different land-use systems with low per plot replication:

- Maria A. Tsiafouli et al., 'Intensive Agriculture Reduces Soil Biodiversity across Europe', *Global Change Biology* 21, no. 2 (2015): 973–85, <https://doi.org/10.1111/gcb.12752>.
- Melanie M. Pollierer et al., 'Diversity and Functional Structure of Soil Animal Communities Suggest Soil Animal Food Webs to Be Buffered against Changes in Forest Land Use', *Oecologia*, (2021), <https://doi.org/10.1007/s00442-021-04910-1>.
- Winda Ika Susanti et al., 'Conversion of Rainforest into Oil Palm and Rubber Plantations Affects the Functional Composition of Litter and Soil Collembola', *Ecology and Evolution* 11 (2021): 10686–708, <https://doi.org/10.1002/ece3.7881>.
- Anton M. Potapov et al., 'Functional Losses in Ground Spider Communities Due to Habitat-Structure Degradation under Tropical Land-Use Change', *Ecology* 101, no. 3 (2020): e02957, <https://doi.org/10.1002/ecy.2957>.

4.4 The biomass and energy flux of birds and arthropods is surprisingly low, you say (this is one of your main conclusions). So what about canopy and ground mammals such as monkeys, deer, reptiles, amphibians. I guess you have missed out a lot of the aboveground mass of these big animals, and hence, their energy fluxes.

Response: We agree and discuss this limitation in our results and the 'Caveats' section (LL 160-166 and 318-323):

“Our aboveground energy flux estimates could also be biased because we did not sample all vertebrate animal groups. Amphibians, reptiles, bats, and other mammals are important invertebrate predators in tropical rainforests. However, as discussed

above, this is unlikely to change our conclusions that are based on >10-fold differences in energy fluxes, with the same applying to the potential undersampling of canopy invertebrates (Extended Data Fig. 2).”

“As we did not measure contributions by other vertebrate groups (e.g., bats and amphibians), we cannot be certain about the relative contributions of vertebrates versus invertebrates based on our data. However, including additional vertebrate groups would also increase invertebrate energy flux, as many of them feed on invertebrates, making it unlikely that this would compensate for the 12-fold difference in energy flux we detected. Overall, it is evident that rainforest food webs are energetically dominated by invertebrates and are largely ‘brown’.”

Although we are missing several vertebrate groups, our assessment captures the energetic core of the community since arthropod biomass greatly exceeds that of vertebrates¹³, and arthropods also have a higher metabolism per body mass (smaller size). Despite the fact that we can't prove this using data from our sites, we are convinced that this conclusion is robust.

9. Yinon M. Bar-On, Rob Phillips, and Ron Milo, 'The Biomass Distribution on Earth', Proceedings of the National Academy of Sciences 115, no. 25 (2018): 6506–11, <https://doi.org/10.1073/pnas.1711842115>.

5. WHAT ARE THE IMPLICATIONS? You show that compared to rainforest, plantations have a similar energy flux but a different foodweb structure. So what? Is this ‘good because the energy flux is the same? Is this ‘bad’ because the composition differs from the control? Are the systems ‘worse’ because they have less predators, and more worms and bacteria carbon cycling? Or are they simply different and are still functioning. ? Why would a foodweb with 280 interactions instead of 300 be less resilient? It still seems quite complex to me. So I am not sure what the implications are, apart from the fact that they function differently.

Response: We believe that these changes are rather ‘bad’ since they are associated with previously reported losses in the total biodiversity and decline in many ecosystem functions (e.g. ref 14). Intuitively, a system where >50% of the total energy available to animals is channelled through invasive earthworms, is not ‘healthy’. However, we prefer to avoid statements reflecting our attitude to the observed changes. Our results specifically point to the necessity of complex above-below assessments of (tropical) ecosystems to understand human effects. We focus on the energetic mechanisms and related functions that are associated with restructuring of biodiversity to inform predictions and guide actions. We now considerably expanded the last paragraph in conclusions focussing on potential implications of our results for ecosystem management (LL 350-374): “It is well documented that tropical land-use change results in animal biodiversity losses both above- and belowground. We show here that there are also associated changes in aboveground vs belowground food-web structure, basal resource consumption, and energy fluxes are distinctly different between the above- and belowground realm. We suggest that restoration and management practices in the tropics that alter the energetic balance across ecosystem compartments, taxa, size classes, and trophic levels, need to be more strongly considered and trialled. Plantations, especially oil palm, are very productive, but the available energy for maintaining multitrophic biodiversity is disproportionately low, which is associated with re-allocation of energy fluxes to basal trophic levels in belowground food webs. These findings reveal a large energetic

potential for restoring animal biodiversity in plantations. Improving belowground habitat structure via mulching and reducing herbicide use could be sufficient to partly restore soil biodiversity and energetic balance in belowground food webs. However, it may take time for the effects of these measures to become visible due to high historical inertia of the soil system. Aboveground, measures directly affecting vegetation are needed. For example, increasing canopy complexity by planting trees within monoculture plantations, and designing diverse landscapes could provide more ecological niches likely resulting in re-allocation of more energy to aboveground food webs. In the absence of restoration measures, intensive tropical land use may foster earthworm invasion belowground, further depletion of soil organic stocks, and increase risks of aboveground pest outbreaks. This is likely to result in intensification of fertilizer, herbicide, and pesticide use. Experimental studies exploring the effect of restoration measures on the energy distribution and trophic functions of food webs across above- and belowground compartments of tropical ecosystems will be crucial for better management of the energy of tropical ecosystems to sustain tropical biodiversity and ecosystem services.”

10. Claudia Dislich et al., 'A Review of the Ecosystem Functions in Oil Palm Plantations, Using Forests as a Reference System', *Biological Reviews* 92, no. 3 (2017): 1539–69, <https://doi.org/10.1111/brv.12295>.

MINOR COMMENTS

1. Better describe your land use systems. What tree sizes? What densities. I have no clue what a jungle rubber system is.

Response: We added information on the average tree height and tree density in each studied land-use system in the 'Study region and design' section.

2. L101. You state that the pervading idea is that tropical ecosystems are a green world. Why? Because most of the vegetation biomass is aboveground? Please explain and provide a reference. I thought that odum (1969) Science had predicted that old growth forests are a brown detritus based food web

Response: Rainforest canopies host diverse communities of vertebrate and invertebrate animals species and arthropod biomass (see e.g. 15). It is known that >80% of primary production is allocated to detritus in forest ecosystems. However, to our knowledge there is no comparison of energy flux in animal food webs in tropical aboveground versus belowground ecosystem compartments. We now added an explanation in the text (LL 117-118): “These figures are in contrast to the pervading idea of tropical ecosystems as a ‘green world’ with very high animal biomass in canopies.”

11. Martin D. F. Ellwood and William A. Foster, 'Doubling the Estimate of Invertebrate Biomass in a Rainforest Canopy', *Nature* 429, no. 6991 (2004): 549–51, <https://doi.org/10.1038/nature02560>.

3. L242. Why does this only apply for rubber? Palms also have a more simple canopy structure

Response: We agree that simple canopy stature cannot explain the difference in predation between rubber and oil palm. Since the high predation in rubber canopies was mainly associated with a large biomass of diptera (blood-sucking gnats and mosquitoes), it may be explained by the presence of small water bodies (rubber sap collection buckets) in this system that can host dipteran larvae. However, this speculation needs to be tested in experiments. We added this alternative explanation in the text (LL 268-271):

“High predation in rubber canopies might be associated with a simple canopy structure⁴⁴, but this does not explain low predation in oil palm. Since the high predation in rubber canopies was mainly associated with a large biomass of blood-sucking gnats and mosquitoes, it may be explained by the presence of small water bodies (rubber sap collection buckets) in rubber plantations that can host aquatic dipteran larvae.”

4. Rubber agroforests have a higher pH and therefore more worms. Were these rubber agroforests all by coincidence on a different soil type, or did the vegetation modify the soil conditions?

Response: Increase in pH with rainforest transformation to agricultural systems are caused by ash inputs after rainforest burning and liming of plantation systems. Soil type was similar across land-use systems in Harapan (loamy Acrisols) and Bukit Duabelas (clayey Acrisols) regions¹⁶

12. Kara Allen et al., ‘Spatial Variability Surpasses Land-Use Change Effects on Soil Biochemical Properties of Converted Lowland Landscapes in Sumatra, Indonesia’, *Geoderma* 284 (2016): 42–50, <https://doi.org/10.1016/j.geoderma.2016.08.010>.

5. L538. Please report how many species/taxa and individuals you measured per plot (means and the range across plots). Do the same for the canopy arthropods and the soil arthropods and worms. That gives a better understanding about your data.

Response: We now provide tables with overview of collected individuals for each animal group in the Supplementary material. We also report the total numbers of collected individuals in 2013 and 2016-2017 in the methods.

6. The discussion was sometimes difficult to follow. You report many patterns and give many different explanations. I got a bit lost. Please select your 3 main take home messages and focus on those

Response: We now added explanations in our results to improve clarity. We also simplified sections “Aboveground to belowground shift with habitat transformation” and “Predation

decline with rainforest transformation". In our conclusions we give 4 main bullet points and now added a paragraph on implications of our results (LL 333-374):

“Our study provides the first energetic description of tropical rainforest and plantation food webs across aboveground and belowground compartments, demonstrating generalities of land-use effects previously observed only in temperate ecosystems. In addition, we report novel and nuanced patterns of food-web responses depending on specific land uses and ecosystem compartments. Overall, we conclude that (1) rainforest animal communities are energetically dominated by arthropods in belowground food webs; (2) animal communities in tropical canopies suffer higher total energetic losses due to rainforest transformation than those in belowground food webs (however, the energy in belowground food webs in plantations is locked in the biomass of invasive earthworms⁹); (3) land-use change is associated with a decline in predation and an increase in relative herbivory both above- and belowground in jungle rubber and oil palm, however, the high predation in rubber suggests that crop choices can have predictable outcomes for trophic functions in food webs; (4) belowground food webs in plantations rely on different basal resources than those in rainforest, promoting faster energy channels. These changes are associated with previously observed depletion of carbon stocks¹⁷, but the causal relationships here still need to be tested.

It is well documented that tropical land-use change results in animal biodiversity losses both above- and belowground^{36,37}. We show here that there are also associated changes in aboveground vs belowground food-web structure, basal resource consumption, and energy fluxes are distinctly different between the above- and belowground realm. We suggest that restoration and management practices in the tropics that alter the energetic balance across ecosystem compartments, taxa, size classes, and trophic levels, need to be more strongly considered and trialled. Plantations, especially oil palm, are very productive¹⁷, but the available energy for maintaining multitrophic biodiversity is disproportionately low, which is associated with re-allocation of energy fluxes to basal trophic levels in belowground food webs. These findings reveal a large energetic potential for restoring animal biodiversity in plantations. Improving belowground habitat structure via mulching^{43,56} and reducing herbicide use⁵⁷ could be sufficient to partly restore soil biodiversity and energetic balance in belowground food webs. However, it may take time for the effects of these measures to become visible due to high historical inertia of the soil system. Aboveground, measures directly affecting vegetation are needed. For example, increasing canopy complexity by planting trees within monoculture plantations^{58,59}, and designing diverse landscapes³⁶ could provide more ecological niches likely resulting in re-allocation of more energy to aboveground food webs. In the absence of restoration measures, intensive tropical land use may foster earthworm invasion belowground, further depletion of soil organic stocks, and increase risks of aboveground pest outbreaks. This is likely to result in intensification of fertilizer, herbicide, and pesticide use. Experimental studies exploring the effect of restoration measures on the energy distribution and trophic functions of food webs across above- and belowground compartments of tropical ecosystems will be crucial for better management of the energy of tropical ecosystems to sustain tropical biodiversity and ecosystem services. “

Reviewer Reports on the Second Revision:

Referees' comments:

Referee #1 (Remarks to the Author):

A short comment on the proposed metric for the carbon balance. Having such a metric is indeed valuable to evaluate the soil trophic activity from an environmental perspective. But I found the choice of taking the ratio between the production of faeces (i.e., unassimilated food) and the consumption of soil organic matter by all soil invertebrates intuitively not the most clearest approach. Faecal production indicates of course the trophic activity of the food web, but in terms of material cycling faeces stays in the soil ecosystem as part of the detrital pool. For a soil ecosystem 'balance' metric I would prefer taking the ratio between carbon respiration and total consumption, as that carbon leaves the soil in the form of CO₂ emission. If possible of course, but for a rough estimate maybe enough information is available regarding energy production efficiencies (see e.g. Hunt et al. 1987).

A Comparative Analysis of Soil Fauna Populations and Their Role in Decomposition Processes
Henning Petersen, Malcolm Luxton
Oikos, Vol. 39, No. 3, Quantitative Ecology of Microfungi and Animals in Soil and Litter (Dec., 1982),
pp. 288-388 (101 pages)
<https://doi.org/10.2307/3544689>

Referee #2 (Remarks to the Author):

The paper is a much better read, presents a clear message and call for additional work.

The authors have addressed the concerns that were raised and followed up on many of the suggestions that were made in the previous reviews. I still think that connecting the observed shift in energy flux towards earthworms and the loss of soil carbon in the plantations is tenuous. The authors use earthworms as a proxy for consumption and flux within the bacterial channel as they key in on studies that show earthworms consume soils and can mineralize old soil carbon as the basis for their conclusion. However, a large body of literature has demonstrated that earthworms increase soil carbon sequestration leading to gains in soil carbon. Acknowledging this and ending on a note for further study, particularly ones that disentangle consumption by microbes and that include consumers within the bacterial channel may be the best option.

Referee #3 (Remarks to the Author):

The manuscript by Potapov compares above- and belowground foodweb structure and fluxes across 4 different tropical land use systems. Their main findings are that 1) in rainforest the belowground foodweb is larger in terms of biomass and energy flux than the aboveground foodweb, 2) compared, to rainforests, plantations have a similar energy flux, but somewhat less foodweb interactions, relatively less predators belowground, a higher dominance by earthworms and bacteria dominated carbon cycling.

As I said In my earlier review, the strengths of the manuscript are 1) a fascinating study system (foodwebs), 2) a rather complete description of the foodweb (both above and belowground, birds, canopy arthropods, and belowground arthropods and worms), and 3) the comparison of rainforest with common forms of commodity plantations.

This is the third time I review the manuscript. I appreciate the changes made by the authors and am also happy that they indicated how and where they included my comments in the first revision of the manuscript. Overall, you are nearly there and I have a few comments left. Please find my major and minor comments below.

Yours sincerely,
Lourens Poorter

MAJOR COMMENTS

“WIDESPREAD PERCEPTION OF TROPICAL ECOSYSTEMS AS A GREEN WORLD”. In my last review I said that I did not understand why you said there was a ‘pervading idea of tropical forests as a green world’. I did not know where this idea came from and to me it felt (and still feels) like a strawmen’s argument to sell the novelty of your study. The study you cited (Ellwood & Foster 2004) simply try to estimate invertebrate biomass in rainforest canopies and makes the point that basket ferns contribute a lot to this. They nowhere mention that rainforests should have less soil arthropods. You answer that ‘here is a perception of rainforests as a green world, which is clearly seen in the research focus on aboveground tropical biodiversity’. The fact that most people do study aboveground tropical diversity does not mean that they say that the aboveground foodweb is more important than the belowground foodweb. Your rephrasing to ‘widespread perception’ is very suggestive and incorrect, and you really should take this out else this idea is going to perpetuate in the literature and the belowground community will happily cite you that it is time to reshape old myths. Why not simply saying that ‘Most rainforest studies have only focused on aboveground foodwebs (e.g., ..., ..., and ...) but we show that when you include the belowground foodweb’ That is what you do, it is a great addition, and you do not need to oversell yourself and to use strawmens arguments. Change this in the abstract and the results (L136-139).

You also replied “The cited paper of Ellwood and Foster 2004 is just an example of invertebrate biomass estimation from epiphytic ferns, which was published in Nature, indicating high interest in the estimation of invertebrate biomass in canopies. At the same time, none of the estimations of the tropical soil invertebrate biomass were published in high-rank interdisciplinary journals to our knowledge.” I was quite surprised by your argumentation. I expect you to cite papers because of

their scientific quality and relevance for the topic. Please realize that because something has been published in Nature it does not by definition mean that the topic is important, the science is good, or that it is the most relevant work to cite. It is scientifically best practice to cite the most relevant work (irrespective of the impact factor or reputation of the journal), not the one that receives most media attention. This again feels like a strawmen's argument.

L45-47 'We highlight the pervasive role of land-use choices on multitrophic and multistrata functioning of tropical ecosystems and call for multidimensional restoration measures to manage energy for sustainable land use'. This is the concluding sentence of your abstract and the wider implication, and therefore very important. I have asked you 2 times before to end with a clear understandable message, and unfortunately it is still unclear to me. What do you mean with 'multistrata'? I guess it is the aboveground and belowground compartment, but at this point the generally interested reader does not know what you are referring to. What do you mean with 'multidimensional'? What should be 'restored' to what? Do you mean not using any production system and restore rainforest again? What exactly is 'To manage energy for sustainable land use'? I am lost here. Please write something more hands on and concrete how to improve the system, or else refrain from suggesting any management implications and stick with the description of the pattern and the mechanisms, which is a nice advance as well.

L263. FOOD WEB SIMPLIFICATION? You say that 'the number of trophic interactions in BOTH above and belowground webs decreased by 13-37%, showing food web simplification above- and belowground. I have problems with your word 'simplification'. I assume that you refer with the plantations to rubber and oil palm, and not to jungle rubber. First, for the number of aboveground interactions, for oil palm it is not significant from rainforest. Second, if you look at the absolute number it declines from ca 350 interactions in rainforest to 290 in rubber and 300 in oil palm. Belowground it declines from ca 500 interactions in the rainforest to 350 in rubber and 300 in plantation This is still a very high number of interactions (300!), and I would not call that simplification. The nuance is here in the wording, and what your temperate reader will remember is 'simplified systems' and (s)he involuntarily will think of the diversity of a temperate high intensity potato field. Please reword and say 'reduced diversity' as that is what it is. I think there is a lot of redundancy, and I would call 300 interactions still a well working system. So please rephrase line 263 'Therefore, soil animal communities rely on A FEW strong interactions, reflecting a documented loss of biodiversity and multifunctionality' . 300 interactions is not 'a few'. 5 is a few. 10 is a few. Maybe 20. So better say what it is 'LESS interactions (i.e., ca 300 compared to 550)'

MINOR COMMENTS

L106. Your hypothesis 2 feels like a circular argument (low predation in plantations because monocultures sustain lower predation than diverse communities). Please explain better why

L107. What are 'basal resources'? I think it is the first time you mention it here, so please define, else it is unclear for the generally interested reader what your hypothesis is about

L140. So there are no microorganisms in the canopy?? What about all the endophytes etc?

L244. 'rainforest transformation increased the belowground, but not the aboveground flux'. You

then support your statement by presenting the ratio's between the belowground and aboveground flux, but a ratio is not the same as absolute numbers. Better say that the aboveground flux declined (I trust that it is significant, but can you report a statistical test?) and that the belowground flux did not change significantly???. (In Fig 2 the sum of earthworms and soil arthropods for rainforest =296, rubber 305, oil palm=306, so that is so close that it does not feel significant to me, but it would be good if you provide a statistical test). You then can say that rainforest transformation RESULTED IN A RELATIVE INCREASE in belowground COMPARED TO aboveground fluxes

L251. Why do you think there is a 'delayed impact of land use change on below- compared to aboveground diversity'? If you see shifts in energy partitioning amongst belowground compartments after 15 years you could also see shifts in belowground diversity, isn't it? Or are worms responding fast and the microorganism responding slowly? Maybe the belowground diversity is relative inert, as long as there is energy and it is still a relatively closed forest like system (i.e., not a grassland system)? I do love rainforest and I think we really should conserve them, but please better explain your assertions.

L287. Please explain in the manuscript why diverse plant communities sustain higher predation rates than less diverse communities

L317. Please explain in the manuscript why a lower microbial biomass would lead to a higher utilization of primary food sources

L344. Please explain in the manuscript why a shift towards fast energy channel dominated foodwebs may destabilise the system. Destabilisation in terms of what?

L341-342. Here you say that increased microbial biomass production contributes to organic matter formation and stabilization, whereas a few lines earlier (line 334) you say that fast energy channel (=microbes) may accelerate depletion of carbon stocks

L407. What do you mean with 'These findings reveal a large energetic potential for restoring animal biodiversity in plantations'? I do not understand

L637. As I said last time, avoid the use of 'primary', as it is often interpreted as forest without human intervention. And humans have altered and modified the forests everywhere, even in the 6 million km² of the Amazon (C. Laevis et al. 2017 Science. Persistent effects of pre-Columbian plant domestication on Amazonian forest composition), and for sure in Indonesia, given that humans have lived there for such a long time. And for "primary degraded lowland rainforest" primary and degraded are in contradiction with each other. You reply to this saying that you used the term before in your own work. This is not a valid and convincing reason. You should use clear and consistent terminology if you convey these important messages, and not sloppy terminology. If you say instead 'degraded lowland rainforest' then everybody will still understand what you mean, and you do not need to use the word 'primary'

Author Rebuttals to Second Revision:

Referees' comments (revision3)

Referee #1 (Remarks to the Author):

A short comment on the proposed metric for the carbon balance. Having such a metric is indeed valuable to evaluate the soil trophic activity from an environmental perspective. But I found the choice of taking the ratio between the production of faeces (i.e., unassimilated food) and the consumption of soil organic matter by all soil invertebrates intuitively not the most clearest approach. Faecal production indicates of course the trophic activity of the food web, but in terms of material cycling faeces stays in the soil ecosystem as part of the detrital pool. For a soil ecosystem 'balance' metric I would prefer taking the ratio between carbon respiration and total consumption, as that carbon leaves the soil in the form of CO₂ emission. If possible of course, but for a rough estimate maybe enough information is available regarding energy production efficiencies (see e.g. Hunt et al. 1987).

A Comparative Analysis of Soil Fauna Populations and Their Role in Decomposition Processes

Henning Petersen, Malcolm Luxton

Oikos, Vol. 39, No. 3, Quantitative Ecology of Microfungi and Animals in Soil and Litter (Dec., 1982), pp. 288-388 (101 pages)

<https://doi.org/10.2307/3544689>

Response: We agree that this metric may be not optimal and should be tested. However, we prefer to keep it for a number of reasons. In steady-state energetic modelling we assume that energetic losses are compensated by gains (the assumption is discussed in LL118-121 and LL832-837). This specifically implies that consumed energy equals respired energy + unassimilated energy ('Energy rejected' in Petersen and Luxton p. 342). Respired and unassimilated energy are interconnected through the assimilation efficiency, which means that knowing one of the three parameters (consumption/respiration/rejection = faeces production), we can calculate all others. In this case we assume that unassimilated energy = energy stored. However, with this we ignore the fact that 'soil' feeders re-use energy from the 'unassimilated' pool / faeces. This repeated re-cycling loop has a great potential to change the carbon balance and this is what we aim to reflect in our metric. However, we agree that this metric requires experimental tests. Thus we have (1) specified our assumptions and refer to the non-resolved effect of soil feeders; (2) toned down the conclusions; (3) renamed 'Carbon balance' to 'Production/consumption' ratio, which is more accurate. We hope that this clarifies our approach to the reviewer and subsequent readers.

LL344-357: "To quantify animal effects on soil carbon stocks, we here calculated the ratio between the production of faeces (i.e. unassimilated food) and the consumption of soil organic matter by all soil invertebrates. It has been shown that conversion of plant materials into faeces by soil invertebrates increases microbial biomass production, which is the key process contributing to soil organic matter formation and stabilisation. In turn, invertebrates are able to mobilise and re-cycle this stored carbon while feeding on bulk soil. Supporting the link between the belowground food-web structure and net carbon loss in plantations, we

found that the production-to-consumption ratio decreased by more than 75% from 27.6 ± 29.6 in rainforest to 3.8 ± 2.9 in jungle rubber, 6.2 ± 10.4 in rubber and 2.3 ± 0.3 in oil palm plantations (Fig. 3f). Overall, our analysis suggests that changes in energy flux distribution due to habitat transformation have large functional consequences for carbon cycling, however, exact mechanisms and quantification of these animal effects over time requires dynamic ecosystem-level modelling and targeted experiments.”

LL862-874: “...(6) ratio between the production of faeces and the consumption of soil organic matter (a proxy for soil organic matter/carbon balance). The latter indicator is based on three main lines of evidence: (i) conversion of plant material into faeces by soil invertebrates increases microbial biomass production; (ii) microbial biomass production is the key process contributing to soil organic matter formation and stabilisation; (iii) consumption of soil organic matter by invertebrates (a sum of outgoing fluxes from soil organic matter) leads to consumption of associated microbial biomass, and thus has opposite effects to the first two lines of evidence. We highlight that this indicator is novel and should be validated through controlled experiments, as the effect of soil feeders on soil organic matter sequestration is context-dependent (although often negative as predicted)”

LL401-406: “(4) belowground food webs in plantations rely on different basal resources than those in rainforest, promoting faster energy channeling and shifting carbon balance from production of faeces to consumption of soil organic matter. These changes are associated with previously observed depletion of carbon stocks, but the mechanisms of animal effects in this context remains to be tested experimentally.”

Referee #2 (Remarks to the Author):

The paper is a much better read, presents a clear message and call for additional work.

The authors have addressed the concerns that were raised and followed up on many of the suggestions that were made in the previous reviews. I still think that connecting the observed shift in energy flux towards earthworms and the loss of soil carbon in the plantations is tenuous. The authors use earthworms as a proxy for consumption and flux within the bacterial channel as they key in on studies that show earthworms consume soils and can mineralize old soil carbon as the basis for their conclusion. However, a large body of literature has demonstrated that earthworms increase soil carbon sequestration leading to gains in soil carbon. Acknowledging this and ending on a note for further study, particularly ones that disentangle consumption by microbes and that include consumers within the bacterial channel may be the best option.

Response: Thank you for your positive and constructive feedback. We agree that the mechanisms behind the carbon loss and the role of earthworms in these mechanisms require further research. There is published evidence for both promotion of carbon sequestration and carbon mineralisation by earthworms. We have now explicitly stated that this is an open issue in the text:

LL343-344: “However, the net effect of earthworm feeding activity on carbon sequestration and emission remains a controversial topic in soil ecology.

LL354-357: “Overall, our analysis suggests that changes in energy flux distribution due to habitat transformation have large functional consequences for carbon cycling. However, the exact mechanisms and quantification of these animal effects over time requires dynamic ecosystem-level modelling and targeted experiments.”

LL404-406: “These changes are associated with previously observed depletion of carbon stocks, but the mechanisms of animal effects in this context remain to be tested experimentally.”

New recent references on the effects of earthworms on soil organic matter dynamics were added and cited in the text:

55. Garnier, P., Makowski, D., Hedde, M. et al. Changes in soil carbon mineralization related to earthworm activity depend on the time since inoculation and their density in soil. *Sci Rep* 12, 13616 (2022). <https://doi.org/10.1038/s41598-022-17855-z>

56. Angst, G., Frouz, J., van Groenigen, J. W., Scheu, S., Kögel-Knabner, I., & Eisenhauer, N. (2022). Earthworms as catalysts in the formation and stabilization of soil microbial necromass. *Global Change Biology*, 28, 4775–4782. <https://doi.org/10.1111/gcb.16208>

Referee #3 (Remarks to the Author):

The manuscript by Potapov compares above- and belowground foodweb structure and fluxes across 4 different tropical land use systems. Their main findings are that 1) in rainforest the belowground foodweb is larger in terms of biomass and energy flux than the aboveground foodweb, 2) compared, to rainforests, plantations have a similar energy flux, but somewhat less foodweb interactions, relatively less predators belowground, a higher dominance by earthworms and bacteria dominated carbon cycling.

As I said In my earlier review, the strengths of the manuscript are 1) a fascinating study system (foodwebs), 2) a rather complete description of the foodweb (both above and belowground, birds, canopy arthropods, and belowground arthropods and worms), and 3) the comparison of rainforest with common forms of commodity plantations.

This is the third time I review the manuscript. I appreciate the changes made by the authors and am also happy that they indicated how and where they included my comments in the first revision of the manuscript. Overall, you are nearly there and I have a few comments left. Please find my major and minor comments below.

Yours sincerely,
Lourens Poorter

MAJOR COMMENTS

“WIDESPREAD PERCEPTION OF TROPICAL ECOSYSTEMS AS A GREEN WORLD”. In my last review I said that I did not understand why you said there was a ‘pervading idea of tropical forests as a green world’. I did not know where this idea came from and to me it felt (and still feels) like a strawmen’s argument to sell the novelty of your study. The study you cited (Ellwood & Foster 2004) simply try to estimate invertebrate biomass in rainforest canopies and makes the point that basket ferns contribute a lot to this. They nowhere mention that rainforests should have less soil arthropods. You answer that ‘here is a perception of rainforests as a green world, which is clearly seen in the research focus on aboveground tropical biodiversity’. The fact that most people do study aboveground tropical diversity does not mean that they say that the aboveground foodweb is more important than the belowground foodweb. Your rephrasing to ‘widespread perception’ is very suggestive and incorrect, and you really should take this out else this idea is going to perpetuate in the literature and the belowground community will happily cite you that it is time to reshape old myths. Why not simply saying that ‘Most rainforest studies have only focused on aboveground foodwebs (e.g., ..., ..., and ...) but we show that when you include the belowground foodweb ...’ That is what you do, it is a great addition, and you do not need to oversell yourself and to use strawmens arguments. Change this in the abstract and the results (L136-139).

You also replied “The cited paper of Ellwood and Foster 2004 is just an example of invertebrate biomass estimation from epiphytic ferns, which was published in Nature, indicating high interest in the estimation of invertebrate biomass in canopies. At the same time, none of the estimations of the tropical soil invertebrate biomass were published in high-rank interdisciplinary journals to our knowledge.” I was quite surprised by your argumentation. I expect you to cite papers because of their scientific quality and relevance for the topic. Please realize that because something has been published in Nature it does not by definition mean that the topic is important, the science is good, or that it is the most relevant work to cite. It is scientifically best practice to cite the most relevant work (irrespective of the impact factor or reputation of the journal), not the one that receives most media attention. This again feels like a strawmen’s argument.

Response: We agree that ‘research focus’ is a more objective statement than ‘perceptions’ and appreciate your patience in explaining this comment. We also do not think that research published in Nature by definition is important, good, or the most relevant to cite. However, it is to be admitted that publications in high-ranking journals are often indicative for the research area to be of wide interest and these publications are more visible for the general public and scientists through various media, which creates ‘perceptions’. However, to avoid fuzzy statements and myth creations, we have now focused on rather solid facts, as suggested. The sentences were rephrased as follows:

LL32-34 (Abstract): “Despite most rainforest studies to date focussing on aboveground food webs, our results indicate that most of the energy in rainforests is channelled to the belowground animal food web.”

LL136-137 (Main text): “These figures question the existing research focus on aboveground tropical food webs and animal biomass”

L45-47 'We highlight the pervasive role of land-use choices on multitrophic and multistrata functioning of tropical ecosystems and call for multidimensional restoration measures to manage energy for sustainable land use'. This is the concluding sentence of you abstract and the wider implication, and therefore very important. I have asked you 2 times before to end with a clear understandable message, and unfortunately it is still unclear to me. What do you mean with 'multistrata'? I guess it is the aboveground and belowground compartment, but at this point the generally interested reader does not know what you are referring to. What do you mean with 'multidimensional'? What should be 'restored' to what? Do you mean not using any production system and restore rainforest again? What exactly is 'To manage energy for sustainable land use'? I am lost here. Please write something more hands on and concrete how to improve the system, or else refrain from suggesting any management implications and stick with the description of the pattern and the mechanisms, which is a nice advance as well.

Response: Thank you again for explaining your argument here. We agree with this comment and have decided to focus on the patterns and mechanisms covered in the study. The abstract now ends with:

LL43-46: "Our study shows that previously documented animal biodiversity declines with tropical land-use change are associated with vast energetic and functional restructuring in food webs across animal phyla, trophic levels, energy channels, and above- and belowground ecosystem compartments."

L263. FOOD WEB SIMPLIFICATION? You say that 'the number of trophic interactions in BOTH above and belowground webs decreased by 13-37%, showing food web simplification above- and belowground. I have problems with your word 'simplification'. I assume that you refer with the plantations to rubber and oil palm, and not to jungle rubber. First, for the number of aboveground interactions, for oil palm it is not significant from rainforest. Second, if you look at the absolute number it declines from ca 350 interactions in rainforest to 290 in rubber and 300 in oil palm. Belowground it declines from ca 500 interactions in the rainforest to 350 in rubber and 300 in plantation This is still a very high number of interactions (300!), and I would not call that simplification. The nuance is here in the wording, and what your temperate reader will remember is 'simplified systems' and (s)he involuntarily will think of the diversity of a temperate high intensity potato field. Please reword and say 'reduced diversity' as that is what it is. I think there is a lot of redundancy, and I would call 300 interactions still a well working system. So please rephrase line 263 'Therefore, soil animal communities rely on A FEW strong interactions, reflecting a documented loss of biodiversity and multifunctionality' . 300 interactions is not 'a few'. 5 is a few. 10 is a few. Maybe 20. So better say what it is 'LESS interactions (i.e., ca 300 compared to 550)'

Response: We rephrased these statements as suggested to avoid misinterpretations:

LL263-264: “The number of trophic interactions in both above- and belowground webs in plantation systems decreased by 13 to 37%, reflecting reduced biodiversity above- and belowground (Fig. 3c).”

LL265-266: “Therefore, soil animal communities in plantations rely on fewer interactions (on average -21%)...”

MINOR COMMENTS

L106. Your hypothesis 2 feels like a circular argument (low predation in plantations because monocultures sustain lower predation than diverse communities). Please explain better why

Response: We rephrased this hypothesis (LL105-107): “across trophic levels less energy is channelled to higher trophic levels in plantations because monocultures cannot sustain abundant and diverse predator communities”. Detailed explanations are provided in the specific section (“Predation decline with rainforest transformation”).

L107. What are ‘basal resources’? I think it is the first time you mention it here, so please define, else it is unclear for the generally interested reader what your hypothesis is about

Response: We have rephrased this hypothesis to make the term self-explanatory (LL107-109): “across resources at the base of the food web living plants are more important, while leaf litter is less important in plantations because of lower predation pressure, monodominant plant species, and a reduction in litterfall”

L140. So there are no microorganisms in the canopy?? What about all the endophytes etc?

Response: There are numerous endophytic and epiphytic microorganisms in canopies, especially in the tropics. Moreover, they might play some energetic role in aboveground (invertebrate) food webs. However, it is not possible to include them with the present state of knowledge because we have so little quantitative information about them. In our model, they are considered as a part of ‘living plant resources’ aboveground. We believe that in the context of this paragraph it is not necessary to open this discussion. Consideration about belowground microorganisms was added only to highlight the novelty of our finding (comparatively large belowground invertebrate contribution in the tropics).

We have now specified that endophytic and epiphytic microorganisms are included in plant resources in the model (LL 785-786): “plants (including phototrophic microorganisms and endo-/epiphytic microorganisms aboveground)”

L244. ‘rainforest transformation increased the belowground, but not the aboveground flux’. You then support your statement by presenting the ratio’s between the belowground and aboveground flux, but a ratio is not the same as absolute numbers. Better say that the aboveground flux declined (I trust that it is significant, but can you report a statistical test?) and that the belowground flux did not change significantly??? (In Fig 2 the sum of earthworms and soil arthropods for rainforest =296, rubber 305, oil palm=306, so that is so

close that it does not feel significant to me, but it would be good if you provide a statistical test). You then can say that rainforest transformation RESULTED IN A RELATIVE INCREASE in belowground COMPARED TO aboveground fluxes

Response: The sentence was corrected, thank you. Results of statistical tests are given in Fig. 3 with a Chisq anova table (system:compartment interaction) and linear mixed-effects model contrasts (asterisks indicate that the difference was significant above-, but not belowground). We have rephrased the text as follows to clarify this point:

LL243-247: “However, contrary to our hypothesis, rainforest transformation resulted in a relative increase in belowground compared to aboveground fluxes. The belowground energy flux was higher than the aboveground in rainforest (ca. 14-fold), and this difference increased in jungle rubber (ca. 30-fold), rubber (55-fold) and oil palm monocultures (68-fold), with an even higher difference in biomass (Fig. 3a,b; significant system:compartment interactions).”

L251. Why do you think there is a ‘delayed impact of land use change on below- compared to aboveground diversity’? If you see shifts in energy partitioning amongst belowground compartments after 15 years you could also see shifts in belowground diversity, isn’t it? Or are worms responding fast and the microorganism responding slowly? Maybe the belowground diversity is relative inert, as long as there is energy and it is still a relatively closed forest like system (i.e., not a grassland system)? I do love rainforest and I think we really should conserve them, but please better explain your assertions.

Response: We have specified our point, focussing on contrasting responses to land use. As suggested by other reviewers, soil inertia is the most likely explanation for such contrasts. The sentence was rephrased as:

LL253-256: “The differing energetic responses of above- and belowground systems to land use in tropical landscapes echo the recently demonstrated differences in above- and belowground biodiversity responses observed in temperate grasslands. This implies that such diverging responses might have a universal character, fitting the ‘green-brown imbalance’ hypothesis, ...”

L287. Please explain in the manuscript why diverse plant communities sustain higher predation rates than less diverse communities

Response: It has been suggested that diverse plant communities avoid resource concentrations and promote nutrient heterogeneity, which prevent (specialised) herbivores from being very abundant; at the same time, diverse plant communities provide greater refuge and resources for (generalist) predators than monocultures, which jointly sustain higher predation-to-herbivory rates. This explanation is now added to the text and supported with two references (LL288-292).

L317. Please explain in the manuscript why a lower microbial biomass would lead to a higher utilization of primary food sources

Response: In this hypothesis, we are referring to the proportional contribution of different resources to the food web. In the case of scarcity of one resource, we expect proportional increases in the others. We have now specified this expectation in the text, referring primarily to living plants (LL320-323): "...we expected proportionally higher utilisation of primary basal food resources, especially living plants, in plantations due to a decrease in alternative resources, such as microbial biomass and leaf litter."

L344. Please explain in the manuscript why a shift towards fast energy channel dominated foodwebs may destabilise the system. Destabilisation in terms of what?

Response: This argument builds on the seminal paper of Rooney and McCann where they show that fast energy channels in food webs are typically dominated by strong interactions, which tend to destabilise the system dynamics. Destabilisation in this case is a generic term that can refer e.g., to population dynamics, total community biomass, or interaction structure. We have now specified that we expect the system be more prone to perturbations due to increases in strong interactions (LL338-341): "A shift from the naturally observed balance to fast energy channel-dominated food webs may make the system more prone to perturbations (due to an increase in strong interactions) and may accelerate depletion of carbon stocks; the latter has been observed in rubber and oil palm plantations".

L341-342. Here you say that increased microbial biomass production contributes to organic matter formation and stabilization, whereas a few lines earlier (line 334) you say that fast energy channel (=microbes) may accelerate depletion of carbon stocks

Response: We believe that these two lines of argument have two important differences: (1) above, we refer to the bacterial-to-fungi ratio (with bacteria representing the 'fast' energy channel) rather than the total microbial biomass and (2) consumption (~energy channelling) can affect production, but is not exactly equivalent to production. The total microbial consumption (microbial energy channel) is contrasted here with total litter consumption and transformation, which facilitates microbial production (the litter energy channel).

L407. What do you mean with 'These findings reveal a large energetic potential for restoring animal biodiversity in plantations'? I do not understand

Response: The sentence was specified to address this (LL417-420): "The high total energy flux indicates that energy is not a limiting factor for animal biodiversity in plantations and restoration measures should focus on other ecosystem aspects. Improving belowground habitat structure via mulching and reducing herbicide use..."

L637. As I said last time, avoid the use of 'primary', as it is often interpreted as forest without human intervention. And humans have altered and modified the forests everywhere, even in the 6 million km² of the Amazon (C. Laevis et al. 2017 Science. Persistent effects of

pre-Columbian plant domestication on Amazonian forest composition), and for sure in Indonesia, given that humans have lived there for such a long time. And for “primary degraded lowland rainforest” primary and degraded are in contradiction with each other. You reply to this saying that you used the term before in your own work. This is not a valid and convincing reason. You should use clear and consistent terminology if you convey these important messages, and not sloppy terminology. If you say instead ‘degraded lowland rainforest’ then everybody will still understand what you mean, and you do not need to use the word ‘primary’

Response: The term “primary degraded lowland rainforest” is clearly defined in the classification by Margono et al. 2014 (which we refer to in the text [66]), who provide a clear definition for the state of the studied forest sites. The study of Margono et al. is an external source, which is not associated with authors of this paper. However, to avoid further unclarity, we rephrased the term to “primary [but slightly degraded] degraded lowland rainforest”.

We are deeply grateful for your thorough assessment of our work, which has greatly improved the scope and accessibility of our publication.

Reviewer Reports on the Third Revision:

Referees' comments:

Referee #1 (Remarks to the Author):

I agree with response to my questions regarding CO₂ emission versus unassimilated fraction. Herewith my comments on the manuscript are all satisfactory addressed.

Referee #3 (Remarks to the Author):

This is the third time that I review the ms. I am happy with all responses and changes made. Although it was a lengthy revision process I think it has become a stronger and better manuscript. I congratulate the authors with the final result.

I have a few tiny remarks:

- Fig 3 Why do you report for Total biomass Chi² and for the other variables an F value? Shouldn't all be F values?
- Change the legend line 286 to .. belowground compartment (C) and their interaction (S:C) on the tested parameters...
- L297. A decline of -18% is confusing. Shouldn't it be a decline of 18%?
- L325. Again, shouldn't the Chi² be a F value?
- L395-375: Good!
- L385: Change to: ... but future replanting (NORMALLY DONE AFTER X YEARS)..
- L247: Maybe good to add for sake of clarity that the aboveground energy flux declined but the belowground flux remained constant (is my interpretation right?). Then the reader understands better why the ratio changes.

With kind regards,
Lourens Poorter

Author Rebuttals to Third Revision:

Referee #3 (Lourens Poorter):

This is the third time that I review the ms. I am happy with all responses and changes made. Although it was a lengthy revision process I think it has become a stronger and better manuscript. I congratulate the authors with the final result.

I have a few tiny remarks:

- Fig 3 Why do you report for Total biomass Chi2 and for the other variables an F value? Shouldn't all be F values?

Response: We used different type (and test) of generalized linear model for animal biomass (generalized least-squares model) due to a strong heteroscedasticity of variance across above- and belowground compartments. F-type test is used in other linear models in this analysis. This information is provided in LL847-851 in methods.

- Change the legend line 286 to .. belowground compartment (C) and their interaction (S:C) on the tested parameters...

Response: Information was added to the caption.

- L297. A decline of -18% is confusing. Shouldn't it be a decline of 18%?

Response: minuses were removed from this sentence.

- L325. Again, shouldn't the Chi2 be a F value?

Response: According to the lme4 package documentation, Chi-square test is used in linear mixed-effects models.

- L395-375: Good!

Response: Thank you.

- L385: Change to: ... but future replanting (NORMALLY DONE AFTER X YEARS)..

Response: The information was added: "...but future plantation replanting (normally done 25 years)..."

- L247: Maybe good to add for sake of clarity that the aboveground energy flux declined but the belowground flux remained constant (is my interpretation right?). Then the reader understands better why the ratio changes.

Response: This is right. We clarified this in the text: "This change in the ratios resulted from reduction of the total aboveground energy flux by -75 to -79% in both monoculture plantation types in comparison to rainforest (could be up to -92% considering potential undersampling of canopy arthropods; Extended Data Fig. 2), while belowground energy flux changed little."

We again appreciate thorough assessment on the text.